# Adversarial Training Can Provably Improve Robustness: Theoretical Analysis of Feature Learning Process Under Structured Data

**Binghui Li**
Center for Machine Learning Research
Peking University
libinghui@pku.edu.cn

**Yuanzhi Li**
Machine Learning Department
Carnegie Mellon University
yuanzhil@andrew.cmu.edu

## Abstract

Adversarial training is a widely-applied approach to training deep neural networks to be robust against adversarial perturbation. However, although adversarial training has achieved empirical success in practice, it still remains unclear why adversarial examples exist and how adversarial training methods improve model robustness. In this paper, we provide a theoretical understanding of adversarial examples and adversarial training algorithms from the perspective of feature learning theory. Specifically, we focus on a multiple classification setting, where the structured data can be composed of two types of features: the robust features, which are resistant to perturbation but sparse, and the non-robust features, which are susceptible to perturbation but dense. We train a two-layer smoothed ReLU convolutional neural network to learn our structured data. First, we prove that by using standard training (gradient descent over the empirical risk), the network learner primarily learns the non-robust feature rather than the robust feature, which thereby leads to the adversarial examples that are generated by perturbations aligned with negative non-robust feature directions. Then, we consider the gradient-based adversarial training algorithm, which runs gradient ascent to find adversarial examples and runs gradient descent over the empirical risk at adversarial examples to update models. We show that the adversarial training method can provably strengthen the robust feature learning and suppress the non-robust feature learning to improve the network robustness. Finally, we also empirically validate our theoretical findings with experiments on real-image datasets, including MNIST, CIFAR10 and SVHN.

## 1 Introduction

Recently, large-scale neural networks have achieved remarkable performance in many disciplines, especially in computer vision (Krizhevsky et al., 2012; Dosovitskiy et al., 2021; Kirillov et al., 2023) and natural language processing (Kenton & Toutanova, 2019; Brown et al., 2020; Ouyang et al., 2022; Achiam et al., 2023). However, it is well-known that neural networks are vulnerable to small but adversarial perturbations, i.e., natural data with strategic perturbations called adversarial examples (Biggio et al., 2013; Szegedy et al., 2013; Goodfellow et al., 2014), which can confuse well-trained network classifiers. This potentially leads to reliability and security issues in real-world applications.

To mitigate this problem, one seminal approach to improve robustness of models is called adversarial training (Goodfellow et al., 2014; Madry et al., 2018; Shafahi et al., 2019; Zhang et al., 2019; Pang et al., 2022; Wang et al., 2023), which iteratively generates adversarial examples from the training data and updates the model with these adversarial examples rather than the original training examples.

However, despite the significant empirical success of adversarial training in enhancing the robustness of neural networks across various datasets, the theoretical understanding of adversarial examples and adversarial training still remains unclear, particularly from the perspective of network optimization. Therefore, we ask the following fundamental theoretical questions:

**Q1:** *Why do neural networks trained with standard training tend to converge to non-robust solutions that fail to classify adversarially-perturbed data?*

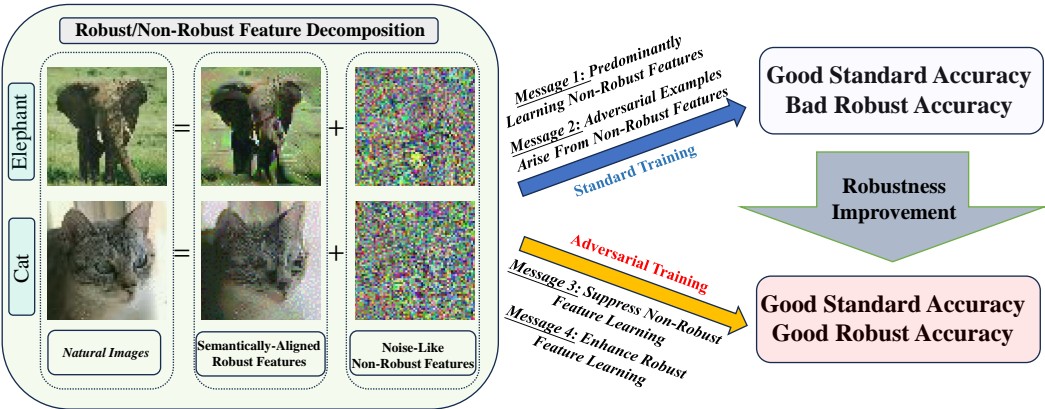

Figure 1: **An overview of our paper:** robust/non-robust-feature-decomposition-based framework and key messages about standard/adversarial training. And the robust/non-robust features of elephant and cat are generated in the same way of Ilyas et al. (2019) from random noise to ImageNet instances.

**Q2:** *How does the adversarial training algorithm assist in optimizing neural networks to enhance their robustness against adversarial perturbations?*

Indeed, we emphasize that a common challenge in analyzing adversarial robustness is the gap between theory and practice, primarily attributed to the data assumptions in theoretical frameworks (see detailed discussion in Section 1.1), which motives us considering realistic data model. In our paper, the data foundation that we leverage is predicated on the decomposition of robust and non-robust features, which suggests that data is comprised of two distinct types of features: *robust features*, characterized by their strength yet sparsity, and *non-robust features*, noted for their vulnerability yet density (Assumption 2.3). This decomposition has been empirically investigated in a series of previous studies (Tsipras et al., 2019; Ilyas et al., 2019; Kim et al., 2021; Tsilivis & Kempe, 2022; Han et al., 2023). Furthermore, we mathematically represent this concept as the *patch-structured data* proposed in the recent work of Allen-Zhu & Li (2023a), in which they utilize multi-view-based patch-structured data to provide a fruitful setting for theoretically understanding the benefits of ensembles in deep learning. Specifically, inspired by Allen-Zhu & Li (2023a) and Ilyas et al. (2019), we propose a novel patch-structured data model based on *robust/non-robust feature decomposition* (Definition 2.2), and show our patch data model enables us to rigorously establish the existence of adversarial examples and demonstrate the efficacy of adversarial training by directly analyzing the feature learning process for two-layer networks under our structured data. More precisely, the main results in our work are summarized as follows:

- By analyzing the feature learning process on *robust/non-robust feature decomposition based data*, we demonstrate that in standard training, the neural network *predominantly learns non-robust features* rather than robust features (Theorem 4.3). This leads to the generation of adversarial examples with perturbations *stemming from these non-robust features*

- Furthermore, we show that adversarial training algorithms can provably both *suppress* the learning of non-robust features and *enhance* the learning of robust features (Theorem 4.4), thereby improving models robustness.

- We also substantiate the theoretical findings about robust and non-robust feature learning discussed in Section 4 through a series of experiments conducted on real-image datasets (MNIST, CIFAR10, and SVHN). Detailed results can be viewed in Figure 4.

## 1.1 RELATED WORKS

**Theoretical Explanations for Adversarial Examples.** A line of works (Daniely & Shacham, 2020; Bubeck et al., 2021a; Bartlett et al., 2021; Montanari & Wu, 2023) demonstrates the existence of adversarial examples in random-weight neural networks with various architectures. Another line of works (Bubeck et al., 2021b; Bubeck & Sellke, 2021; Li et al., 2022; Li & Li, 2023) suggests that

over-parameterization is necessary to achieve robustness, and that non-robustness may stem from the expressive power of neural networks. However, these works do not consider the optimization process when explaining adversarial examples in trained networks. Recently, Frei et al. (2024) and Li et al. (2024) proved that, for two-layer ReLU networks, gradient method leads to well-generalizing but non-robust solutions under a synthetic multi clusters data assumption. In our paper, we analyze the feature learning process under a more realistic structured data model inspired by the robust/non-robust feature decomposition proposed in Ilyas et al. (2019). Indeed, we not only prove that standard training causes a two-layer neural network to converge to a non-robust solution, but also rigorously analyze how adversarial training algorithm provably guides the network towards a robust solution.

**Theoretical Understanding of Adversarial Training for Linear Models.** A series of works (Li et al., 2020; Javanmard & Soltanolkotabi, 2022; Chen et al., 2023) demonstrate that a linear classifier trained through adversarial training can achieve robustness under the Gaussian-mixture data model. However, standard training does not explicitly converge to non-robust solutions under these conditions. This discrepancy does not align with the empirical observation that networks trained by standard methods exhibit poor robust performance (Biggio et al., 2013; Szegedy et al., 2013; Goodfellow et al., 2014). For example, as noted in Chen et al. (2023), similar to standard training, adversarial training also directionally converges to the maximum $\ell_2$-margin solution when considering a Gaussian-mixture data model with $\ell_2$ perturbations. This suggests that, under their settings, standard training alone can achieve adversarial robustness due to the maximum-margin implicit bias, even though neural networks trained with standard training typically exhibit non-robustness in practice. In our paper, to bridge the gap between theory and practice, we consider a more structured data assumption and apply a non-linear two-layer CNN as the learner, which ensures that both robust global minima and non-robust global minima exist due to the non-linearity of our data model and and non-convexity of our learner model (see a detailed discussion in Section 3).

**Feature Learning Theory of Deep Learning.** The feature learning theory of neural networks, as proposed in various recent studies (Wen & Li, 2021; Allen-Zhu & Li, 2022; Chen et al., 2022; Jelassi et al., 2022; Chidambaram et al., 2023; Allen-Zhu & Li, 2023b;a; Lu et al., 2024; Chen et al., 2024; Li et al., 2024), aims to explore how features are learned in deep learning tasks. This theory extends the theoretical optimization analysis paradigm beyond the scope of the neural tangent kernel (NTK) theory (Jacot et al., 2018; Du et al., 2019b;a; Allen-Zhu et al., 2019; Arora et al., 2019). Based on the sparse coding model, Allen-Zhu & Li (2022) consider a binary robust classification problem and proposes a principle called feature purification to explain the workings of adversarial training. In our paper, we focus on a multiple robust classification problem by leveraging more image-like, patch-structured data with an assumption of robust/non-robust feature decomposition. We study how the feature learning process differs when applying adversarial training instead of standard training.

## 2 PROBLEM SETUP

### 2.1 NOTATIONS

Throughout this work, we use letters for scalars and bold letters for vectors. For any given two sequences $\{A_n\}_{n=0}^{\infty}$ and $\{B_n\}_{n=0}^{\infty}$, we denote $A_n = O(B_n)$ if there exist some absolute constant $C_1 > 0$ and $N_1 > 0$ such that $|A_n| \leq C_1 |B_n|$ for all $n \geq N_1$. Similarly, we denote $A_n = \Omega(B_n)$ if there exist $C_2 > 0$ and $N_2 > 0$ such that $|A_n| \geq C_2 |B_n|$ for all $n > N_2$. We say $A_n = \Theta(B_n)$ if $A_n = O(B_n)$ and $A_n = \Omega(B_n)$ both holds. We use $\widetilde{O}(\cdot), \widetilde{\Omega}(\cdot)$, and $\widetilde{\Theta}(\cdot)$ to hide logarithmic factors in these notations respectively. Moreover, we denote $A_n = \text{poly}(B_n)$ if $A_n = O(B_n^K)$ for some positive constant $K$, and $A_n = \text{polylog}(B_n)$ if $B_n = \text{poly}(\log(B_n))$. We say $A_n = o(B_n)$ (or $A_n \ll B_n$ or $B_n \gg A_n$) if for arbitrary positive constant $C_3 > 0$, there exists $N_3 > 0$ such that $|A_n| < C_3|B_n|$ for all $n > N_3$. And we also use $A_n \approx B_n$ to denote $A_n = B_n + o(1)$.

### 2.2 DATA DISTRIBUTION

In this paper, we consider a $k$-class classification problem involving data that is structured into $P$ patches, with each patch having a dimension of $d$. Specifically, each labeled data point is represented by $(\boldsymbol{X}, y)$, where $\boldsymbol{X} = (\boldsymbol{x}_1, \boldsymbol{x}_2, \ldots, \boldsymbol{x}_P) \in (\mathbb{R}^d)^P$ denotes the data vector, and $y \in [k]$ signifies the data label. We first present the formal definition of robust and non-robust features as follows.

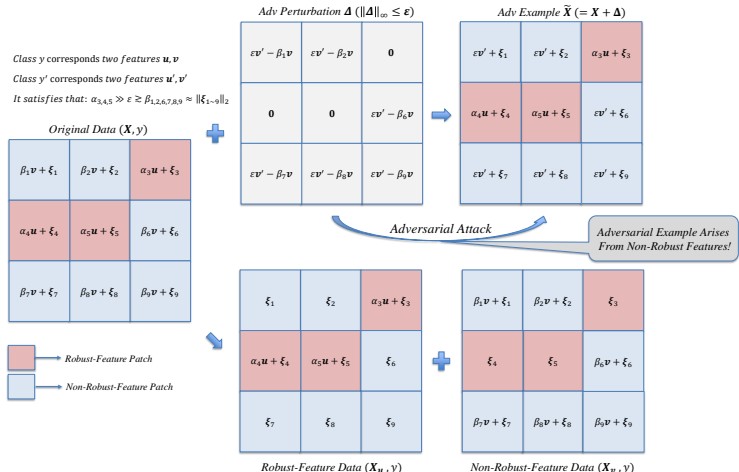

Figure 2: **Illustration of our patch data:** Each patch in data point $(\boldsymbol{X}, y)$ has the form $\boldsymbol{x}_p = \alpha_p \boldsymbol{u} + \boldsymbol{\xi}_p$ (robust-feature patch) or $\boldsymbol{x}_p = \beta_p \boldsymbol{v} + \boldsymbol{\xi}_p$ (non-robust-feature patch), where $\boldsymbol{u}, \boldsymbol{v}$ are the corresponding features for class $y$. For non-robust-feature patches, adversarial perturbation $\boldsymbol{\Delta}$ replaces non-robust feature $\boldsymbol{v}$ with other non-robust feature $\boldsymbol{v}'$ (corresponding to other class $y'$), which causes adversarial example $\tilde{\boldsymbol{X}}$ with incorrect label $y'$ when the network learner trained by standard training mainly learns non-robust features $\boldsymbol{v}, \boldsymbol{v}'$ rather than robust features $\boldsymbol{u}, \boldsymbol{u}'$. And we construct robust-feature/non-robust-feature data $\boldsymbol{X}_{\boldsymbol{u}}/\boldsymbol{X}_{\boldsymbol{v}}$ by replacing $\boldsymbol{v}/\boldsymbol{u}$ with all-zero vector $\boldsymbol{0}$.

**Definition 2.1** (Robust and Non-robust Features). We assume that each label class $j \in [k]$ is associated with two types of features, for the sake of mathematical simplicity, represented as two feature vectors $\boldsymbol{u}_j$ (robust feature) and $\boldsymbol{v}_j$ (non-robust feature), both in $\mathbb{R}^d$. For notation simplicity, we also assume that all the features are orthonormal and parallel to the coordinate axes. Namely, the set of all features is defined as

$$\mathcal{F} := \{\boldsymbol{u}_j, \boldsymbol{v}_j\}_{j \in [k]},$$

which satisfies that

$$\forall \boldsymbol{f} \in \mathcal{F}, \|\boldsymbol{f}\|_2 = \|\boldsymbol{f}\|_\infty = 1 \quad and \quad \forall \boldsymbol{f} \neq \boldsymbol{f}' \in \mathcal{F}, \boldsymbol{f} \perp \boldsymbol{f}'.$$

For simplicity, we focus on the case when the dimension of patch $d$ is sufficiently large (i.e. we assume $d = \text{poly}(k)$ for a large polynomial) such that all $2k$ features can be orthogonal in the space $\mathbb{R}^d$. And we use "with high probability" to denote with probability at least $1 - e^{-\Omega(\log^2 d)}$.

Now, we give the following robust/non-robust-feature-decomposition-based patch-structured data distribution and some assumptions about it.

**Definition 2.2** (Patch Data Distribution). Each data pair $(\boldsymbol{X}, y) \in \left(\mathbb{R}^d\right)^P \times [k]$ is generated from the distribution $\mathcal{D}$ with latent distributions $\{(\mathcal{D}_{\mathcal{J},y}, \mathcal{D}_{\alpha,y}, \mathcal{D}_{\beta,y})\}_{y \in [k]}$, where $\mathcal{D}_{\mathcal{J},y}$ is a probability distribution over all 2-partitions of $[P]$ and $\mathcal{D}_{\alpha,y}, \mathcal{D}_{\beta,y}$ are two distributions over the positive real number. Then, it generates data points as follows.

1. The label $y$ is uniformly drawn from $[k]$.

2. Uniformly draw the two-type patch index sets $(\mathcal{J}_R, \mathcal{J}_{NR}) \subset [P] \times [P]$ from the distribution $\mathcal{D}_{\mathcal{J},y}$, where $\mathcal{J}_R$ and $\mathcal{J}_{NR}$ corresponds to the robust-feature patches and non-robust feature patches such that $\mathcal{J}_R \cup \mathcal{J}_{NR} = [P]$ and $\mathcal{J}_R \cap \mathcal{J}_{NR} = \emptyset$.

3. For each $p \in [P]$, the corresponding patch vector is generated as

$$\boldsymbol{x}_p := \begin{cases} \alpha_p \boldsymbol{u}_y + \boldsymbol{\xi}_p, & \text{if } p \in \mathcal{J}_R \quad \text{(robust-feature patch)} \\ \beta_p \boldsymbol{v}_y + \boldsymbol{\xi}_p, & \text{if } p \in \mathcal{J}_{NR} \quad \text{(non-robust-feature patch)} \end{cases}$$

where $\alpha_p, \beta_p > 0$ are the random coefficients sampled from the distribution $\mathcal{D}_{\alpha,y}, \mathcal{D}_{\beta,y}$ respectively, and $\boldsymbol{\xi}_p \sim \mathcal{N}(\boldsymbol{0}, \sigma_n^2 \mathcal{I}_d)$ is the random Gaussian noise with variance $\sigma_n^2$.

**Assumption 2.3.** We suppose that the following conditions holds for the data distribution $\mathcal{D}$. In a data point $(\boldsymbol{X}, y)$ sampled from $\mathcal{D}$, with high probability, it satisfies:

- Robust feature is stronger than non-robust feature: $\forall (p, p') \in \mathcal{J}_R \times \mathcal{J}_{NR}, \alpha_p \gg \beta_{p'}$.

- Non-robust feature is denser than robust feature: $\exists \tau \geq 0, \sum_{p \in \mathcal{J}_R} \alpha_p^\tau \ll \sum_{p \in \mathcal{J}_{NR}} \beta_p^\tau$.

**Regarding the first condition of Assumption 2.3.** We further assume that $\alpha_p, \beta_p$ concentrate on their expectations (i.e., w.h.p. $\alpha_p \approx \mathbb{E}[\alpha_p], \beta_p \approx \mathbb{E}[\beta_p]$) and $\mathbb{E}[\alpha_p] \gg \mathbb{E}[\beta_p] = \Theta(\sigma_n \sqrt{d})$ for simplicity. Then, we know, with high probability over a sampled data, it holds that $\forall (p, p') \in \mathcal{J}_R \times \mathcal{J}_{NR}, \alpha_p \gg \beta_p \approx \|\boldsymbol{\xi}_{p'}\|_2 \approx \|\boldsymbol{\xi}_p\|_2 \approx \sigma_n \sqrt{d}$, which means that robust-feature patches $\boldsymbol{x}_p = \alpha_p \boldsymbol{u}_y + \boldsymbol{\xi}_p \approx \alpha_p \boldsymbol{u}_y$ appear more prominent, but non-robust-feature patches $\boldsymbol{x}_{p'} = \beta_p \boldsymbol{v}_y + \boldsymbol{\xi}_{p'}$ are noise-like.

**Regarding the second condition of Assumption 2.3.** Here, $\tau$ is an absolutely constant. We notice that when $\tau = 0$, it implies w.h.p. $|\mathcal{J}_R| \ll |\mathcal{J}_{NR}|$, which manifests that non-robust-feature patches are denser than robust-feature patches. And we assume $\tau \geq 3$ for simplifying our mathematical analysis.

**Our Patch Data Aligns with Realistic Images.** In all, it shows that Assumption 2.3 can be tied to a down-sized version of convolutional networks applied to image classification data. With a small kernel size, high-magnitude good features that are easily perceivable by humans in an image typically appear only at a few patches (such as the ears of a cat or the nose of an elephant), and most other patches look like random noise to human observers (such as the textures of cats and elephants blended into a random background). See illustrations of real images and our patch data in Figures 1 and 2.

**Remark 2.4.** *Previous empirical works of Ilyas et al. (2019); Kim et al. (2021) characterize robust features as useful for both clean and robust classification, whereas non-robust features, although helpful for clean classification, fail in robust scenarios. Consistent with these observations, our definitions and assumptions suggest that networks leveraging robust features $\{\boldsymbol{u}_i\}_{i \in [k]}$ act as robust classifiers, while those relying on non-robust features $\{\boldsymbol{v}_i\}_{i \in [k]}$ perform well on clean data but not on perturbed data, as detailed in Propositions 3.1 and 3.2 in Section 3.*

## 2.3 NETWORK LEARNER

We consider the setting of learning the data distribution $\mathcal{D}$ by applying the same two-layer convolutional architecture used in Allen-Zhu & Li (2023a) with the following smoothed ReLU activation.

**Activation.** For integer $q \geq 2$ and threshold $\varrho$, the smoothed ReLU is defined as $\widetilde{\mathrm{ReLU}}(z) := 0$ for $z \leq 0$; $\widetilde{\mathrm{ReLU}}(z) := \frac{z^q}{q\varrho^{q-1}}$ for $z \in [0, \varrho]$; and $\widetilde{\mathrm{ReLU}}(z) := z - \left(1 - \frac{1}{q}\right)\varrho$ for $z \geq \varrho$.

$\widetilde{\mathrm{ReLU}}$ addresses the non-smoothness of original ReLU function at zero. We focus on the case when $q = 3$ and $\varrho = \frac{1}{\mathrm{polylog}(d)}$ for simplicity, while our result indeed applies to other constants $q \in [3, \tau]$.

**Network Model.** For the $k$-class classification task, we consider the following two-layer convolutional neural network as $\boldsymbol{F}(\boldsymbol{X}) = (F_1(\boldsymbol{X}), F_2(\boldsymbol{X}), \ldots, F_k(\boldsymbol{X})) : \left(\mathbb{R}^d\right)^P \to \mathbb{R}^k$, and $F_i(\boldsymbol{X})$ denotes

$$F_i(\boldsymbol{X}) := \sum_{r \in [m]} \sum_{p \in [P]} \widetilde{\mathrm{ReLU}}(\langle \boldsymbol{w}_{i,r}, \boldsymbol{x}_p \rangle),$$

where $\{\boldsymbol{w}_{i,r} \in \mathbb{R}^d\}_{(i,r) \in [k] \times [m]}$ are learnable weights for different convolutional filters. We set the width $m = \mathrm{polylog}(d)$ to achieve mildly over-parameterization for efficient optimization purpose.

## 2.4 STANDARD TRAINING

**Training Objective.** During the standard training, we learn the concept class (namely, the labeled data distribution $\mathcal{D}$) by minimizing the cross-entropy loss function $\mathcal{L}_{CE}$ using $N = \mathrm{poly}(d)$ training data points $\mathcal{Z} = \{(\boldsymbol{X}_i, y_i)\}_{i \in [N]}$ randomly sampled from $\mathcal{D}$. By denoting $\mathcal{L}_{CE}(\boldsymbol{F}; \boldsymbol{X}, y) := -\log \frac{e^{F_y(\boldsymbol{X})}}{\sum_{j \in [k]} e^{F_j(\boldsymbol{X})}}$, we use the empirical loss $\mathcal{L}_{CE}(\boldsymbol{F}) := \mathbb{E}_{(\boldsymbol{X}, y) \sim \mathcal{Z}}[\mathcal{L}_{CE}(\boldsymbol{F}; \boldsymbol{X}, y)]$ as objective.

**Network Initialization.** We randomly initialize the network $\boldsymbol{F}$ by letting each $\boldsymbol{w}_{i,r}^{(0)} \sim \mathcal{N}\left(0, \sigma_0^2 \mathcal{I}_d\right)$ for $\sigma_0^2 = \frac{1}{d}$, which is the standard Xavier initialization (Glorot & Bengio, 2010).

**Training Algorithm.** To train the model, at each iteration $t$ we update using the gradient descent (GD) with small learning rate $\eta \leq \frac{1}{\text{poly}(d)}$: $\boldsymbol{w}_{i,r}^{(t+1)} = \boldsymbol{w}_{i,r}^{(t)} - \eta \mathbb{E}_{(\boldsymbol{X},y)\sim\mathcal{Z}} \left[ \nabla_{\boldsymbol{w}_{i,r}} \mathcal{L}_{CE}\left(\boldsymbol{F}^{(t)}; \boldsymbol{X}, y\right) \right]$,

where we run the algorithm for $T = \frac{\text{poly}(d)}{\eta}$ iterations. We use $\boldsymbol{F}^{(t)}$ to denote the model at iteration $t$.

## 2.5 ADVERSARIAL TRAINING

$\ell_p$**-Adversarial Robustness.** In our work, we consider $\ell_p$-robustness within a perturbation radius $\epsilon > 0$, especially the $\ell_\infty$-norm on which we focus henceforth for notation simplicity. Our main results can be easily extended to other $\ell_p$-norm case ($p \geq 2$).

**Small Perturbation Radius.** In our setting, we choose $\epsilon = \Theta(\sigma_n\sqrt{d})$. Then, for two data points $(\boldsymbol{X}, y), (\boldsymbol{X}', y') \sim \mathcal{D}$ with distinct labels $y \neq y' \in [k]$, it can be checked that w.h.p. $\|\boldsymbol{X} - \boldsymbol{X}'\|_\infty \gg \Theta(\sigma_n\sqrt{d}) = \Theta(\epsilon)$, which is consistent with the empirical observation that typical perturbation radius is often much smaller than the separation distance between different classes (Yang et al., 2020).

**Adversarial Example.** For a given network $\boldsymbol{F} := (F_1, F_2, \ldots, F_k)$ and a data point $(\boldsymbol{X}, y)$, we say that $\widetilde{\boldsymbol{X}}$ is an adversarial example (Szegedy et al., 2013) if the classifier predicts a wrong label for it (i.e. $\text{argmax}_{j\in[k]} F_j(\widetilde{\boldsymbol{X}}) \neq y$) and the perturbation $\boldsymbol{\Delta} := \widetilde{\boldsymbol{X}} - \boldsymbol{X}$ satisfies $\|\boldsymbol{\Delta}\|_\infty \leq \epsilon$.

**Adversarial Training Algorithm.** During adversarial training, we first find the adversarial examples $(\widetilde{\boldsymbol{X}}, y)$ by one-step gradient ascent with learning rate $\tilde{\eta}$ ($\gg \eta$) over the margin loss $\mathcal{L}_{margin}(\boldsymbol{F}; \boldsymbol{X}, y)$ using training data points $(\boldsymbol{X}, y) \in \mathcal{Z}$. Here, we choose $\mathcal{L}_{margin}(\boldsymbol{F}; \boldsymbol{X}, y) := -F_y(\boldsymbol{X})$ for simplicity, and our theoretical analysis can also be extended to the standard margin-based adversarial-attack objective function $\mathcal{L}_{margin}(\boldsymbol{F}; \boldsymbol{X}, y) := -\left(F_y(\boldsymbol{X}) - \max_{j\in[k]\setminus\{y\}} F_j(\boldsymbol{X})\right)$ (Carlini & Wagner, 2017; Gowal et al., 2019; Sriramanan et al., 2020). And then we train the network parameters $\{\boldsymbol{w}_{i,r}\}_{(i,r)\in[k]\times[m]}$ by taking gradient descent over the adversarial loss $\mathbb{E}_{(\boldsymbol{X},y)\sim\mathcal{Z}} \left[ \mathcal{L}_{CE}\left(\boldsymbol{F}; \widetilde{\boldsymbol{X}}, y\right) \right]$.

Concretely, the adversarial examples $\{(\widetilde{\boldsymbol{X}}^{(t)}, y)\}$ and the network $\boldsymbol{F}^{(t)}$ are updated alternatively as

$$\begin{cases} \widetilde{\boldsymbol{X}}^{(t)} = \boldsymbol{X} + \text{Clip}_{\infty,\epsilon}\left(\tilde{\eta}\nabla_{\boldsymbol{X}}\mathcal{L}_{margin}\left(\boldsymbol{F}^{(t)}; \boldsymbol{X}, y\right)\right), & \forall(\boldsymbol{X}, y) \in \mathcal{Z}, \\ \boldsymbol{w}_{i,r}^{(t+1)} = w_{i,r}^{(t)} - \eta\mathbb{E}_{(\boldsymbol{X},y)\sim\mathcal{Z}}\left[\nabla_{\boldsymbol{w}_{i,r}}\mathcal{L}_{CE}\left(\boldsymbol{F}^{(t)}; \widetilde{\boldsymbol{X}}^{(t)}, y\right)\right], & \forall(i, r) \in [k] \times [m], \end{cases}$$

where $\epsilon > 0$ is the $\ell_\infty$-perturbation radius and patch-wise clip function $\text{Clip}_{\infty,\epsilon}(\cdot)$ is used to enable $\widetilde{\boldsymbol{X}}^{(t)} \in \mathbb{B}_\infty(\boldsymbol{X}, \epsilon)$, which is defined as, for the clip radius $\rho > 0$ and a given flattened patch data $\boldsymbol{Z} = (z_1, z_2, \ldots, z_{Pd}) \in \left(\mathbb{R}^d\right)^P$, $\text{Clip}_{\infty,\rho}(\boldsymbol{Z}) := (\tilde{z}_1, \tilde{z}_2, \ldots, \tilde{z}_{Pd}), \tilde{z}_j = \frac{z_j}{\max\{1, \|z_j\|_\infty/\rho\}}, \forall j \in [Pd]$.

**Remark 2.5.** *Indeed, a more general form of adversarial example update in adversarial training algorithm is to directly maximize the loss value over the perturbed data point, i.e.*

$$\widetilde{\boldsymbol{X}}^{(t)} = \boldsymbol{X} + \text{argmax}_{\|\boldsymbol{\Delta}\|_\infty\leq\epsilon}\mathcal{L}_{margin}\left(\boldsymbol{F}^{(t)}; \boldsymbol{X} + \boldsymbol{\Delta}, y\right) \quad \forall(\boldsymbol{X}, y) \in \mathcal{Z}.$$

*However, different from some previous works (Li et al., 2020; Javanmard & Soltanolkotabi, 2022; Chen et al., 2023) that study training dynamics of adversarial training under linear classifier, we are unable to derive the closed-form solution of adversarial examples due to the high non-linearity and non-convexity of the objective function $\mathcal{L}_{margin}\left(\boldsymbol{F}^{(t)}; \boldsymbol{X} + \boldsymbol{\Delta}, y\right)$ over $\boldsymbol{\Delta}$. To overcome this challenge, we use one-step gradient ascent method to approximate the optimal solution.*

## 3 WARM UP: THERE EXIST BOTH NON-ROBUST AND ROBUST GLOBAL MINIMA

In this section, as a warm up, we show that there exist both robust global minima and non-robust global minima due to the non-convexity of empirical risk $\mathcal{L}_{CE}(\boldsymbol{F})$ over the parameters $\{\boldsymbol{w}_{i,r}\}_{(i,r)\in[k]\times[m]}$.

**Proposition 3.1** (The Existence of Non-robust Global Minima). *We consider the special case when $m = 1$ and $\boldsymbol{w}_{i,1} = \gamma\boldsymbol{v}_i$, where $\gamma > 0$ is a scale coefficient. Then, it holds that the standard empirical risk satisfies $\lim_{\gamma\to\infty}\mathcal{L}_{CE}(\boldsymbol{F}) = o(1)$, but the adversarial test error satisfies $\lim_{\gamma\to\infty}\mathbb{P}_{(\boldsymbol{X},y)\sim\mathcal{D}}\left[\exists\boldsymbol{\Delta} \in \left(\mathbb{R}^d\right)^P \text{ s.t. } \|\boldsymbol{\Delta}\|_\infty \leq \epsilon, \text{argmax}_{i\in[k]} F_i(\boldsymbol{X} + \boldsymbol{\Delta}) \neq y\right] = 1 - o(1)$.*

**Proposition 3.2** (The Existence of Robust Global Minima). *We consider the special case when $m = 1$ and $\boldsymbol{w}_{i,1} = \gamma \boldsymbol{u}_i$, where $\gamma > 0$ is a scale coefficient. Then, it holds that the standard empirical risk satisfies $\lim_{\gamma \to \infty} \mathcal{L}_{CE}(\boldsymbol{F}) = o(1)$, and the adversarial test error satisfies $\lim_{\gamma \to \infty} \mathbb{P}_{(\boldsymbol{X},y)\sim\mathcal{D}} \left[ \exists \boldsymbol{\Delta} \in (\mathbb{R}^d)^P \text{ s.t. } \|\boldsymbol{\Delta}\|_\infty \le \epsilon, \operatorname{argmax}_{i\in[k]} F_i(\boldsymbol{X} + \boldsymbol{\Delta}) \ne y \right] = o(1).$*

Proposition 3.1 and Proposition 3.2 demonstrate that a network is vulnerable to adversarial perturbations if it relies solely on learning non-robust features. Conversely, a network that learns all robust features can achieve a state of robustness. In general, by calculating the gradient of empirical loss, it seems that the whole weights during gradient-based training will have the following form

$$\boldsymbol{w}_{i,r} \approx A_{i,r}\boldsymbol{u}_i + B_{i,r}\boldsymbol{v}_i + Noise,$$

where $A_{i,r}, B_{i,r} > 0$ represent the coefficients for learning robust and non-robust features, respectively, and the 'Noise' term encompasses elements learned from other non-diagonal features $\boldsymbol{u}_j, \boldsymbol{v}_j (j \ne i)$, as well as random noise $\boldsymbol{\xi}_p$.

Therefore, we know that the network learns the $i$-th class if and only if either $A_{i,r}$ or $B_{i,r}$ is sufficiently large. However, to robustly learn the $i$-th class, the network must primarily learn the robust feature $\boldsymbol{u}_i$, rather than the non-robust feature $\boldsymbol{v}_i$, which motivates us to analyze the feature learning process of standard training and adversarial training to understand the underlying mechanism why adversarial examples exist and how adversarial training algorithm works.

## 4 MAIN RESULTS

We first formally introduce the concept, feature learning accuracy, as the following definition.

**Definition 4.1** (Feature Learning Accuracy). For a given feature subset $\mathcal{H} \subset \mathcal{F}$ ($\mathcal{F}$ is all feature set as the same as Definition 2.1), $\mathcal{H}$-extended feature representative distribution $\mathcal{D}_\mathcal{H}$ and classifier model $\boldsymbol{F}$, we define the feature learning accuracy as $\mathbb{P}_{(\boldsymbol{X_f},y)\sim\mathcal{D}_\mathcal{H}} \left[ \operatorname{argmax}_{i\in[k]} F_i(\boldsymbol{X_f}) = y \right]$, where $\boldsymbol{X_f}(\boldsymbol{f} \in \mathcal{H})$ is the $\boldsymbol{f}$-extended representative and $y$ is the label which feature $\boldsymbol{f}$ corresponds to.

Here, we choose $\mathcal{F}_R := \{\boldsymbol{u}_i\}_{i\in[k]}, \mathcal{F}_{NR} := \{\boldsymbol{v}_i\}_{i\in[k]} \subset \mathcal{F}$ as robust/non-robust feature sets. We construct $\mathcal{D}_{\mathcal{F}_R}/\mathcal{D}_{\mathcal{F}_{NR}}$ by sampling $(\boldsymbol{X}, y) \sim \mathcal{D}$ and setting $\boldsymbol{X_f}$ to $\boldsymbol{X}$ with all instances of feature $\boldsymbol{v}_i/\boldsymbol{u}_i$ replaced by all-zero vector (a figurative illustration is presented in Figure 2).

**Remark 4.2.** *We define feature learning accuracy based on whether the model $\boldsymbol{F}$ can accurately classify data points when presented with only a single signal feature $\boldsymbol{f}$, which indeed generalizes the notion of weight-feature correlation $\langle \boldsymbol{w}_{i,r}, \boldsymbol{f} \rangle$ to general non-linear models and non-linear features.*

Now, we state the main theorems in this paper as follows.

**Theorem 4.3** (Standard Training Converges to Non-robust Global Minima). *For sufficiently large $d$, suppose we train the model using the standard training starting from the random initialization, then after $T = \Theta(\operatorname{poly}(d)/\eta)$ iterations, with high probability over the sampled training dataset $\mathcal{Z}$, the model $\boldsymbol{F}^{(T)}$ satisfies:*

- *Standard training is perfect: for all $(\boldsymbol{X}, y) \in \mathcal{Z}$, all $i \in [k]\backslash\{y\}$ : $F_y^{(T)}(\boldsymbol{X}) > F_i^{(T)}(\boldsymbol{X})$.*

- *Non-robust features are learned: $\mathbb{P}_{(\boldsymbol{X_f},y)\sim\mathcal{D}_{\mathcal{F}_{NR}}} \left[ \operatorname{argmax}_{i\in[k]} F_i^{(T)}(\boldsymbol{X_f}) \ne y \right] = o(1).$*

- *Standard test accuracy is good: $\mathbb{P}_{(\boldsymbol{X},y)\sim\mathcal{D}} \left[ \operatorname{argmax}_{i\in[k]} F_i^{(T)}(\boldsymbol{X}) \ne y \right] = o(1).$*

- *Robust test accuracy is bad: for any given data $(\boldsymbol{X}, y)$, using the following perturbation $\boldsymbol{\Delta}(\boldsymbol{X}, y) := (\boldsymbol{\delta}_1, \boldsymbol{\delta}_2, \ldots, \boldsymbol{\delta}_P)$, where $\boldsymbol{\delta}_p := -\beta_p \boldsymbol{v}_y + \epsilon \boldsymbol{v}_{y'}$ for $p \in \mathcal{J}_{NR}$; $\boldsymbol{\delta}_p := \boldsymbol{0}$ for $p \in \mathcal{J}_R$, and $y'$ is randomly chosen from $[k] \setminus \{y\}$ (which does not depend on the model $\boldsymbol{F}^{(T)}$ and is illustrated in Figure 2), we have*

$$\mathbb{P}_{(\boldsymbol{X},y)\sim\mathcal{D}} \left[ \operatorname{argmax}_{i\in[k]} F_i^{(T)}(\boldsymbol{X} + \boldsymbol{\Delta}(\boldsymbol{X}, y)) \ne y \right] = 1 - o(1).$$

Theorem 4.3 states that standard training of a neural network achieves good standard accuracy but poor robust performance. This is due to the dominance of non-robust feature learning during the training dynamics. Moreover, we notice that a perturbation based on non-robust features is sufficient to confuse the network. This implies that adversarial examples may stem from non-robust features, which could also help explain the transferability of adversarial attacks (Papernot et al., 2016).

Next, we present our main results about adversarial training as the following theorem.

**Theorem 4.4** (Adversarial Training Converges to Robust Global Minima). *For sufficiently large d, suppose we train the model using the adversarial training algorithm starting from the random initialization, then after $T = \Theta(\mathrm{poly}(d)/\eta)$ iterations, with high probability over the sampled training dataset $\mathcal{Z}$, the model $\boldsymbol{F}^{(T)}$ satisfies:*

- *Adversarial training is perfect: for all $(\boldsymbol{X}, y) \in \mathcal{Z}$ and all perturbation $\boldsymbol{\Delta}$ satisfying $\|\boldsymbol{\Delta}\|_\infty \leq \epsilon$, all $i \in [k]\backslash\{y\} : F_y^{(T)}(\boldsymbol{X} + \boldsymbol{\Delta}) > F_i^{(T)}(\boldsymbol{X} + \boldsymbol{\Delta})$.*

- *Robust features are learned: $\mathbb{P}_{(\boldsymbol{X_f}, y) \sim \mathcal{D}_{\mathcal{F}_R}} \left[ \mathrm{argmax}_{i \in [k]} F_i^{(T)}(\boldsymbol{X_f}) \neq y \right] = o(1)$.*

- *Robust test accuracy is good:*

$$\mathbb{P}_{(\boldsymbol{X}, y) \sim \mathcal{D}} \left[ \exists \boldsymbol{\Delta} \in \left( \mathbb{R}^d \right)^P \text{ s.t. } \|\boldsymbol{\Delta}\|_\infty \leq \epsilon, \mathrm{argmax}_{i \in [k]} F_i^{(T)}(\boldsymbol{X} + \boldsymbol{\Delta}) \neq y \right] = o(1).$$

Theorem 4.4 shows that the network learner provably learns robust features through adversarial training method, which thereby improves the network robustness against adversarial perturbations.

## 5 TECHNIQUE OVERVIEW: LEARNING PROCESS ANALYSIS

We present a high level proof intuition for the training dynamics. For simplicity, we consider a simplified setup with the noiseless population risk (i.e. $\sigma_n = 0, w.h.p. \ \alpha_p \gg \epsilon \gtrsim \beta_p, \mathcal{Z} = \mathcal{D}$).

By analyzing gradient descent dynamics of the model $\boldsymbol{F}$, we know that there exists time-variant coefficient sequences $\{A_{i,r}^{(t)}\}_{t=0}^\infty$ and $\{B_{i,r}^{(t)}\}_{t=0}^\infty$ such that, for any pair $(i, r) \in [k] \times [m]$, it holds that $\boldsymbol{w}_{i,r}^{(t)} \approx A_{i,r}^{(t)} \boldsymbol{u}_i + B_{i,r}^{(t)} \boldsymbol{v}_i$. Then, we focus on dynamics of these two coefficient sequences.

### 5.1 LEARNING PROCESS ANALYSIS FOR STANDARD TRAINING

For standard training, by denoting $\mathrm{logit}_i(\boldsymbol{F}, \boldsymbol{X}) := \frac{e^{F_i(\boldsymbol{X})}}{\sum_{j \in [k]} e^{F_j(\boldsymbol{X})}}$, we have the following lemma.

**Lemma 5.1** (Feature Learning Iteration for Standard Training). *During standard training, for any time $t \geq 0$ and pair $(i, r) \in [k] \times [m]$, the two sequences $\{A_{i,r}^{(t)}\}$ and $\{B_{i,r}^{(t)}\}$ satisfy:*

$$\begin{cases} A_{i,r}^{(t+1)} = A_{i,r}^{(t)} + \frac{\eta}{k} \mathbb{E}_{\mathcal{D}_{\mathcal{J}}, i, \mathcal{D}_{\alpha, i}} \left[ \left( 1 - \mathrm{logit}_i(\boldsymbol{F}^{(t)}, \boldsymbol{X}) \right) \sum_{p \in \mathcal{J}_R} \widetilde{\mathrm{ReLU}}' \left( \alpha_p A_{i,r}^{(t)} \right) \alpha_p \right], \\ B_{i,r}^{(t+1)} = B_{i,r}^{(t)} + \frac{\eta}{k} \mathbb{E}_{\mathcal{D}_{\mathcal{J}}, i, \mathcal{D}_{\beta, i}} \left[ \left( 1 - \mathrm{logit}_i(\boldsymbol{F}^{(t)}, \boldsymbol{X}) \right) \sum_{p \in \mathcal{J}_{NR}} \widetilde{\mathrm{ReLU}}' \left( \beta_p B_{i,r}^{(t)} \right) \beta_p \right]. \end{cases}$$

**Approximation Near Initialization.** At the start of training, due to our random initialization, i.e., $\boldsymbol{w}_{i,r} \sim \mathcal{N}(\boldsymbol{0}, \boldsymbol{I}_d / \mathrm{poly}(d))$, we have a constant loss derivative, namely $1 - \mathrm{logit}_i(\boldsymbol{F}^{(t)}, \boldsymbol{X}) = \Theta(1)$. And we know that, w.h.p., the activation function predominantly lies within the polynomial part.

**Non-Robust Feature Learning Dominates.** Under Assumption 2.3, it holds that $\mathbb{E}\left[ \sum_{p \in \mathcal{J}_{NR}} \beta_p^q \right] \gg \mathbb{E}\left[ \sum_{p \in \mathcal{J}_R} \alpha_p^q \right]$, which implies that the non-robust feature learning $\max_{r \in [m]} B_{i,r}^{(t)}$ increases more rapidly than the robust feature learning $\max_{r \in [m]} A_{i,r}^{(t)}$. Moreover, by applying Tensor Power Method Lemma (Allen-Zhu & Li, 2023a), we know that $\max_{r \in [m]} B_{i,r}^{(t)}$ attains an order of $\tilde{\Theta}(1)$, while $\max_{r \in [m]} A_{i,r}^{(t)}$ still maintains $\tilde{o}(1)$-order. Afterward, the loss derivative approaches zero (i.e. $1 - \mathrm{logit}_i(\boldsymbol{F}^{(t)}, \boldsymbol{X}) = o(1)$), and the network ultimately converges within the linear region of the $\widetilde{\mathrm{ReLU}}$.

Figure 3: **Simulations on synthetic data.** *The two left figures:* dynamics of normalized weight-feature correlations for std/adv training. *The two right figures:* learning curves for std/adv training.

## 5.2 LEARNING PROCESS ANALYSIS FOR ADVERSARIAL TRAINING

For adversarial training, we divide the learning process into two phases via the following lemma.

**Lemma 5.2** (Feature Learning Iteration for Adversarial Training at Polynomial Part). *During adversarial training, there exists some time threshold $T_0 > 0$ such that, for any early time $0 \leq t \leq T_0$ and pair $(i, r) \in [k] \times [m]$, the two sequences $\{A_{i,r}^{(t)}\}$ and $\{B_{i,r}^{(t)}\}$ satisfy:*

$$
\begin{cases}
A_{i,r}^{(t+1)} \approx A_{i,r}^{(t)} + \Theta(\eta) \left( A_{i,r}^{(t)} \right)^{q-1} \mathbb{E}\left[ \sum_{p \in \mathcal{J}_R} \alpha_p^q \left( 1 - \min\left\{ \frac{\epsilon}{\alpha_p}, \tilde{\Theta}(\tilde{\eta}) \sum_{s \in [m]} \left( A_{i,s}^{(t)} \right)^q \right\} \right)^q \right], \\
B_{i,r}^{(t+1)} \approx B_{i,r}^{(t)} + \Theta(\eta) \left( B_{i,r}^{(t)} \right)^{q-1} \mathbb{E}\left[ \sum_{p \in \mathcal{J}_{NR}} \beta_p^q \left( 1 - \min\left\{ \frac{\epsilon}{\beta_p}, \tilde{\Theta}(\tilde{\eta}) \sum_{s \in [m]} \left( B_{i,s}^{(t)} \right)^q \right\} \right)^q \right].
\end{cases}
$$

**Phase I: First, Network Partially Learns Non-Robust Features.** At the beginning, due to our small initialization, we know all feature learning coefficients $A_{i,r}^{(t)}, B_{i,r}^{(t)} = o(1)$, which suggests that the total feature learning $\sum_{s \in [m]} \left( A_{i,s}^{(t)} \right)^q$ and $\sum_{s \in [m]} \left( B_{i,s}^{(t)} \right)^q$ are sufficiently small. Then, the feature learning process is similar to standard training until the non-robust feature learning becomes large.

**Phase II: Next, Robust Feature Learning Starts Increasing.** Once the total non-robust feature learning $\sum_{s \in [m]} \left( B_{i,s}^{(t)} \right)^q$ attains an order of $\tilde{\Theta}(\tilde{\eta}^{-1})$, it is known that the non-robust feature learning will stop, due to $\frac{\epsilon}{\beta_p} \gtrsim 1$ and $1 - \tilde{\Theta}(\tilde{\eta}) \sum_{s \in [m]} \left( B_{i,s}^{(t)} \right)^q \approx 0$. In contrast, the robust feature learning continues to increase since it always holds that $1 - \min\left\{ \frac{\epsilon}{\alpha_p}, \tilde{\Theta}(\tilde{\eta}) \sum_{s \in [m]} \left( A_{i,s}^{(t)} \right)^q \right\} \geq 1 - \frac{\epsilon}{\alpha_p} \geq \Omega(1)$. Thus, the robust feature learning will increase over the non-robust feature learning finally, and the network converges to robust regime, i.e. $\max_{r \in [m]} A_{i,r}^{(T)} \gg \max_{r \in [m]} B_{i,r}^{(T)}$ for large $T$.

## 6 EXPERIMENTS

### 6.1 SIMULATIONS ON SYNTHETIC DATA

**Experiment Settings.** We first perform numerical experiments on synthetic data to verify our theoretical results. Here, our synthetic data is generated according to Definition 2.2. We choose the hyperparameters as: $k = 2, d = 100, P = 16, q = \tau = 3, \varrho = 1, \epsilon = 1.2, N = 100, m = 100, \sigma_0 = 0.01, \sigma_n = 0.1, \eta = 0.1, \tilde{\eta} = 10^3, T = 1000$, and $|\mathcal{J}_R| \equiv 1, |\mathcal{J}_{NR}| \equiv 15, \alpha_p \equiv 2, \beta_p \equiv 1$ for each $(\boldsymbol{X}, y) \sim \mathcal{D}$. Then, we run the standard training and adversarial training algorithms, and we characterize the feature learning process via the dynamics of normalized weight-feature correlations: $\max_{r \in [m]} \langle \boldsymbol{w}_{i,r}, \boldsymbol{u}_i \rangle / \|\boldsymbol{u}_i\|_2, i = 1, 2$ (robust feature learning), and $\max_{r \in [m]} \langle \boldsymbol{w}_{i,r}, \boldsymbol{v}_i \rangle / \|\boldsymbol{v}_i\|_2, i = 1, 2$ (non-robust feature learning). We calculate the robust test accuracy by the standard PGD attack.

**Experiment Results.** The numerical results are reported in Figure 3. We observe that, in standard training, non-robust feature learning dominates during training process. There exists a phase transition during adversarial training (it happens nearly at 150-epoch). Phase I: the network learner mainly learns non-robust features to achieve perfect standard test accuracy, but robust test accuracy maintains zero. Phase II: the increments of non-robust feature learning is restrained while robust feature learning and robust test accuracy start to increase. These results empirically verify our analysis in Section 5.

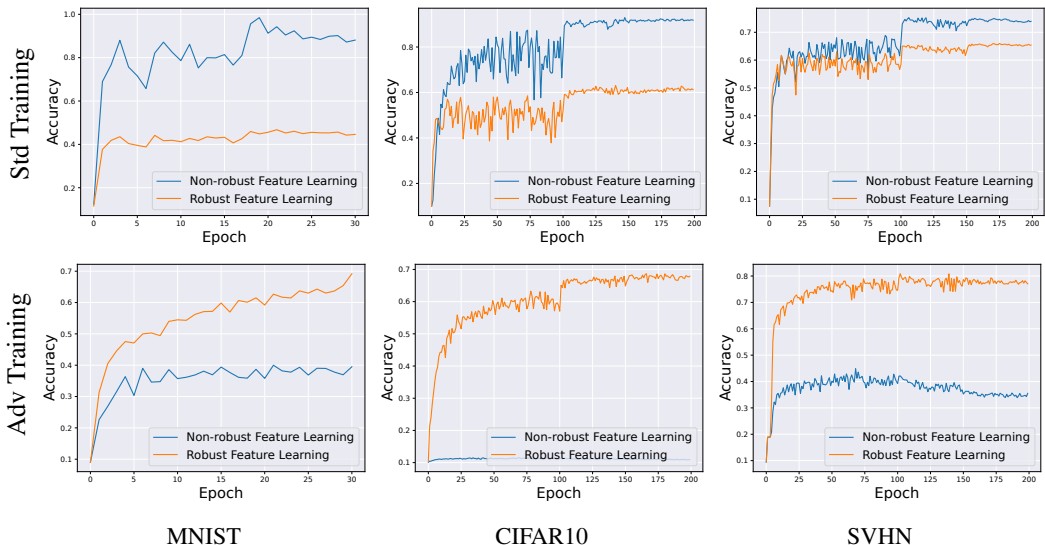

Figure 4: **Feature learning process on real-image datasets.** *Top row:* feature learning accuracy during standard training. *Bottom row:* feature learning accuracy during adversarial training.

## 6.2 EXPERIMENTS ON REAL-WORLD DATASETS

**Experiment Settings.** Instead of weight-feature correlation used in synthetic data setting, here on MNIST, CIFAR10 and SVHN datasets, we apply **feature learning accuracy** in Definition 4.1 to measure non-robust/robust feature learning during training dynamics. Similar to the method proposed in Ilyas et al. (2019), we reconstruct datasets $\hat{\mathcal{D}}_{NR}, \hat{\mathcal{D}}_R$ as the feature representative distributions by a one-to-one mapping $\boldsymbol{X} \mapsto \hat{\boldsymbol{X}}$. Specifically, we solve the following optimization problem to derive $\hat{\boldsymbol{X}}$: $\min_{\hat{\boldsymbol{X}}} \|\boldsymbol{G}(\hat{\boldsymbol{X}}) - \boldsymbol{G}(\boldsymbol{X})\|_2$, where $\boldsymbol{X} \in \mathcal{D}$ is the target data point, and $\boldsymbol{G}$ is the mapping from input $\boldsymbol{X}$ to the representation layer for network learners. When $\boldsymbol{G}$ is chosen from a standard/adversarial-trained network, we derive the non-robust/robust representative dataset $\hat{\mathcal{D}}_{NR}/\hat{\mathcal{D}}_R$. Then, we run the standard training and adversarial training algorithms and record the dynamics of feature learning accuracy w.r.t. $\hat{\mathcal{D}}_{NR}$ and $\hat{\mathcal{D}}_R$. For MNIST, we choose ResNet18 and $\ell_\infty$-perturbation with radius 0.3, and we run algorithms for 30 iterations. For CIFAR10 and SVHN, we choose WideResNet-34-10 and $\ell_\infty$-perturbation with radius $8/255$ and run algorithms for 200 iterations.

**Experiment Results.** The results are presented in Figure 4. For all three datasets, we could see that, in standard training, network learners predominantly learn non-robust features rather than robust features, while adversarial training can both inhabit non-robust feature learning and strengthen robust feature learning, which empirically demonstrates our theoretical results in Section 4.

## 7 CONCLUSION, LIMITATIONS AND FUTURE WORKS

In this paper, we provide a theoretical explanation why adversarial examples widely exist and how the adversarial training method improves the model robustness. Based on robust/non-robust feature decomposition, we prove an implicit bias of standard training that network mainly learns non-robust features, leading to adversarial examples. Furthermore, we demonstrate that adversarial training can provably enhance the robust feature learning and suppress the non-robust feature learning. We believe our theory gives some insights into the inner workings of adversarial robust learning in deep learning. However, we also believe that our results can be significantly improved if we build on a more realistic setup. For example, an important future direction is to extend our theoretical analysis to deep neural networks. Another interesting direction is to extend our theoretical analysis method to adversarial training based on multi-step gradient ascent algorithms, such as PGD.

ACKNOWLEDGMENTS

Binghui Li is supported by the Elite Ph.D. Program in Applied Mathematics for PhD Candidates in Peking University. We thank Yuhui Li, Zhongao Wang, Zixin Wen, Zean Xu and Qinhan Yu for helpful discussions and anonymous reviewers for their valuable suggestions.

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

CONTENTS

## A    ADDITIONAL EXPERIMENT RESULTS ABOUT REAL-IMAGE DATASETS

### A.1    FEATURE LEARNING PROCESS ON REAL-WORLD DATASETS

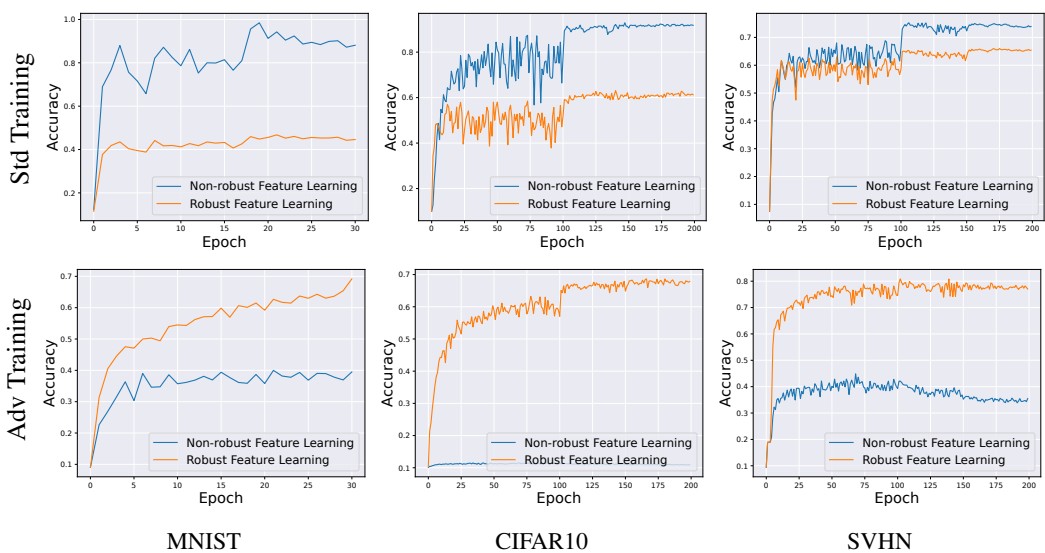

Figure 5: **Feature learning process on real-image datasets.** *Top row:* feature learning accuracy during standard training. *Bottom row:* feature learning accuracy during adversarial training.

Table 1: Feature learning results on real-world datasets

| Dataset | Algorithm | Test Accuracy | | Feature Learning | |
|---|---|---|---|---|---|
| | | **Standard** | **Robust** | **Non-Robust** | **Robust** |
| **MNIST** | Std Train | 99.56 | 0.04 | 88.11 | 44.61 |
| | Adv Train | 99.41 | 94.43 | 39.49 | 69.19 |
| **CIFAR10** | Std Train | 95.74 | 0.00 | 91.79 | 61.26 |
| | Adv Train | 86.28 | 45.13 | 10.88 | 67.88 |
| **SVHN** | Std Train | 96.84 | 0.16 | 73.91 | 65.38 |
| | Adv Train | 91.99 | 58.55 | 35.50 | 77.04 |

**Experiment Settings.** Instead of weight-feature correlation used in synthetic data setting, here on MNIST, CIFAR10 and SVHN datasets, we apply **feature learning accuracy** in Definition 4.1 to measure non-robust/robust feature learning during training dynamics.

**Definition A.1** (Feature Learning Accuracy). For a given feature subset $\mathcal{H} \subset \mathcal{F}$ ($\mathcal{F}$ is all feature set as the same as Definition 2.1), $\mathcal{H}$-extended feature representative distribution $\mathcal{D}_{\mathcal{H}}$ and classifier model $\boldsymbol{F}$, we define the feature learning accuracy as $\mathbb{P}_{(\boldsymbol{X}_{\boldsymbol{f}},y)\sim\mathcal{D}_{\mathcal{H}}}\left[\arg\max_{i\in[k]} F_i(\boldsymbol{X}_{\boldsymbol{f}}) = y\right]$, where $\boldsymbol{X}_{\boldsymbol{f}}(\boldsymbol{f}\in\mathcal{H})$ is the $\boldsymbol{f}$-extended representative and $y$ is the label which feature $\boldsymbol{f}$ corresponds to.

**Remark A.2.** *We define feature learning accuracy based on whether the model $\boldsymbol{F}$ can accurately classify data points when presented with only a single signal feature $\boldsymbol{f}$, which indeed generalizes the notion of weight-feature correlation $\langle \boldsymbol{w}_{i,r}, \boldsymbol{f} \rangle$ to general non-linear models and non-linear features.*

Similar to the method proposed in Ilyas et al. (2019), we reconstruct datasets $\hat{\mathcal{D}}_{NR}, \hat{\mathcal{D}}_{R}$ as the feature representative distributions by a one-to-one mapping $\boldsymbol{X} \mapsto \hat{\boldsymbol{X}}$. Specifically, we solve the following optimization problem to derive $\hat{\boldsymbol{X}}$:

$$\min_{\hat{\boldsymbol{X}}} \|\boldsymbol{G}(\hat{\boldsymbol{X}}) - \boldsymbol{G}(\boldsymbol{X})\|_2,$$

where $X \in \mathcal{D}$ is the target data point, and $G$ is the mapping from input $X$ to the representation layer for network learners. When $G$ is chosen from a standard/adversarial-trained network, we derive the non-robust/robust representative dataset $\hat{\mathcal{D}}_{NR}/\hat{\mathcal{D}}_R$. Then, we run the standard training and adversarial training algorithms and record the dynamics of feature learning accuracy w.r.t. $\hat{\mathcal{D}}_{NR}$ and $\hat{\mathcal{D}}_R$. For MNIST, we choose ResNet18 (He et al., 2016) and $\ell_\infty$-perturbation with radius 0.3, and we run algorithms for 30 iterations. For CIFAR10 and SVHN, we choose WideResNet-34-10 (Zagoruyko & Komodakis, 2016) as network architecture and $\ell_\infty$-perturbation with radius $8/255$ and run algorithms for 200 iterations by using a single NVIDIA RTX 4090 GPU.

**Experiment Results.** The results are presented in Figure 5 and Table 1. For all three datasets, we could see that, in standard training, network learners predominantly learn non-robust features rather than robust features, while adversarial training can both inhabit non-robust feature learning and strengthen robust feature learning, which empirically demonstrates our theoretical results (Theorem 4.3 and Theorem 4.4) in Section 4.

### A.2    TARGETED ADVERSARIAL ATTACK ON REAL-WORLD DATASETS

Table 2: Targeted attack success rates on CIFAR10

| Model | Attack | Source → Target | | | |
|---|---|---|---|---|---|
| | | Cat → Dog | Dog → Cat | Car → Plane | Plane → Car |
| **Std Trained** | NRF-PGD | $71.41 \pm 1.17$ | $80.36 \pm 0.28$ | $54.08 \pm 0.99$ | $76.74 \pm 0.77$ |
| | RF-PGD | $11.30 \pm 0.55$ | $9.58 \pm 0.58$ | $1.24 \pm 0.10$ | $2.63 \pm 0.13$ |
| **Adv Trained** | NRF-PGD | $9.60 \pm 0.18$ | $15.16 \pm 0.23$ | $0.34 \pm 0.04$ | $0.40 \pm 0.00$ |
| | RF-PGD | $19.38 \pm 0.29$ | $26.00 \pm 0.67$ | $2.64 \pm 0.18$ | $1.96 \pm 0.13$ |

**Experiment Settings.** To verify whether adversarial examples primarily stem from non-robust features or robust features, we propose two corresponding model-free attack algorithms: **non-robust-feature based PGD (NRF-PGD)** and **robust-feature based PGD (RF-PGD)**. Concretely, we use $G_{std}(\cdot)$ and $G_{adv}(\cdot)$ to denote the mapping from input $X$ to the representation layer for standard-trained network and adversarially-trained network, respectively. Similar to the previous section and Ilyas et al. (2019), we can regard $G_{std}(\cdot)$ and $G_{adv}(\cdot)$ as non-robust-feature extractor and robust-feature extractor. Then, we define NRF-PGD by using PGD method to solve the following optmization problem over the perturbation $\mathbf{\Delta}$:

$$\min_{\|\mathbf{\Delta}\|_\infty \leq \epsilon} \|G_{std}(X + \mathbf{\Delta}) - G_{std}(X')\|_2,$$

where $X$ is the original image from the source class, $X'$ is a random image sampled from the target class, and $\epsilon$ is the perturbation radius. Similarly, we can define RF-PGD by using PGD method to solve the following optmization problem over the perturbation $\mathbf{\Delta}$:

$$\min_{\|\mathbf{\Delta}\|_\infty \leq \epsilon} \|G_{adv}(X + \mathbf{\Delta}) - G_{adv}(X')\|_2.$$

Then, we evaluate the performance of the two attack methods on the standard-trained network $F_{std}(\cdot)$ and the adversarially-trained network $F_{adv}(\cdot)$ (they are different from the reference networks from which we choose our $G_{std}(\cdot)$ and $G_{adv}(\cdot)$). For CIRAF10 dataset, we apply WideResNet-34-10 (Zagoruyko & Komodakis, 2016) as our learner networks, and choose the typical perturbation radius $\epsilon = 8/255$ for $\ell_\infty$-attack. And all experiments are repeated over 5 random seeds.

**Experiment Results.** We select two pairs ((cat, dog) and (car, plane)) to show the targeted attack success rate, which is presented in Table 2. It is evident that, for the standard-trained classifier, the success rates of PGD attacks using non-robust features are significantly high. Specifically, targeted attacks achieve the following success rates with NRF-PGD: Cat → Dog at 71.41%, Dog → Cat at 80.36%, Car → Plane at 54.08%, and Plane → Car at 76.74%. Conversely, when utilizing robust features for PGD, the success rates are considerably lower. For example, RF-PGD attacks yield the following success rates: Cat → Dog at 9.60%, Dog → Cat at 15.16%, Car → Plane at 0.34%, and Plane → Car at 0.40%. In the case of the adversarially-trained network, the situation alters, but in practice, after adversarial training, the success rates for both types of attacks remain low. These findings indicate that adversarial examples predominantly originate from non-robust features.

## B    PRELIMINARY FOR PROOF TECHNIQUE

### B.1    NOTATIONS

Throughout this work, we use letters for scalars and bold letters for vectors. For any given two sequences $\{A_n\}_{n=0}^{\infty}$ and $\{B_n\}_{n=0}^{\infty}$, we denote $A_n = O(B_n)$ if there exist some absolute constant $C_1 > 0$ and $N_1 > 0$ such that $|A_n| \leq C_1 |B_n|$ for all $n \geq N_1$. Similarly, we denote $A_n = \Omega(B_n)$ if there exist $C_2 > 0$ and $N_2 > 0$ such that $|A_n| \geq C_2 |B_n|$ for all $n > N_2$. We say $A_n = \Theta(B_n)$ if $A_n = O(B_n)$ and $A_n = \Omega(B_n)$ both holds. We use $\widetilde{O}(\cdot), \widetilde{\Omega}(\cdot)$, and $\widetilde{\Theta}(\cdot)$ to hide logarithmic factors in these notations respectively. Moreover, we denote $A_n = \text{poly}(B_n)$ if $A_n = O(B_n^K)$ for some positive constant $K$, and $A_n = \text{polylog}(B_n)$ if $B_n = \text{poly}(\log(B_n))$. We say $A_n = o(B_n)$ (or $A_n \ll B_n$ or $B_n \gg A_n$) if for arbitrary positive constant $C_3 > 0$, there exists $N_3 > 0$ such that $|A_n| < C_3|B_n|$ for all $n > N_3$. And we also use $A_n \approx B_n$ to denote $A_n = B_n + o(1)$.

### B.2    PRELIMINARY LEMMAS

**Lemma B.1.** *Let $X = (X_1, \ldots, X_n)$, where $X_i \sim \mathcal{N}(0, 1)$ are i.i.d., then we have*

- $\mathbb{P}\left[\|X\|_\infty \geq \sqrt{2\log(2n)} + t\right] \leq \frac{1}{2}\exp\left(-t^2/2\right) (t > 0)$, *and*

- $\mathbb{P}\left[\|X\|_\infty \leq \sqrt{2\log(2n) - \delta}\right] \leq \exp\left\{-\frac{e^{\delta/2}}{\sqrt{2\pi}(\sqrt{2\log(2n)}+1)}\right\}$, *where we can take $\delta = K\log\log(n)$ for large $K$ such that this probability is small.*

**Lemma B.2** (Tensor Power Method, from Allen-Zhu & Li (2023a)). *Let $q \geq 3$ be a constant and $x_0, y_0 = o(1)$. Let $\{x_t, y_t\}_{t\geq 0}$ be two positive sequences updated as*

- $x_{t+1} \geq x_t + \eta C_t x_t^{q-1}$ *for some $C_t = \Theta(1)$, and*

- $y_{t+1} \leq y_t + \eta S C_t y_t^{q-1}$ *for some constant $S = \Theta(1)$,*

*where $\eta = O(1/\text{poly}(d))$ for a sufficiently large polynomial in $d$. Suppose $x_0 \geq y_0 S^{\frac{1}{q-2}}\left(1 + \Theta\left(\frac{1}{\text{polylog}(d)}\right)\right)$. For every $A = O(1)$, letting $T_x$ be the first iteration such that $x_t \geq A$, we must have that*

$$y_{T_x} = O(y_0 \text{polylog}(d))$$

### B.3    MORE DETAILED DATA ASSUMPTION

**Assumption B.3** (Choice of Hyperparameters). **We assume that:**

$$d = \text{poly}(k) \gg 2k, \quad P = \Theta(1), \quad \alpha := \inf \alpha_p, \beta := \inf \beta_p,$$

$$\sigma_0 = \sigma_n = \frac{1}{\sqrt{d}}, \quad \alpha, \beta, \epsilon = \Theta(1), \quad \alpha \gg \epsilon > \beta,$$

$$m = \text{polylog}(d), \quad N = \text{poly}(d), \quad q = \tau = 3, \quad \varrho = \frac{1}{\text{polylog}(d)},$$

$$\eta = \frac{1}{\text{poly}(d)}, \quad \tilde{\eta} = d^{c_0},$$

where $c_0 \in (0, 1)$ are any positive constant.

**Discussion of Hyperparameter Choices.** While the choices of these hyperparameters are not unique, we make specific selections above for the sake of calculations in our proofs. However, it is the relationships between them that are of primary importance. We select the dimension of the patch $d$ to be sufficiently large (i.e., we assume $d = \text{poly}(k)$ for a suitably large polynomial) to ensure that all $2k$ features can be orthogonal within the space $\mathbb{R}^d$. The number of patches $P$ is held constant. The conditions $\alpha \gg \epsilon > \beta$ ensure the first part of Assumption 2.3 and accommodate the small perturbation radius condition previously mentioned. The width of the network learner is chosen

as $m = \text{polylog}(d)$ to achieve mild over-parameterization for efficient optimization. Furthermore, the adversarial learning rate $\tilde{\eta}$ is significantly larger than the weight learning rate $\eta$, aligning with practical implementations (Carlini & Wagner, 2017; Gowal et al., 2019; Sriramanan et al., 2020).

## C    DETAILED PROOFS FOR SECTION 3

In this section, we provide a detailed proof for Section 3 (including Proposition 3.1 and Proposition 3.2). And we also give a more detailed discussion about it.

### C.1    PROOF OF PROPOSITION 3.1

**Theorem C.1** (Restatement of Proposition 3.1). *We consider the special case when $m = 1$ and $\boldsymbol{w}_{i,1} = \gamma \boldsymbol{v}_i$, where $\gamma > 0$ is a scale coefficient. Then, it holds that the standard empirical risk satisfies $\lim_{\gamma \to \infty} \mathcal{L}_{CE}(\boldsymbol{F}) = o(1)$, but the adversarial test error satisfies $\lim_{\gamma \to \infty} \mathbb{P}_{(\boldsymbol{X}, y) \sim \mathcal{D}} \left[ \exists \boldsymbol{\Delta} \in \left( \mathbb{R}^d \right)^P \text{ s.t. } \|\boldsymbol{\Delta}\|_\infty \leq \epsilon, \operatorname{argmax}_{i \in [k]} F_i(\boldsymbol{X} + \boldsymbol{\Delta}) \neq y \right] = 1 - o(1).$*

**Proof Sketch.** For a given data point $(\boldsymbol{X}, y) \in \mathcal{Z}$ and sufficiently large $\gamma$, we calculate the margin and derive w.h.p. $F_y(\boldsymbol{X}) \gg F_j(\boldsymbol{X}), \forall j \in [k] \setminus \{y\}$, which implies $\mathcal{L}_{CE}(\boldsymbol{F}) \to o(1)$. However, if we choose the perturbation $\boldsymbol{\Delta}(\boldsymbol{X}, y) := (\boldsymbol{\delta}_1, \boldsymbol{\delta}_2, \dots, \boldsymbol{\delta}_P)$, where $\boldsymbol{\delta}_p := -\beta_p \boldsymbol{v}_y + \epsilon \boldsymbol{v}_{y'}$ for $p \in \mathcal{J}_{NR}$; $\boldsymbol{\delta}_p := \boldsymbol{0}$ for $p \in \mathcal{J}_R$, and $y'$ is randomly chosen from $[k] \setminus \{y\}$, we know w.h.p. $F_{y'}(\boldsymbol{X} + \boldsymbol{\Delta}) \gg F_j(\boldsymbol{X} + \boldsymbol{\Delta}), \forall j \in [k] \setminus \{y'\}$, which suggests the adversarial test error is $1 - o(1)$.

Now, we give the detailed proof as follows.

*Proof.* For a given data point $(\boldsymbol{X}, y) \in \mathcal{Z}$ and sufficiently large $\gamma$, we calculate the margin as follows. With probability $1 - o(1)$, it holds that

$$
\begin{aligned}
F_y(\boldsymbol{X}) &= \sum_{p \in \mathcal{J}_R} \widetilde{\operatorname{ReLU}}(\langle \gamma \boldsymbol{v}_y, \alpha_p \boldsymbol{u}_y + \boldsymbol{\xi}_p \rangle) + \sum_{p \in \mathcal{J}_{NR}} \widetilde{\operatorname{ReLU}}(\langle \gamma \boldsymbol{v}_y, \beta_p \boldsymbol{v}_y + \boldsymbol{\xi}_p \rangle) \\
&= \sum_{p \in \mathcal{J}_R} \widetilde{\operatorname{ReLU}}(\gamma \langle \boldsymbol{v}_y, \boldsymbol{\xi}_p \rangle) + \sum_{p \in \mathcal{J}_{NR}} \widetilde{\operatorname{ReLU}}(\gamma (\beta_p + \langle \boldsymbol{v}_y, \boldsymbol{\xi}_p \rangle)) \\
&\geq \sum_{p \in \mathcal{J}_{NR}} \widetilde{\operatorname{ReLU}}(\gamma(\beta_p - \Theta(\sigma_n)) \geq \gamma \Theta \left( \sum_{p \in \mathcal{J}_{NR}} \beta_p \right).
\end{aligned}
$$

And for any $j \in [P] \setminus \{y\}$, we have

$$
\begin{aligned}
F_j(\boldsymbol{X}) &= \sum_{p \in \mathcal{J}_R} \widetilde{\operatorname{ReLU}}(\langle \gamma \boldsymbol{v}_j, \alpha_p \boldsymbol{u}_y + \boldsymbol{\xi}_p \rangle) + \sum_{p \in \mathcal{J}_{NR}} \widetilde{\operatorname{ReLU}}(\langle \gamma \boldsymbol{v}_j, \beta_p \boldsymbol{v}_y + \boldsymbol{\xi}_p \rangle) \\
&\leq P \widetilde{\operatorname{ReLU}}(\gamma \Theta(\sigma_n)) \leq \gamma \Theta(\sigma_n).
\end{aligned}
$$

Since $\sum_{p \in \mathcal{J}_{NR}} \beta_p \gg \sigma_n$, we know $\lim_{\gamma \to \infty} \mathcal{L}_{CE}(\boldsymbol{F}) = o(1)$.

Let $\boldsymbol{\Delta}(\boldsymbol{X}, y) := (\boldsymbol{\delta}_1, \boldsymbol{\delta}_2, \dots, \boldsymbol{\delta}_P)$, where $\boldsymbol{\delta}_p := -\beta_p \boldsymbol{v}_y + \epsilon \boldsymbol{v}_{y'}$ for $p \in \mathcal{J}_{NR}$; $\boldsymbol{\delta}_p := \boldsymbol{0}$ for $p \in \mathcal{J}_R$, and $y'$ is randomly chosen from $[k] \setminus \{y\}$, then we derive that, with probability $1 - o(1)$, it satisfies that

$$
\begin{aligned}
F_{y'}(\boldsymbol{X} + \boldsymbol{\Delta}(\boldsymbol{X}, y)) &= \sum_{p \in \mathcal{J}_R} \widetilde{\operatorname{ReLU}}(\langle \gamma \boldsymbol{v}_{y'}, \alpha_p \boldsymbol{u}_y + \boldsymbol{\xi}_p \rangle) + \sum_{p \in \mathcal{J}_{NR}} \widetilde{\operatorname{ReLU}}(\langle \gamma \boldsymbol{v}_{y'}, \epsilon \boldsymbol{v}_{y'} + \boldsymbol{\xi}_p \rangle) \\
&\geq \sum_{p \in \mathcal{J}_{NR}} \widetilde{\operatorname{ReLU}}(\gamma(\epsilon - \Theta(\sigma_n))) \geq \gamma \Theta(\epsilon).
\end{aligned}
$$

However, for other class $j \in [k] \setminus \{y'\}$, we know

$$
\begin{aligned}
F_j(\boldsymbol{X} + \boldsymbol{\Delta}(\boldsymbol{X}, y)) &= \sum_{p \in \mathcal{J}_R} \widetilde{\operatorname{ReLU}}(\langle \gamma \boldsymbol{v}_j, \alpha_p \boldsymbol{u}_y + \boldsymbol{\xi}_p \rangle) + \sum_{p \in \mathcal{J}_{NR}} \widetilde{\operatorname{ReLU}}(\langle \gamma \boldsymbol{v}_j, \epsilon \boldsymbol{v}_{y'} + \boldsymbol{\xi}_p \rangle) \\
&\leq \sum_{p \in [P]} \widetilde{\operatorname{ReLU}}(\gamma \langle \boldsymbol{v}_j, \boldsymbol{\xi}_p \rangle) \leq \gamma \Theta(\sigma_n).
\end{aligned}
\tag{1}
$$

Due to $\epsilon \gg \sigma_n$, we know $F_{y'}(\boldsymbol{X} + \boldsymbol{\Delta}(\boldsymbol{X}, y)) \gg F_j(\boldsymbol{X} + \boldsymbol{\Delta}(\boldsymbol{X}, y)), \forall j \in [k] \setminus \{y'\}$.

Thus, we have

$$
\lim_{\gamma \to \infty} \mathbb{P}_{(\boldsymbol{X}, y) \sim \mathcal{D}} \left[ \exists \boldsymbol{\Delta} \in \left( \mathbb{R}^d \right)^P \text{ s.t. } \|\boldsymbol{\Delta}\|_\infty \leq \epsilon, \operatorname{argmax}_{i \in [k]} F_i(\boldsymbol{X} + \boldsymbol{\Delta}) \neq y \right] = 1 - o(1).
$$

$\square$

## C.2 PROOF OF PROPOSITION 3.2

**Theorem C.2** (Restatement of Proposition 3.2). *We consider the special case when* $m = 1$ *and* $\boldsymbol{w}_{i,1} = \gamma \boldsymbol{u}_i$, *where* $\gamma > 0$ *is a scale coefficient. Then, it holds that the standard empirical risk satisfies* $\lim_{\gamma \to \infty} \mathcal{L}_{CE}(\boldsymbol{F}) = o(1)$, *and the adversarial test error satisfies* $\lim_{\gamma \to \infty} \mathbb{P}_{(\boldsymbol{X}, y) \sim \mathcal{D}} \left[ \exists \boldsymbol{\Delta} \in \left( \mathbb{R}^d \right)^P \text{ s.t. } \|\boldsymbol{\Delta}\|_\infty \le \epsilon, \operatorname{argmax}_{i \in [k]} F_i(\boldsymbol{X} + \boldsymbol{\Delta}) \ne y \right] = o(1)$.

**Proof Sketch.** For a given data point $(\boldsymbol{X}, y) \sim \mathcal{D}$, sufficiently large $\gamma$ and any perturbation $\boldsymbol{\Delta} \in \left( \mathbb{R}^d \right)^P$ satisfying $\|\boldsymbol{\Delta}\|_\infty \le \epsilon$, we show that w.h.p. $F_y(\boldsymbol{X} + \boldsymbol{\Delta}) \gtrsim \gamma \sum_{p \in \mathcal{J}_R} \alpha_p \gg \gamma(\Theta(\sigma_n) + \epsilon) \gtrsim F_j(\boldsymbol{X} + \boldsymbol{\Delta}), \forall j \in [k] \setminus \{y\}$, which implies that the adversarial test error is at most $o(1)$.

Now, we give the detailed proof as follows.

*Proof.* For a given data point $(\boldsymbol{X}, y) \sim \mathcal{D}$, sufficiently large $\gamma$ and any perturbation $\boldsymbol{\Delta} = (\boldsymbol{\delta}_1, \boldsymbol{\delta}_2, \ldots, \boldsymbol{\delta}_p) \in \left( \mathbb{R}^d \right)^P$ satisfying $\|\boldsymbol{\Delta}\|_\infty \le \epsilon$, we calculate the perturbed margin as follows. With probability $1 - o(1)$, it holds that

$$
\begin{aligned}
F_y(\boldsymbol{X} + \boldsymbol{\Delta}) &= \sum_{p \in \mathcal{J}_R} \widetilde{\operatorname{ReLU}}(\langle \gamma \boldsymbol{u}_y, \alpha_p \boldsymbol{u}_y + \boldsymbol{\xi}_p + \boldsymbol{\delta}_p \rangle) + \sum_{p \in \mathcal{J}_{NR}} \widetilde{\operatorname{ReLU}}(\langle \gamma \boldsymbol{u}_y, \beta_p \boldsymbol{v}_y + \boldsymbol{\xi}_p + \boldsymbol{\delta}_p \rangle) \\
&= \sum_{p \in \mathcal{J}_R} \widetilde{\operatorname{ReLU}}(\gamma(\alpha_p + \langle \boldsymbol{u}_y, \boldsymbol{\xi}_p \rangle + \langle \boldsymbol{u}_y, \boldsymbol{\delta}_p \rangle) + \sum_{p \in \mathcal{J}_{NR}} \widetilde{\operatorname{ReLU}}(\gamma(\langle \boldsymbol{u}_y, \boldsymbol{\xi}_p \rangle + \langle \boldsymbol{u}_y, \boldsymbol{\delta}_p \rangle)) \\
&\ge \sum_{p \in \mathcal{J}_R} \widetilde{\operatorname{ReLU}}(\gamma(\alpha_p - \Theta(\sigma_n) - \epsilon)) \ge \gamma \Theta \left( \sum_{p \in \mathcal{J}_R} \alpha_p \right).
\end{aligned}
$$

And for any $j \in [P] \setminus \{y\}$, we have

$$
\begin{aligned}
F_j(\boldsymbol{X} + \boldsymbol{\Delta}) &= \sum_{p \in \mathcal{J}_R} \widetilde{\operatorname{ReLU}}(\langle \gamma \boldsymbol{u}_j, \alpha_p \boldsymbol{u}_y + \boldsymbol{\xi}_p + \boldsymbol{\delta}_p \rangle) + \sum_{p \in \mathcal{J}_{NR}} \widetilde{\operatorname{ReLU}}(\langle \gamma \boldsymbol{u}_j, \beta_p \boldsymbol{v}_y + \boldsymbol{\xi}_p + \boldsymbol{\delta}_p \rangle) \\
&\le P \widetilde{\operatorname{ReLU}}(\gamma(\Theta(\sigma_n) + \epsilon)) \lesssim \gamma(\Theta(\sigma_n) + \epsilon).
\end{aligned}
$$

Since $\sum_{p \in \mathcal{J}_R} \alpha_p \gg \Theta(\sigma_n) + \epsilon$, we know $\lim_{\gamma \to \infty} \mathcal{L}_{CE}(\boldsymbol{F}) = o(1)$ and

$$
\lim_{\gamma \to \infty} \mathbb{P}_{(\boldsymbol{X}, y) \sim \mathcal{D}} \left[ \exists \boldsymbol{\Delta} \in \left( \mathbb{R}^d \right)^P \text{ s.t. } \|\boldsymbol{\Delta}\|_\infty \le \epsilon, \operatorname{argmax}_{i \in [k]} F_i(\boldsymbol{X} + \boldsymbol{\Delta}) \ne y \right] = o(1).
$$

$\square$

## C.3 ANALYZING LEARNING PROCESS VIA WEIGHT-FEATURE CORRELATIONS

Proposition 3.1 and Proposition 3.2 demonstrate that a network is vulnerable to adversarial perturbations if it relies solely on learning non-robust features. Conversely, a network that learns all robust features can achieve a state of robustness. In general, by calculating the gradient of empirical loss, it seems that the whole weights during gradient-based training will have the following form

$$
\boldsymbol{w}_{i,r} \approx A_{i,r} \boldsymbol{u}_i + B_{i,r} \boldsymbol{v}_i + Noise,
$$

where $A_{i,r}, B_{i,r} > 0$ represent the coefficients for learning robust and non-robust features, respectively, and the 'Noise' term encompasses elements learned from other non-diagonal features $\boldsymbol{u}_j, \boldsymbol{v}_j (j \ne i)$, as well as random noise $\boldsymbol{\xi}_p$.

Therefore, we know that the network learns the $i$-th class if and only if either $A_{i,r}$ or $B_{i,r}$ is sufficiently large. However, to robustly learn the $i$-th class, the network must primarily learn the robust feature $\boldsymbol{u}_i$, rather than the non-robust feature $\boldsymbol{v}_i$, which motivates us to analyze the feature learning process of standard training and adversarial training to understand the underlying mechanism why adversarial examples exist and how adversarial training algorithm works.

# D  DETAILED PROOF FOR SECTION 5

In this section, we provide a detailed proof for Section 5, considering the simplified setting where the data is noiseless and we use population risk instead of empirical risk. For the general case (empirical risk with data noise), we assert that the proof idea is similar to this simplified case. This is because we can demonstrate that noise terms are always sufficiently small under our setting, as shown in the next section (Appendix E).

## D.1  PROOF FOR STANDARD TRAINING

First, we present the restatement of Theorem 4.3 under the simplified setting.

**Theorem D.1** (Restatement of Theorem 4.3 Under the Simplified Setting, Standard Training Converges to Non-robust Global Minima). *For sufficiently large $d$, suppose we train the model using the standard training starting with population risk from the random initialization, then after $T = \Theta(\mathrm{poly}(d)/\eta)$ iterations, with probability $1 - o(1)$ over the randomness of weight initialization, the model $\boldsymbol{F}^{(T)}$ satisfies:*

- *Non-robust features are learned: $\mathbb{P}_{(\boldsymbol{X_f}, y) \sim \mathcal{D}_{\mathcal{F}_{NR}}} \left[ \mathrm{argmax}_{i \in [k]} F_i^{(T)}(\boldsymbol{X_f}) \neq y \right] = 0.$*

- *Standard test accuracy is good: $\mathbb{P}_{(\boldsymbol{X}, y) \sim \mathcal{D}} \left[ \mathrm{argmax}_{i \in [k]} F_i^{(T)}(\boldsymbol{X}) \neq y \right] = 0.$*

- *Robust test accuracy is bad: for any given data $(\boldsymbol{X}, y)$, using the following perturbation $\boldsymbol{\Delta}(\boldsymbol{X}, y) := (\boldsymbol{\delta}_1, \boldsymbol{\delta}_2, \ldots, \boldsymbol{\delta}_P)$, where $\boldsymbol{\delta}_p := -\beta_p \boldsymbol{v}_y + \epsilon \boldsymbol{v}_{y'}$ for $p \in \mathcal{J}_{NR}$; $\boldsymbol{\delta}_p := \boldsymbol{0}$ for $p \in \mathcal{J}_R$, and $y'$ is randomly chosen from $[k] \setminus \{y\}$ (which does not depend on the model $\boldsymbol{F}^{(T)}$ and is illustrated in Figure 2), we have*

$$\mathbb{P}_{(\boldsymbol{X}, y) \sim \mathcal{D}} \left[ \mathrm{argmax}_{i \in [k]} F_i^{(T)}(\boldsymbol{X} + \boldsymbol{\Delta}(\boldsymbol{X}, y)) \neq y \right] = 1.$$

**Proof Sketch.** To prove Theorem D.1, we study the feature learning process of standard training by decomposing the weights of the neural network into a linear combination of all features ($\boldsymbol{f} \in \mathcal{F}$). We then demonstrate that non-diagonal weight-feature correlations (i.e., $\langle \boldsymbol{w}_{i,r}, \boldsymbol{u}_j \rangle$ and $\langle \boldsymbol{w}_{i,r}, \boldsymbol{v}_j \rangle$, where $j \neq i$) are always smaller than their diagonal counterparts (i.e., $\langle \boldsymbol{w}_{i,r}, \boldsymbol{u}_i \rangle$ and $\langle \boldsymbol{w}_{i,r}, \boldsymbol{v}_i \rangle$). Next, by applying the Tensor Power Method Lemma (Allen-Zhu & Li, 2023a), we show that non-robust feature learning will dominate throughout the entire process, which directly implies that the network converges to a non-robust solution.

Now, we give the detailed proof as follows.

### D.1.1  WEIGHT DECOMPOSITION

By analyzing the gradient descent update, we derive the following weight decomposition lemma, which represents each weight through weight-feature correlations.

**Lemma D.2** (Weight Decomposition Under the Simplified Setting). *For any time $t \geq 0$, each neuron $\boldsymbol{w}_{i,r}$ $((i, r) \in [k] \times [m])$, we have*

$$\boldsymbol{w}_{i,r}^{(t)} = \underbrace{\boldsymbol{P}_{\mathcal{F}}^{\perp} \boldsymbol{w}_{i,r}^{(0)}}_{orthogonal\ init} + \underbrace{A_{i,r}^{(t)} \boldsymbol{u}_i + B_{i,r}^{(t)} \boldsymbol{v}_i}_{diagonal\ correlations} + \underbrace{\sum_{y \neq i} (C_{i,r,y}^{(t)} \boldsymbol{u}_y + D_{i,r,y}^{(t)} \boldsymbol{v}_y)}_{non\text{-}diagonal\ correlations},$$

*where $A_{i,r}^{(t)}, B_{i,r}^{(t)}, C_{i,r,y}^{(t)}$ and $D_{i,r,y}^{(t)}$ are some time-variant coefficients, and $\boldsymbol{P}_{\mathcal{F}}^{\perp} := \mathcal{I}_d - \sum_{\boldsymbol{f} \in \mathcal{F}} \boldsymbol{f} \boldsymbol{f}^{\top}$ projects a vector into all features $\boldsymbol{f} \in \mathcal{F}$'s orthogonal complementary space.*

*Proof.* Notice that the following update iteration:

$$\boldsymbol{w}_{i,r}^{(t+1)} = \boldsymbol{w}_{i,r}^{(t)} - \eta \mathbb{E}_{(\boldsymbol{X}, y) \sim \mathcal{D}} \left[ \nabla_{\boldsymbol{w}_{i,r}} \mathcal{L}_{CE} \left( \boldsymbol{F}^{(t)}; \boldsymbol{X}, y \right) \right]$$

$$= \boldsymbol{w}_{i,r}^{(t)} + \eta \mathbb{E}_{(\boldsymbol{X}, y) \sim \mathcal{D}} \left[ \left( \mathbb{1}_{\{y=i\}} - \mathrm{logit}_i(\boldsymbol{F}^{(t)}, \boldsymbol{X}) \right) \sum_{p \in [P]} \widetilde{\mathrm{ReLU}}'(\langle \boldsymbol{w}_{i,r}^{(t)}, \boldsymbol{x}_p \rangle) \boldsymbol{x}_p \right].$$

Due to the definition of patch data $\boldsymbol{x}_p$, we can derive Lemma D.2 by induction. □

At the outset of the algorithm, we establish that the feature-weight correlations possess the following characteristic property.

**Lemma D.3.** *For each feature $\boldsymbol{f} \in \mathcal{F}$ and each $i \in [k]$, with probability $1 - o(1)$, we have* $\max_{r \in [m]} \langle \boldsymbol{w}_{i,r}^{(0)}, \boldsymbol{f} \rangle = \Theta(\log(m)/\sqrt{d})$.

*Proof.* Due to the definition of all features and random initialization $\boldsymbol{w}_{i,r}^{(0)} \sim \mathcal{N}(\mathbf{0}, \frac{1}{d}\mathcal{I}_d)$, we know that, for each $i \in [k]$, $\boldsymbol{f} \in \mathcal{F}$, it holds that $\langle \boldsymbol{w}_{i,r}^{(0)}, \boldsymbol{f} \rangle$ are i.i.d. Gaussian random variables with the same variance $1/d$. By applying Lemma B.1, we can derive the result above. □

Then, we can present the learning process by the following dynamics of weight-feature correlations.

**Lemma D.4** (Feature Learning Iteration for Standard Training Under the Simplified Setting). *During standard training, for any time $t \geq 0$ and pair $(i,r) \in [k] \times [m], y \in [k] \setminus \{i\}$, the two sequences $\{A_{i,r}^{(t)}\}, \{B_{i,r}^{(t)}\}, \{C_{i,r,y}^{(t)}\}$ and $\{D_{i,r,y}^{(t)}\}$ satisfy:*

$$
\begin{cases}
A_{i,r}^{(t+1)} = A_{i,r}^{(t)} + \frac{\eta}{k} \mathbb{E}_{\mathcal{D}_{\mathcal{J},i}, \mathcal{D}_{\alpha,i}} \left[ \left(1 - \text{logit}_i(\boldsymbol{F}^{(t)}, \boldsymbol{X})\right) \sum_{p \in \mathcal{J}_R} \widetilde{\text{ReLU}}' \left(\alpha_p A_{i,r}^{(t)}\right) \alpha_p \right], \\
B_{i,r}^{(t+1)} = B_{i,r}^{(t)} + \frac{\eta}{k} \mathbb{E}_{\mathcal{D}_{\mathcal{J},i}, \mathcal{D}_{\beta,i}} \left[ \left(1 - \text{logit}_i(\boldsymbol{F}^{(t)}, \boldsymbol{X})\right) \sum_{p \in \mathcal{J}_{NR}} \widetilde{\text{ReLU}}' \left(\beta_p B_{i,r}^{(t)}\right) \beta_p \right] \\
C_{i,r,y}^{(t+1)} = C_{i,r,y}^{(t)} + \frac{\eta}{k} \mathbb{E}_{\mathcal{D}_{\mathcal{J},y}, \mathcal{D}_{\alpha,y}} \left[ -\text{logit}_i(\boldsymbol{F}^{(t)}, \boldsymbol{X}) \sum_{p \in \mathcal{J}_R} \widetilde{\text{ReLU}}' \left(\alpha_p C_{i,r,y}^{(t)}\right) \alpha_p \right], \\
D_{i,r,y}^{(t+1)} = D_{i,r,y}^{(t)} + \frac{\eta}{k} \mathbb{E}_{\mathcal{D}_{\mathcal{J},y}, \mathcal{D}_{\beta,y}} \left[ -\text{logit}_i(\boldsymbol{F}^{(t)}, \boldsymbol{X}) \sum_{p \in \mathcal{J}_{NR}} \widetilde{\text{ReLU}}' \left(\beta_p D_{i,r,y}^{(t)}\right) \beta_p \right].
\end{cases}
$$

*Proof.* Notice that we have the following update iteration:

$$
\boldsymbol{w}_{i,r}^{(t+1)} = \boldsymbol{w}_{i,r}^{(t)} + \eta \mathbb{E}_{(\boldsymbol{X},y) \sim \mathcal{D}} \left[ \left(1_{\{y=i\}} - \text{logit}_i(\boldsymbol{F}^{(t)}, \boldsymbol{X})\right) \sum_{p \in [P]} \widetilde{\text{ReLU}}' (\langle \boldsymbol{w}_{i,r}^{(t)}, \boldsymbol{x}_p \rangle) \boldsymbol{x}_p \right].
$$

Then, we project the iteration above onto the directions of features to derive

$$
\langle \boldsymbol{w}_{i,r}^{(t+1)}, \boldsymbol{f} \rangle = \langle \boldsymbol{w}_{i,r}^{(t)}, \boldsymbol{f} \rangle + \eta \mathbb{E}_{(\boldsymbol{X},y) \sim \mathcal{D}} \left[ \left(1_{\{y=i\}} - \text{logit}_i(\boldsymbol{F}^{(t)}, \boldsymbol{X})\right) \sum_{p \in [P]} \widetilde{\text{ReLU}}' (\langle \boldsymbol{w}_{i,r}^{(t)}, \boldsymbol{x}_p \rangle) \langle \boldsymbol{x}_p, \boldsymbol{f} \rangle \right],
$$

where feature vector $\boldsymbol{f} \in \mathcal{F}$.

For feature $\boldsymbol{f} \in \{\boldsymbol{u}_i, \boldsymbol{v}_i\}$ (diagonal features), according to the definition of patch data $\boldsymbol{x}_p$, we know that $\langle \boldsymbol{x}_p, \boldsymbol{f} \rangle$ is not zero if and only if data point $(\boldsymbol{X}, y)$ belongs to class $i$. Thus, we derive

$$
\begin{cases}
A_{i,r}^{(t+1)} = A_{i,r}^{(t)} + \frac{\eta}{k} \mathbb{E}_{\mathcal{D}_{\mathcal{J},i}, \mathcal{D}_{\alpha,i}} \left[ \left(1 - \text{logit}_i(\boldsymbol{F}^{(t)}, \boldsymbol{X})\right) \sum_{p \in \mathcal{J}_R} \widetilde{\text{ReLU}}' \left(\alpha_p A_{i,r}^{(t)}\right) \alpha_p \right], \\
B_{i,r}^{(t+1)} = B_{i,r}^{(t)} + \frac{\eta}{k} \mathbb{E}_{\mathcal{D}_{\mathcal{J},i}, \mathcal{D}_{\beta,i}} \left[ \left(1 - \text{logit}_i(\boldsymbol{F}^{(t)}, \boldsymbol{X})\right) \sum_{p \in \mathcal{J}_{NR}} \widetilde{\text{ReLU}}' \left(\beta_p B_{i,r}^{(t)}\right) \beta_p \right].
\end{cases}
$$

For feature $\boldsymbol{f} \in \{\boldsymbol{u}_j, \boldsymbol{v}_j\}_{j \in [k] \setminus \{i\}}$ (non-diagonal features), according to the definition of patch data $\boldsymbol{x}_p$, we know that $\langle \boldsymbol{x}_p, \boldsymbol{f} \rangle$ is not zero if and only if data point $(\boldsymbol{X}, y)$ belongs to class $j$. Thus, we derive

$$
\begin{cases}
C_{i,r,y}^{(t+1)} = C_{i,r,y}^{(t)} + \frac{\eta}{k} \mathbb{E}_{\mathcal{D}_{\mathcal{J},y}, \mathcal{D}_{\alpha,y}} \left[ -\text{logit}_i(\boldsymbol{F}^{(t)}, \boldsymbol{X}) \sum_{p \in \mathcal{J}_R} \widetilde{\text{ReLU}}' \left(\alpha_p C_{i,r,y}^{(t)}\right) \alpha_p \right], \\
D_{i,r,y}^{(t+1)} = D_{i,r,y}^{(t)} + \frac{\eta}{k} \mathbb{E}_{\mathcal{D}_{\mathcal{J},y}, \mathcal{D}_{\beta,y}} \left[ -\text{logit}_i(\boldsymbol{F}^{(t)}, \boldsymbol{X}) \sum_{p \in \mathcal{J}_{NR}} \widetilde{\text{ReLU}}' \left(\beta_p D_{i,r,y}^{(t)}\right) \beta_p \right].
\end{cases}
$$

□

### D.1.2 LOGIT APPROXIMATION

To analyze the feature learning iteration, we require the following approximations for both diagonal and non-diagonal logits. Indeed, we can divide the full learning process into two stages: During the early stage, the non-diagonal logits and the diagonal logits maintain a constant relationship, and then they decrease at a rate proportional to $O\left(\frac{1}{d^c}\right)$, where $c$ reflects the order of the diagonal function output $F_y^{(t)}(\boldsymbol{X})$. First, we give the following diagonal logit approximation lemma.

**Lemma D.5** (Diagonal Logit Approximation). *For any data point $(\boldsymbol{X}, y) \sim \mathcal{D}$, suppose that $\max_{i \in [k], r \in [m]} B_{i,r}^{(t)} = O(\log(d))$ and $\max_{i \in [k], r \in [m], y \in [k] \setminus \{i\}} C_{i,r,y}^{(t)} = O(\log(d)/\sqrt{d})$ and $\max_{i \in [k], r \in [m], y \in [k] \setminus \{i\}} D_{i,r,y}^{(t)} = O(\log(d)/\sqrt{d})$, and $F_y(\boldsymbol{X}) \geq c \log(d)$ for some $c \geq 0$, then we have the following approximation of diagonal logit:*

$$1 - \mathrm{logit}_y(\boldsymbol{F}^{(t)}, \boldsymbol{X}) = \begin{cases} \Theta(1) & , \text{if } c = 0; \\ O\left(\frac{1}{d^c}\right) & , \text{if } c > 0. \end{cases}$$

*Proof.* By assumptions, we know that all of the off-diagonal correlations are at most $O(\log(d)/\sqrt{d})$. Thus, we have, for each class $j \in [k] \setminus \{y\}$,

$$
\begin{aligned}
F_j^{(t)}(\boldsymbol{X}) &= \sum_{r \in [m]} \sum_{p \in [P]} \widetilde{\mathrm{ReLU}}(\langle \boldsymbol{w}_{j,r}, \boldsymbol{x}_p \rangle) \\
&= \sum_{r \in [m]} \sum_{p \in \mathcal{J}_R} \frac{\alpha_p^q}{q\varrho^{q-1}} (C_{j,r,y}^{(t)})^q + \sum_{r \in [m]} \sum_{p \in \mathcal{J}_{NR}} \frac{\beta_p^q}{q\varrho^{q-1}} (D_{j,r,y}^{(t)})^q \\
&\leq m \left( \sum_{p \in \mathcal{J}_R} \frac{\alpha_p^q}{q\varrho^{q-1}} (\max_{r \in [m]} C_{j,r,y}^{(t)})^q + \sum_{p \in \mathcal{J}_{NR}} \frac{\beta_p^q}{q\varrho^{q-1}} (\max_{r \in [m]} D_{j,r,y}^{(t)})^q \right) \\
&\leq \tilde{O}(1/d^{\frac{q}{2}}),
\end{aligned}
$$

which implies that

$$\exp(F_j^{(t)}(\boldsymbol{X})) \leq 1 + \tilde{O}(1/d^{\frac{q}{2}}),$$

where we use the inequality $e^x \leq 1 + x + x^2$ for $x \leq 1$.

Now by the assumption that $F_y(\boldsymbol{X}) \geq c \log(d)$, we know

$$
\begin{aligned}
1 - \mathrm{logit}_y(\boldsymbol{F}^{(t)}, \boldsymbol{X}) &= 1 - \frac{\exp(F_y(\boldsymbol{X}))}{\exp(F_y(\boldsymbol{X})) + \sum_{j \neq y} \exp(F_j(\boldsymbol{X}))} \\
&\leq 1 - \frac{d^c}{d^c + (k-1) + o(1)} = O(1/d^c).
\end{aligned}
$$

$\square$

Then, we give the following non-diagonal logit approximation lemma.

**Lemma D.6** (Non-diagonal Logit Approximation). *For any data point $(\boldsymbol{X}, y) \sim \mathcal{D}$, suppose that $\max_{i \in [k], r \in [m]} B_{i,r}^{(t)} = O(\log(d))$ and $\max_{i \in [k], r \in [m], y \in [k] \setminus \{i\}} C_{i,r,y}^{(t)} = O(\log(d)/\sqrt{d})$ and $\max_{i \in [k], r \in [m], y \in [k] \setminus \{i\}} D_{i,r,y}^{(t)} = O(\log(d)/\sqrt{d})$, and $F_y(\boldsymbol{X}) \geq c \log(d)$ for some $c \geq 0$, then we have the following approximation of non-diagonal logit, for $i \neq y$:*

$$\mathrm{logit}_i(\boldsymbol{F}^{(t)}, \boldsymbol{X}) = \begin{cases} \Theta(1/k) & , \text{if } c = 0; \\ O\left(\frac{1}{d^c}\right) & , \text{if } c > 0. \end{cases}$$

*Proof.* Similar to the diagonal case, we have, for each class $j \in [k] \setminus \{y\}$,

$$F_j^{(t)}(\boldsymbol{X}) \leq \tilde{O}(1/d^{\frac{q}{2}}).$$

Then, we know

$$
\text{logit}_i(\boldsymbol{F}^{(t)}, \boldsymbol{X}) = \frac{\exp(F_i(\boldsymbol{X}))}{\exp(F_y(\boldsymbol{X})) + \exp(F_i(\boldsymbol{X})) + \sum_{j \neq y,i} \exp(F_j(\boldsymbol{X}))}
$$
$$
\leq \frac{1 + \tilde{O}(1/d^{\frac{q}{2}})}{d^c + (k-1)} = O(1/d^c).
$$

$\square$

### D.1.3 NON-DIAGONAL TERMS ARE SMALL

Then, we will show that non-diagonal terms are always small during the full learning process.

**Lemma D.7.** *For every time t, non-diagonal wieght-feature correlations $C_{i,r,y}^{(t)}, D_{i,r,y}^{(t)}$ $((i,r) \in [k] \times [m], y \neq i)$ satisfy that $\max_{r \in [m]} C_{i,r,y}^{(t)} = O(\max_{r \in [m]} C_{i,r,y}^{(0)})$ and $\max_{r \in [m]} D_{i,r,y}^{(t)} = O(\max_{r \in [m]} D_{i,r,y}^{(0)})$ for each $i \in [k], y \in [k] \setminus \{i\}$.*

*Proof.* Notice that, for each pair $i \in [k], r \in [m], y \in [k] \setminus \{i\}$, we have

$$
\begin{cases}
C_{i,r,y}^{(t+1)} = C_{i,r,y}^{(t)} + \frac{\eta}{k} \mathbb{E}_{\mathcal{D}_{\mathcal{J},y}, \mathcal{D}_{\alpha,y}} \left[ -\text{logit}_i(\boldsymbol{F}^{(t)}, \boldsymbol{X}) \sum_{p \in \mathcal{J}_R} \widetilde{\text{ReLU}}' \left( \alpha_p C_{i,r,y}^{(t)} \right) \alpha_p \right], \\
D_{i,r,y}^{(t+1)} = D_{i,r,y}^{(t)} + \frac{\eta}{k} \mathbb{E}_{\mathcal{D}_{\mathcal{J},y}, \mathcal{D}_{\beta,y}} \left[ -\text{logit}_i(\boldsymbol{F}^{(t)}, \boldsymbol{X}) \sum_{p \in \mathcal{J}_{NR}} \widetilde{\text{ReLU}}' \left( \beta_p D_{i,r,y}^{(t)} \right) \alpha_p \right].
\end{cases}
$$

Since $-\text{logit}_i(\boldsymbol{F}^{(t)}, \boldsymbol{X})$ is negative and $\widetilde{\text{ReLU}}'(z) = 0$ for $z \geq 0$, we can prove the above lemma by induction. $\square$

### D.1.4 ANALYSIS OF FEATURE LEARNING PROCESS FOR STANDARD TRAINING

Now, we present an enhanced analysis of the feature learning process for standard training scenarios. In fact, the learning process can be conceptualized as comprising two distinct phases: Initially, all weight-feature correlations are closely aligned with their initial values, implying that the activation of all neurons occurs within the polynomial regime of the smoothed $\widetilde{\text{ReLU}}$ function. Subsequently, once the diagonal outputs $F_y(\boldsymbol{X})$ have scaled to an order of $\log(d)$, we demonstrate that the increase in all weight-feature correlations is arrested due to the diminishing impact of small logits.

**Stage I:** Almost neurons lie within the polynomial part of activation $\widetilde{\text{ReLU}}$.

**Lemma D.8.** *During standard training, there exists some time threshold $T_0 > 0$ such that, for any early time $0 \leq t \leq T_0$ and pair $(i,r) \in [k] \times [m]$, the two sequences $\{A_{i,r}^{(t)}\}$ and $\{B_{i,r}^{(t)}\}$ satisfy:*

$$
\begin{cases}
A_{i,r}^{(t+1)} = A_{i,r}^{(t)} + \Theta(\eta) \left( A_{i,r}^{(t)} \right)^{q-1} \mathbb{E}\left[ \sum_{p \in \mathcal{J}_R} \alpha_p^q \right], \\
B_{i,r}^{(t+1)} = B_{i,r}^{(t)} + \Theta(\eta) \left( B_{i,r}^{(t)} \right)^{q-1} \mathbb{E}\left[ \sum_{p \in \mathcal{J}_{NR}} \beta_p^q \right].
\end{cases}
$$

*Proof.* According to Lemma D.4, we know that, for early time $t$, it holds that

$$
\begin{cases}
A_{i,r}^{(t+1)} = A_{i,r}^{(t)} + \frac{\eta}{k} \mathbb{E}_{\mathcal{D}_{\mathcal{J},i}, \mathcal{D}_{\alpha,i}} \left[ \left(1 - \text{logit}_i(\boldsymbol{F}^{(t)}, \boldsymbol{X})\right) \sum_{p \in \mathcal{J}_R} \widetilde{\text{ReLU}}' \left( \alpha_p A_{i,r}^{(t)} \right) \alpha_p \right], \\
B_{i,r}^{(t+1)} = B_{i,r}^{(t)} + \frac{\eta}{k} \mathbb{E}_{\mathcal{D}_{\mathcal{J},i}, \mathcal{D}_{\beta,i}} \left[ \left(1 - \text{logit}_i(\boldsymbol{F}^{(t)}, \boldsymbol{X})\right) \sum_{p \in \mathcal{J}_{NR}} \widetilde{\text{ReLU}}' \left( \beta_p B_{i,r}^{(t)} \right) \beta_p \right].
\end{cases}
$$

And it also holds that

$$
\alpha_p A_{i,r}^{(t)}, \beta_p B_{i,r}^{(t)} \leq \varrho,
$$

which implies that the activation is within polynomial part now.

Combined with Lemma D.5, we derive the lemma above. $\square$

Now, by applying Tensor Power Method Lemma B.2, we can derive the following result.

**Lemma D.9** (Non-robust Feature Learning Dominates at Early Stage). *For each $y \in [k]$, let time $T_y$ denote the first time when $\max_{r \in [m]} B_{y,r}^{(t)}$ reaches $\varrho/\beta$, then we have $\max_{r \in [m]} A_{y,r}^{(T_y)} = O(\text{polylog}(d)/\sqrt{d})$.*

*Proof.* We consider the following sequences $\{x_t\}, \{y_t\}$ and $\{C_t\}$:

$$x_t = \max_{r \in [m]} B_{i,r}^{(t)}, \quad y_t = \max_{r \in [m]} A_{i,r}^{(t)}, \quad C_t = \mathbb{E}\left[\sum_{p \in \mathcal{J}_{NR}} \beta_p^q\right], \quad S = \frac{\mathbb{E}\left[\sum_{p \in \mathcal{J}_R} \alpha_p^q\right]}{\mathbb{E}\left[\sum_{p \in \mathcal{J}_{NR}} \beta_p^q\right]}.$$

Then, we have

$$\begin{cases} x_{t+1} \geq x_t + \Theta(\eta) C_t x_t^{q-1}, \\ y_{t+1} \leq y_t + \Theta(\eta) S C_t y_t^{q-1}. \end{cases}$$

And we also know that, with high probability $1 - o(1)$, we have

$$\frac{x_0}{y_0 S^{\frac{1}{q-2}}} = \frac{\max_{r \in [m]} B_{i,r}^{(0)}}{\max_{r \in [m]} A_{i,r}^{(0)}} \cdot \left(\frac{\mathbb{E}\left[\sum_{p \in \mathcal{J}_{NR}} \beta_p^q\right]}{\mathbb{E}\left[\sum_{p \in \mathcal{J}_R} \alpha_p^q\right]}\right)^{\frac{1}{q-2}} \gg 1 + \frac{1}{\text{polylog}(d)}.$$

Where we use Lemma D.3 and Assumption 2.3.

Finally, by directly applying Tensor Power Method Lemma B.2, we derive the conclusion. $\quad\square$

**Stage II:** Non-robust feature learning arrives at linear region of activation $\widetilde{ReLU}$.

**Lemma D.10.** *For each $y \in [k]$ and $(\boldsymbol{X}, y) \sim \mathcal{D}$, let time $T_y'$ denote the first time such that $F_y(\boldsymbol{X})^{(t)} \geq \log(d)$, then $T_y' = \text{poly}(d) \geq T_y$, and we have $\max_{r \in [m]} A_{y,r}^{(T_y')} = O(\text{polylog}(d)/\sqrt{d})$.*

*Proof.* Now, according to Lemma D.9, we know that $\max_{r \in [m]} B_{y,r}^{(t)} \geq \varrho/\beta$, which manifests that there exists at least one neuron $r^* \in [m]$ has been within the linear regime of activation function. Thus, so long as $F_y(\boldsymbol{X})^{(t)} < \log(d)$ now, we have

$$B_{y,r^*}^{(t+1)} = B_{y,r^*}^{(t)} + \frac{\eta}{k} \mathbb{E}\left[(1 - \text{logit}_y(\boldsymbol{F}^{(t)}, \boldsymbol{X}) \sum_{p \in \mathcal{J}_{NR}} \widetilde{ReLU}'(\beta_p B_{y,r^*}^{(t)}) \beta_p\right]$$

$$\geq B_{y,r^*}^{(t)} + \Theta\left(\frac{\eta}{k}\right) \mathbb{E}\left[\sum_{p \in \mathcal{J}_{NR}} \beta_p\right].$$

Therefore, we know that we can upper bound $T_y$ by the number of iterations it takes $B_{y,r^*}^{(t)}$ to grow to $\Theta(\log(d))$. Indeed, we clearly have that $T_y = O(\log(d)/\eta) = \text{poly}(d)$ for some polynomial in $d$. However, in contrast with $B_{y,r^*}^{(t)}$, for $r \in [m]$ $A_{y,r}^{(t)}$, we can lower bound the number of iterations $T$ it takes for $A_{y,r}^{(t)}$ to grow to by a fixed constant $C$ factor from initialization:

$$T\Theta\left(\frac{\eta C^{q-1} \left(A_{y,r}^{(0)}\right)^{q-1}}{\varrho^{q-1}}\right) \geq (C-1) A_{y,r}^{(0)},$$

which implies that

$$T \geq \Theta\left(\frac{\varrho^{q-1}}{\eta A_{y,r}^{(0)}}\right) \geq \Theta\left(\frac{\varrho^{q-1} d^{\frac{q-2}{2}}}{\eta}\right) \gg \omega\left(\frac{\log(d)}{\eta}\right) = \omega(T_{y'}).$$

$\quad\square$

The final remaining task is to show $F_y^{(t)}(\boldsymbol{X})$ will keep $\Theta(\text{poly}(d))$ for all polynomial time $t$.

**Lemma D.11.** *For all time $t = O(\text{poly}(d)) \geq T_y'$ and each $(\boldsymbol{X}, y) \sim \mathcal{D}$, we have $F_y^{(t)}(\boldsymbol{X}) = O(\log(d))$, and $\max_{r \in [m]} A_{y,r}^{(t)} = O(\text{polylog}(d)/\sqrt{d})$.*

*Proof.* Firstly, we can form the following upper bound for the gradient updates

$$
\begin{aligned}
B_{y,r^*}^{(t+1)} &= B_{y,r^*}^{(t)} + \frac{\eta}{k} \mathbb{E}\left[ (1 - \text{logit}_y(\boldsymbol{F}^{(t)}, \boldsymbol{X}) \sum_{p \in \mathcal{J}_{NR}} \widetilde{\text{ReLU}}'(\beta_p B_{y,r^*}^{(t)}) \beta_p \right] \\
&\geq B_{y,r^*}^{(t)} + \Theta\left(\frac{\eta}{k}\right) \left(1 - \text{logit}_y(\boldsymbol{F}^{(t)}, \boldsymbol{X})\right).
\end{aligned}
$$

Then, we know that it follows that it takes at least $\Theta(d \log(d)/\eta)$ iterations (since the correlations must grow at least $\log(d)$) from $T_y'$ for $F_y(\boldsymbol{X})$ to reach $2\log(d)$. Now let $T_y^{(}c)$ denote the number of iterations it takes for $F_y(\boldsymbol{X})$ to cross $c\log(d)$ after crossing $(c-1)\log(d)$ for the first time. For $c \geq 2$, we necessarily have that $T_y^{(}c) = \Omega(dT_y^{(c-1)})$ by induction.

Let us now further define $T_f$ to be the first iteration at which $F_y^{(T_f)}(\boldsymbol{X}) \geq f(d)\log d$ for some $f(d) = \omega(1)$. However, we have from the above discussion that:

$$
\begin{aligned}
T_f &\geq \Omega(\text{poly}(d)) + \sum_{c=0}^{f(d)-2} \Omega\left(\frac{d^c \log d}{\eta}\right) \\
&\geq \Omega\left(\frac{\log d \left(d^{f(d)-1} - 1\right)}{\eta(d-1)}\right) \\
&\geq \omega(\text{poly}(d))
\end{aligned}
$$

So $F_y^{(t)}(\boldsymbol{X}) = O(\log(d))$ for all $t = O(\text{poly}(d))$. An identical analysis also works for the robust feature correlations $A_{y,r}^{(t)}$, so we are done. $\qquad\square$

### D.1.5 PROOF OF THEOREM D.1

Now, we will prove Theorem D.1, which includes the following three parts.

**Lemma D.12** (Standard Accuracy is Good). *For $T = \Theta(\text{poly}(d))$ and each data point $(\boldsymbol{X}, y) \sim \mathcal{D}$, with probability $1 - o(1)$, we have $F_y^{(T)}(\boldsymbol{X}) > F_i^{(T)}(\boldsymbol{X}), \forall i \in [k] \setminus \{y\}$.*

*Proof.* According to Lemma D.11, we know that, for each data point $(\boldsymbol{X}, y) \sim \mathcal{D}$ and time $T = \Theta(\text{poly}(d)/\eta)$, with probability $1 - o(1)$, it holds that

$$
F_y(\boldsymbol{X}) = \Theta(\log(d)).
$$

However, the non-diagonal outputs can be upper bounded as:

$$
\begin{aligned}
F_i(\boldsymbol{X}) &= \sum_{r \in [m]} \sum_{p \in [P]} \widetilde{\text{ReLU}}(\langle \boldsymbol{w}_{i,r}, \boldsymbol{x}_p \rangle) \\
&= \sum_{r \in [m]} \sum_{p \in \mathcal{J}_R} \widetilde{\text{ReLU}}(\alpha_p C_{i,r,y}^{(t)}) + \sum_{r \in [m]} \sum_{p \in \mathcal{J}_{NR}} \widetilde{\text{ReLU}}(\alpha_p D_{i,r,y}^{(t)}) \\
&\leq m \left( \sum_{p \in \mathcal{J}_R} \widetilde{\text{ReLU}}(\alpha_p \max_{r \in [m]} C_{i,r,y}^{(t)}) + \sum_{p \in \mathcal{J}_{NR}} \widetilde{\text{ReLU}}(\beta_p \max_{r \in [m]} D_{i,r,y}^{(t)}) \right) \\
&\leq \tilde{O}(1/d^{q/2}).
\end{aligned}
$$

Where we use Lemma D.7 to upper bound the non-diagonal weight-feature correlations (they always lie within the polynomial part of activation function).

Thus, we derive that $F_y(\boldsymbol{X}) > F_i(\boldsymbol{X})$ for each $i \in [k] \setminus \{y\}$. $\qquad\square$

**Lemma D.13** (Non-robust Features are Learned Well). *For $T = \Theta(\text{poly}(d))$ and each data point $(\boldsymbol{X}, y) \sim \mathcal{D}_{\mathcal{F}_{NR}}$, with probability $1 - o(1)$, we have $F_y^{(T)}(\boldsymbol{X}) > F_i^{(T)}(\boldsymbol{X}), \forall i \in [k] \setminus \{y\}$.*

*Proof.* Since we have that $\max_{r \in [m]} B_{y,r}^{(T)} = \Theta(\log(d))$ for each class $y \in [k]$, it suggests $F_y^{(T)}(\boldsymbol{X}) = \Theta(\log(d))$. For non-diagonal outputs, we have results similar to Lemma D.12, i.e. $F_i^{(T)}(\boldsymbol{X}) = \tilde{O}(1/d^{q/2})$, which immediately derives the lemma above. $\qquad\square$

**Lemma D.14** (Standard Training Converges to Non-robust Solution). *For $T = \Theta(\text{poly}(d))$ and each data point $(\boldsymbol{X}, y) \sim \mathcal{D}$, let perturbation $\boldsymbol{\Delta}(\boldsymbol{X}, y) := (\boldsymbol{\delta}_1, \boldsymbol{\delta}_2, \ldots, \boldsymbol{\delta}_P)$, where $\boldsymbol{\delta}_p := -\beta_p \boldsymbol{v}_y + \epsilon \boldsymbol{v}_{y'}$ for $p \in \mathcal{J}_{NR}$; $\boldsymbol{\delta}_p := \boldsymbol{0}$ for $p \in \mathcal{J}_R$, and $y'$ is randomly chosen from $[k] \setminus \{y\}$, then, with probability $1 - o(1)$, we have $F_{y'}^{(T)}(\boldsymbol{X} + \boldsymbol{\Delta}(\boldsymbol{X}, y)) > F_i^{(T)}(\boldsymbol{X} + \boldsymbol{\Delta}(\boldsymbol{X}, y)), \forall i \in [k] \setminus \{y'\}$.*

*Proof.* By the definition of perturbation $\boldsymbol{\Delta}(\boldsymbol{X}, y)$ that replaces all non-robust feature patches $\beta_p \boldsymbol{v}_y$ by non-robust feature $\boldsymbol{v}_{y'}$ from another class $y'$, we have

$$
\begin{aligned}
F_{y'}^{(T)}(\boldsymbol{X} + \boldsymbol{\Delta}(\boldsymbol{X}, y)) &= \sum_{r \in [m]} \sum_{p \in [P]} \widetilde{\text{ReLU}}(\boldsymbol{w}_{y',r}, \boldsymbol{x}_p + \boldsymbol{\delta}_p) \\
&\geq \sum_{p \in [P]} \widetilde{\text{ReLU}}(\epsilon \max_{r \in [m]} B_{y',r}^{(T)} + O(\text{polylog}(d)/\sqrt{d})) \\
&\geq \Theta(\log(d)) \gg \tilde{\Omega}(1/d^{q/2}) \geq \max_{i \in [k] \setminus \{y'\}} F_i^{(T)}(\boldsymbol{X} + \boldsymbol{\Delta}(\boldsymbol{X}, y)),
\end{aligned}
$$

which shows this theorem. $\qquad\square$

## D.2 Proof for Adversarial Training

First, we present the restatement of Theorem 4.4 under the simplified setting.

**Theorem D.15** (Restatement of Theorem 4.4 Under the Simplified Setting, Adversarial Training Converges to Robust Global Minima). *For sufficiently large $d$, suppose we train the model using the adversarial training algorithm starting with adversarial population risk from the random initialization, then after $T = \Theta(\text{poly}(d)/\eta)$ iterations, the model $\boldsymbol{F}^{(T)}$ satisfies:*

- *Robust features are learned: $\mathbb{P}_{(\boldsymbol{X_f}, y) \sim \mathcal{D}_{\mathcal{F}_R}} \left[ \text{argmax}_{i \in [k]} F_i^{(T)}(\boldsymbol{X_f}) \neq y \right] = o(1)$.*

- *Robust test accuracy is good:*
$$
\mathbb{P}_{(\boldsymbol{X}, y) \sim \mathcal{D}} \left[ \exists \boldsymbol{\Delta} \in \left(\mathbb{R}^d\right)^P \text{ s.t. } \|\boldsymbol{\Delta}\|_\infty \leq \epsilon, \text{argmax}_{i \in [k]} F_i^{(T)}(\boldsymbol{X} + \boldsymbol{\Delta}) \neq y \right] = o(1).
$$

**Proof Sketch.** The proof of this theorem can also be divided into two stages (according to the activation regions of weight-feature correlations). Unlike standard training, the first phase of adversarial training involves a phase transition. In the initial phase, robust feature learning and non-robust feature learning exhibit behaviors similar to those in standard training. However, once non-robust feature learning reaches a certain magnitude, adversarial training suppresses non-robust feature learning through adversarial examples found by a gradient ascent algorithm, while robust feature learning continues to grow.

Now, we give the detailed proof as follows.

### D.2.1 Weight Decomposition

Since our algorithm is based on gradient information (which runs one-step gradient ascent algorithm for finding adversarial examples and runs gradient descent algorithm for training the neural network model), we can derive a weight decomposition theorem similar to that of standard training.

**Lemma D.16** (Weight Decomposition for Adversarial Training). *For any time $t \geq 0$, each neuron $\boldsymbol{w}_{i,r} ((i, r) \in [k] \times [m])$, we have*

$$
\boldsymbol{w}_{i,r}^{(t)} = \underbrace{\sum_{y \in [k]} \sum_{s \in [m]} E_{i,r,y,s}^{(t)} \boldsymbol{P}_{\mathcal{F}}^\perp \boldsymbol{w}_{y,s}^{(0)}}_{\text{orthogonal to all features}} + \underbrace{A_{i,r}^{(t)} \boldsymbol{u}_i + B_{i,r}^{(t)} \boldsymbol{v}_i}_{\text{diagonal correlations}} + \underbrace{\sum_{j \neq i} (C_{i,r,j}^{(t)} \boldsymbol{u}_j + D_{i,r,j}^{(t)} \boldsymbol{v}_j)}_{\text{non-diagonal correlations}},
$$

where $A_{i,r}^{(t)}$, $B_{i,r}^{(t)}$, $C_{i,r,j}^{(t)}$, $D_{i,r,j}^{(t)}$ and $E_{i,r,y,s}^{(t)}$ are some time-variant coefficients, and $\boldsymbol{P}_{\mathcal{F}}^{\perp} := \mathcal{I}_d - \sum_{\boldsymbol{f} \in \mathcal{F}} \boldsymbol{f}\boldsymbol{f}^{\top}$ projects a vector into all features $\boldsymbol{f} \in \mathcal{F}$'s orthogonal complementary space.

*Proof.* Notice that the following update iteration:

$$
\boldsymbol{w}_{i,r}^{(t+1)} = \boldsymbol{w}_{i,r}^{(t)} - \eta \mathbb{E}_{(\boldsymbol{X},y)\sim\mathcal{D}}\left[\nabla_{\boldsymbol{w}_{i,r}}\mathcal{L}_{CE}\left(\boldsymbol{F}^{(t)}; \widetilde{\boldsymbol{X}}^{(t)}, y\right)\right]
$$

$$
= \boldsymbol{w}_{i,r}^{(t)} + \eta \mathbb{E}_{(\boldsymbol{X},y)\sim\mathcal{D}}\left[\left(1_{\{y=i\}} - \mathrm{logit}_i(\boldsymbol{F}^{(t)}, \widetilde{\boldsymbol{X}}^{(t)})\right) \sum_{p\in[P]} \widetilde{\mathrm{ReLU}}'(\langle \boldsymbol{w}_{i,r}^{(t)}, \tilde{\boldsymbol{x}}_p^{(t)}\rangle)\tilde{\boldsymbol{x}}_p^{(t)}\right].
$$

To prove Lemma D.2, we only need the following result that time-variant adversarial examples also align with the span of all features (Lemma D.17). $\qquad\square$

### D.2.2 TIME-VARIANT ADVERSARIAL EXAMPLES

By analyzing the gradient ascent algorithm, we can derive the following decomposition theorem regarding adversarial examples.

**Lemma D.17.** *For any time $t$ and data point $(\boldsymbol{X}, y) \sim \mathcal{D}$, we have the following decomposition about the time-variant corresponding adversarial example $(\widetilde{\boldsymbol{X}}^{(t)}, y)$, where we use $(\tilde{\boldsymbol{x}}_1, \tilde{\boldsymbol{x}}_2, \ldots, \tilde{\boldsymbol{x}}_p)$ to denote $\widetilde{\boldsymbol{X}}^{(t)}$. Then, it satisfies:*

$$
\tilde{\boldsymbol{x}}_p^{(t)} = \tilde{\alpha}_p^{(t)}\boldsymbol{u}_y + \tilde{\beta}_p^{(t)}\boldsymbol{v}_y + \sum_{j\neq y}(\tilde{\lambda}_{p,j}^{(t)}\boldsymbol{u}_j + \tilde{\mu}_{p,j}^{(t)}\boldsymbol{v}_j) + \sum_{i\in[k]}\sum_{r\in[m]}\tilde{\gamma}_{p,i,r}^{(t)}\boldsymbol{P}_{\mathcal{F}}^{\perp}\boldsymbol{w}_{i,r}^{(0)},
$$

*where $\boldsymbol{P}_{\mathcal{F}}^{\perp} := \mathcal{I}_d - \sum_{\boldsymbol{f}\in\mathcal{F}} \boldsymbol{f}\boldsymbol{f}^{\top}$ projects a vector into all features $\boldsymbol{f} \in \mathcal{F}$'s orthogonal complementary space, and coefficients $\tilde{\alpha}_p^{(t)}$, $\tilde{\beta}_p^{(t)}$, $\tilde{\lambda}_{p,j}^{(t)}$, $\tilde{\mu}_{p,j}^{(t)}$ and $\tilde{\gamma}_{p,i,r}^{(t)}$ are updated by the following iterations*

- *For $p \in \mathcal{J}_R$, we have*

$$
\begin{cases}
\tilde{\alpha}_p^{(t)} = (1 - \min\{\frac{\epsilon}{\alpha_p}, \frac{\tilde{\eta}}{\alpha_p}\sum_{s\in[m]}\widetilde{\mathrm{ReLU}}'(\alpha_p A_{y,s}^{(t)})A_{y,s}^{(t)}\})\alpha_p \\
\tilde{\beta}_p^{(t)} = -\min\{\epsilon, \tilde{\eta}\sum_{s\in[m]}\widetilde{\mathrm{ReLU}}'(\alpha_p A_{y,s}^{(t)})B_{y,s}^{(t)}\} \\
\tilde{\lambda}_{p,j}^{(t)} = -\min\{\epsilon, \tilde{\eta}\sum_{s\in[m]}\widetilde{\mathrm{ReLU}}'(\alpha_p A_{y,s}^{(t)})C_{y,s,j}^{(t)}\} \\
\tilde{\mu}_{p,j}^{(t)} = -\min\{\epsilon, \tilde{\eta}\sum_{s\in[m]}\widetilde{\mathrm{ReLU}}'(\alpha_p A_{y,s}^{(t)})D_{y,s,j}^{(t)}\} \\
\tilde{\gamma}_{p,i,r}^{(t)} = -\min\{\epsilon, \tilde{\eta}\sum_{s\in[m]}\widetilde{\mathrm{ReLU}}'(\alpha_p A_{y,s}^{(t)})E_{y,s,i,r}^{(t)}\}
\end{cases}
$$

- *For $p \in \mathcal{J}_{NR}$, we have*

$$
\begin{cases}
\tilde{\alpha}_p^{(t)} = -\min\{\epsilon, \tilde{\eta}\sum_{s\in[m]}\widetilde{\mathrm{ReLU}}'(\beta_p B_{y,s}^{(t)})A_{y,s}^{(t)}\} \\
\tilde{\beta}_p^{(t)} = (1 - \min\{\frac{\epsilon}{\beta_p}, \frac{\tilde{\eta}}{\beta_p}\sum_{s\in[m]}\widetilde{\mathrm{ReLU}}'(\beta_p B_{y,s}^{(t)})B_{y,s}^{(t)}\})\beta_p \\
\tilde{\lambda}_{p,j}^{(t)} = -\min\{\epsilon, \tilde{\eta}\sum_{s\in[m]}\widetilde{\mathrm{ReLU}}'(\beta_p B_{y,s}^{(t)})C_{y,s,j}^{(t)}\} \\
\tilde{\mu}_{p,j}^{(t)} = -\min\{\epsilon, \tilde{\eta}\sum_{s\in[m]}\widetilde{\mathrm{ReLU}}'(\beta_p B_{y,s}^{(t)})D_{y,s,j}^{(t)}\} \\
\tilde{\gamma}_{p,i,r}^{(t)} = -\min\{\epsilon, \tilde{\eta}\sum_{s\in[m]}\widetilde{\mathrm{ReLU}}'(\beta_p B_{y,s}^{(t)})E_{y,s,i,r}^{(t)}\}
\end{cases}
$$

*Proof.* By substituting the weight decomposition expression into the formula of the gradient ascent algorithm and simplifying, we obtain this lemma. $\qquad\square$

Now, we can derive the following feature learning iteration for adversarial training.

**Lemma D.18** (Feature Learning Iteration for Adversarial Training). *During adversarial training, for any time $t \geq 0$ and pair $(i, r) \in [k] \times [m], j \in [k] \setminus \{i\}$, the two sequences $\{A_{i,r}^{(t)}\}, \{B_{i,r}^{(t)}\}, \{C_{i,r,j}^{(t)}\}$, $\{D_{i,r,j}^{(t)}\}$ and $E_{i,r,y,s}^{(t)}$ satisfy:*

$$
\begin{cases}
A_{i,r}^{(t+1)} = A_{i,r}^{(t)} + \eta \mathbb{E}\left[\left(1_{\{y=i\}} - \text{logit}_i(\boldsymbol{F}^{(t)}, \widetilde{\boldsymbol{X}}^{(t)})\right) \sum_{p \in [P]} \widetilde{\text{ReLU}}' \left(\langle \boldsymbol{w}_{i,r}^{(t)}, \tilde{\boldsymbol{x}}_p^{(t)} \rangle\right) \tilde{\alpha}_p^{(t)}\right], \\
B_{i,r}^{(t+1)} = B_{i,r}^{(t)} + \eta \mathbb{E}\left[\left(1_{\{y=i\}} - \text{logit}_i(\boldsymbol{F}^{(t)}, \widetilde{\boldsymbol{X}}^{(t)})\right) \sum_{p \in [P]} \widetilde{\text{ReLU}}' \left(\langle \boldsymbol{w}_{i,r}^{(t)}, \tilde{\boldsymbol{x}}_p^{(t)} \rangle\right) \tilde{\beta}_p^{(t)}\right], \\
C_{i,r,j}^{(t+1)} = C_{i,r,j}^{(t)} + \eta \mathbb{E}\left[\left(1_{\{y=i\}} - \text{logit}_i(\boldsymbol{F}^{(t)}, \widetilde{\boldsymbol{X}}^{(t)})\right) \sum_{p \in [P]} \widetilde{\text{ReLU}}' \left(\langle \boldsymbol{w}_{i,r}^{(t)}, \tilde{\boldsymbol{x}}_p^{(t)} \rangle\right) \tilde{\lambda}_{p,j}^{(t)}\right], \\
D_{i,r,j}^{(t+1)} = D_{i,r,j}^{(t)} + \eta \mathbb{E}\left[\left(1_{\{y=i\}} - \text{logit}_i(\boldsymbol{F}^{(t)}, \widetilde{\boldsymbol{X}}^{(t)})\right) \sum_{p \in [P]} \widetilde{\text{ReLU}}' \left(\langle \boldsymbol{w}_{i,r}^{(t)}, \tilde{\boldsymbol{x}}_p^{(t)} \rangle\right) \tilde{\mu}_{p,j}^{(t)}\right] \\
E_{i,r,y,s}^{(t+1)} = E_{i,r,y,s}^{(t)} + \eta \mathbb{E}\left[\left(1_{\{y=i\}} - \text{logit}_i(\boldsymbol{F}^{(t)}, \widetilde{\boldsymbol{X}}^{(t)})\right) \sum_{p \in [P]} \widetilde{\text{ReLU}}' \left(\langle \boldsymbol{w}_{i,r}^{(t)}, \tilde{\boldsymbol{x}}_p^{(t)} \rangle\right) \tilde{\gamma}_{p,y,s}^{(t)}\right].
\end{cases}
$$

*Proof.* The proof method is the same as in standard training (Lemma D.4), we simply project the gradient descent recursion onto each feature direction to derive this lemma. $\square$

### D.2.3 LOGIT APPROXIMATION AT ADVERSARIAL EXAMPLES

First, we give the following diagonal adversarial logit approximation lemma.

**Lemma D.19** (Diagonal Adversarial Logit Approximation). *For any adversarial data point $(\widetilde{\boldsymbol{X}}^{(t)}, y) \sim \mathcal{D}$, suppose that $\max_{i \in [k], r \in [m]} A_{i,r}^{(t)} = O(\log(d))$ and $\max_{i \in [k], r \in [m]} B_{i,r}^{(t)} = o(1) \max_{i \in [k], r \in [m], y \in [k] \setminus \{i\}} C_{i,r,y}^{(t)} = O(\log(d)/\sqrt{d})$ and $\max_{i \in [k], r \in [m], y \in [k] \setminus \{i\}} D_{i,r,y}^{(t)} = O(\log(d)/\sqrt{d})$, and $F_y(\widetilde{\boldsymbol{X}}^{(t)}) \geq c \log(d)$ for some $c \geq 0$, then we have the following approximation of logit:*

$$
1 - \text{logit}_y(\boldsymbol{F}^{(t)}, \widetilde{\boldsymbol{X}}^{(t)}) = \begin{cases} \Theta(1) & , \text{if } c = 0; \\ O\left(\frac{1}{d^c}\right) & , \text{if } c > 0. \end{cases}
$$

*Proof.* The proof logic is similar to Lemma D.5. $\square$

Then, we give the following non-diagonal adversarial logit approximation lemma.

**Lemma D.20** (Non-diagonal Adversarial Logit Approximation). *For any adversarial data point $(\widetilde{\boldsymbol{X}}^{(t)}, y) \sim \mathcal{D}$, suppose that $\max_{i \in [k], r \in [m]} A_{i,r}^{(t)} = O(\log(d))$ and $\max_{i \in [k], r \in [m]} B_{i,r}^{(t)} = o(1) \max_{i \in [k], r \in [m], y \in [k] \setminus \{i\}} C_{i,r,y}^{(t)} = O(\log(d)/\sqrt{d})$ and $\max_{i \in [k], r \in [m], y \in [k] \setminus \{i\}} D_{i,r,y}^{(t)} = O(\log(d)/\sqrt{d})$, and $F_y(\widetilde{\boldsymbol{X}}^{(t)}) \geq c \log(d)$ for some $c \geq 0$, for $i \neq y$:*

$$
\text{logit}_i(\boldsymbol{F}^{(t)}, \widetilde{\boldsymbol{X}}^{(t)}) = \begin{cases} \Theta(1/k) & , \text{if } c = 0; \\ O\left(\frac{1}{d^c}\right) & , \text{if } c > 0. \end{cases}
$$

*Proof.* The proof logic is also similar to Lemma D.6. $\square$

### D.2.4 NON-DIAGONAL TERMS ARE SMALL

**Lemma D.21.** *For every time $t$, non-diagonal weight-feature correlations $C_{i,r,y}^{(t)}, D_{i,r,y}^{(t)}$ $((i, r) \in [k] \times [m], y \neq i)$ satisfy that $\max_{r \in [m]} C_{i,r,y}^{(t)} = O(\max_{r \in [m]} C_{i,r,y}^{(0)})$ and $\max_{r \in [m]} D_{i,r,y}^{(t)} = O(\max_{r \in [m]} D_{i,r,y}^{(0)})$ for each $i \in [k], y \in [k] \setminus \{i\}$.*

*Proof.* The proof logic is also similar to Lemma D.7. $\square$

### D.2.5 ANALYSIS OF FEATURE LEARNING PROCESS FOR ADVERSARIAL TRAINING

**Stage I:** Almost neurons lie within the polynomial part of activation $\widetilde{ReLU}$.

**Lemma D.22.** *During adversarial training, there exists some time threshold $T_0 > 0$ such that, for any early time $0 \le t \le T_0$ and pair $(i, r) \in [k] \times [m]$, the two sequences $\{A_{i,r}^{(t)}\}$ and $\{B_{i,r}^{(t)}\}$ satisfy:*

$$
\begin{cases}
A_{i,r}^{(t+1)} \approx A_{i,r}^{(t)} + \Theta(\eta) \left( A_{i,r}^{(t)} \right)^{q-1} \mathbb{E}\left[ \sum_{p \in \mathcal{J}_R} \alpha_p^q \left( 1 - \min\left\{ \frac{\epsilon}{\alpha_p}, \tilde{\Theta}(\tilde{\eta}) \sum_{s \in [m]} \left( A_{i,s}^{(t)} \right)^q \right\} \right)^q \right], \\
B_{i,r}^{(t+1)} \approx B_{i,r}^{(t)} + \Theta(\eta) \left( B_{i,r}^{(t)} \right)^{q-1} \mathbb{E}\left[ \sum_{p \in \mathcal{J}_{NR}} \beta_p^q \left( 1 - \min\left\{ \frac{\epsilon}{\beta_p}, \tilde{\Theta}(\tilde{\eta}) \sum_{s \in [m]} \left( B_{i,s}^{(t)} \right)^q \right\} \right)^q \right].
\end{cases}
$$

*Proof.* The proof logic is similar to Lemma D.4. □

**Phase I:** First, Network Partially Learns Non-Robust Features.

At the beginning, due to our small initialization, we know all feature learning coefficients $A_{i,r}^{(t)}, B_{i,r}^{(t)} = o(1)$, which suggests that the total feature learning $\sum_{s \in [m]} \left( A_{i,s}^{(t)} \right)^q$ and $\sum_{s \in [m]} \left( B_{i,s}^{(t)} \right)^q$ are sufficiently small. Then, the feature learning process is similar to standard training until the non-robust feature learning becomes large.

**Phase II:** Next, Robust Feature Learning Starts Increasing.

By applying Tensor Power Method Lemma B.2, we have the following result.

**Lemma D.23.** *For each $y \in [k]$, let time $T_y$ denote the first time when $\sum_{s \in [m]} \left( B_{y,s}^{(t)} \right)^q$ reaches $\tilde{\Theta}(\eta^{-1})$, then we have $\max_{r \in [m]} A_{y,r}^{(T_y)} = O(\text{polylog}(d)/\sqrt{d})$.*

*Proof.* We choose $x_t = A_{y,r}^{(t)}$ and $y_t = B_{y,s}^{(t)}$. Then, by applying Lemma B.2, we can derive this result as the same way as the proof of Lemma D.9. □

Once the total non-robust feature learning $\sum_{s \in [m]} \left( B_{i,s}^{(t)} \right)^q$ attains an order of $\tilde{\Theta}(\tilde{\eta}^{-1})$, it is known that the non-robust feature learning will stop, due to $\frac{\epsilon}{\beta_p} \gtrsim 1$ and $1 - \tilde{\Theta}(\tilde{\eta}) \sum_{s \in [m]} \left( B_{i,s}^{(t)} \right)^q \approx 0$.

In contrast, the robust feature learning continues to increase since it always holds that $1 - \min\left\{ \frac{\epsilon}{\alpha_p}, \tilde{\Theta}(\tilde{\eta}) \sum_{s \in [m]} \left( A_{i,s}^{(t)} \right)^q \right\} \ge 1 - \frac{\epsilon}{\alpha_p} \ge \Omega(1)$. Thus, the robust feature learning will increase over the non-robust feature learning finally, which can represented as the following lemma.

**Lemma D.24.** *For each $y \in [k]$, let time $T_y'$ denote the first time when $\max_{r \in [m]} A_{y,r}^{(t)}$ reaches $\varrho/\alpha$, then we have $T_y' = O(\text{poly}(d))$ and $\max_{r \in [m]} B_{y,r}^{(T_y')} = O(1/d^{c_0})$.*

*Proof.* The proof logic is similar to Lemma D.10. □

**Stage II:** Robust feature learning arrives at linear region of activation $\widetilde{ReLU}$.

**Lemma D.25.** *For each $y \in [k]$ and $(\boldsymbol{X}, y) \sim \mathcal{D}$, let time $T_y''$ denote the first time such that $F_y^{(t)}(\widetilde{\boldsymbol{X}}^{(t)}) \ge \log(d)$, then $T_y'' = \text{poly}(d) \ge T_y$, and we have $\max_{r \in [m]} B_{y,r}^{(T_y')} = O(\text{polylog}(d)/d^{c_0})$.*

*Proof.* The proof logic is also similar to Lemma D.10. □

**Lemma D.26.** *For all time $t = O(\text{poly}(d)) \ge T_y''$ and each $(\boldsymbol{X}, y) \sim \mathcal{D}$, we have $F_y^{(t)}(\widetilde{\boldsymbol{X}}^{(t)}) = O(\log(d))$, and $\max_{r \in [m]} B_{y,r}^{(t)} = O(\text{polylog}(d)/d^{c_0})$.*

*Proof.* The proof logic is similar to Lemma D.8. □

### D.2.6 PROOF OF THEOREM D.15

**Lemma D.27** (Robust Features are Learned Well). *For $T = \Theta(\mathrm{poly}(d))$ and each data point $(\boldsymbol{X}, y) \sim \mathcal{D}_{\mathcal{F}_R}$, with probability $1 - o(1)$, we have $F_y^{(T)}(\boldsymbol{X}) > F_i^{(T)}(\boldsymbol{X}), \forall i \in [k] \setminus \{y\}$.*

*Proof.* The proof logic is similar to Lemma D.13. $\qquad\square$

**Lemma D.28** (Adversarial Training Converges to Robust Solution). *For $T = \Theta(\mathrm{poly}(d))$ and each data point $(\boldsymbol{X}, y) \sim \mathcal{D}_{\mathcal{F}_R}$, with probability $1 - o(1)$, we have $\forall \boldsymbol{\Delta} \in \left(\mathbb{R}^d\right)^P$ s.t. $\|\boldsymbol{\Delta}\|_\infty \leq \epsilon, \mathrm{argmax}_{i \in [k]} F_i^{(T)}(\boldsymbol{X} + \boldsymbol{\Delta}) = y$.*

*Proof.* For a given data point $(\boldsymbol{X}, y) \sim \mathcal{D}$ and any perturbation $\boldsymbol{\Delta} = (\boldsymbol{\delta}_1, \boldsymbol{\delta}_2, \ldots, \boldsymbol{\delta}_p) \in \left(\mathbb{R}^d\right)^P$ satisfying $\|\boldsymbol{\Delta}\|_\infty \leq \epsilon$, we calculate the perturbed margin as follows.

$$
\begin{aligned}
F_y^{(T)}(\boldsymbol{X} + \boldsymbol{\Delta}) &= \sum_{r \in [m]} \sum_{p \in \mathcal{J}_R} \widetilde{\mathrm{ReLU}}(\langle \boldsymbol{w}_{y,r}^{(T)}, \alpha_p \boldsymbol{u}_y + \boldsymbol{\delta}_p \rangle) + \sum_{r \in [m]} \sum_{p \in \mathcal{J}_{NR}} \widetilde{\mathrm{ReLU}}(\langle \boldsymbol{w}_{y,r}^{(T)}, \beta_p \boldsymbol{v}_y + \boldsymbol{\delta}_p \rangle) \\
&\geq \sum_{p \in \mathcal{J}_R} \widetilde{\mathrm{ReLU}}(\Theta(\alpha_p \max_{r \in [m]} A_{y,r}^{(T)})) \\
&\geq \sum_{p \in \mathcal{J}_R} \Theta(\alpha_p \max_{r \in [m]} A_{y,r}^{(T)}) \\
&\geq \Theta(\log(d)),
\end{aligned}
$$

where we use Lemma D.21 and Lemma D.26 and the first part of Assumption 2.3 (i.e. $\alpha_p \gg \epsilon$).

And for any $j \in [P] \setminus \{y\}$, we have

$$
\begin{aligned}
F_j^{(T)}(\boldsymbol{X} + \boldsymbol{\Delta}) &= \sum_{r \in [m]} \sum_{p \in \mathcal{J}_R} \widetilde{\mathrm{ReLU}}(\langle \boldsymbol{w}_{j,r}^{(T)}, \alpha_p \boldsymbol{u}_y + \boldsymbol{\delta}_p \rangle) + \sum_{r \in [m]} \sum_{p \in \mathcal{J}_{NR}} \widetilde{\mathrm{ReLU}}(\langle \boldsymbol{w}_{j,r}^{(T)}, \beta_p \boldsymbol{v}_y + \boldsymbol{\delta}_p \rangle) \\
&\leq \sum_{p \in \mathcal{J}_R} \widetilde{\mathrm{ReLU}}(\Theta(\alpha_p \max_{r \in [m]} C_{j,r,y}^{(T)})) + \sum_{p \in \mathcal{J}_{NR}} \widetilde{\mathrm{ReLU}}(\Theta(\beta_p \max_{r \in [m]} D_{j,r,y}^{(T)})) \\
&\leq o(\log(d)),
\end{aligned}
$$

where we also use Lemma D.21 and Lemma D.26.

Therefore, we derive the theorem. $\qquad\square$

# E  PROOF FOR SECTION 4

## E.1  PROOF FOR STANDARD TRAINING

**Theorem E.1.** *For sufficiently large d, suppose we train the model using the standard training starting from the random initialization, then after $T = \Theta(\mathrm{poly}(d)/\eta)$ iterations, with high probability over the sampled training dataset $\mathcal{Z}$, the model $\boldsymbol{F}^{(T)}$ satisfies:*

- *Standard training is perfect: for all $(\boldsymbol{X}, y) \in \mathcal{Z}$, all $i \in [k]\backslash\{y\} : F_y^{(T)}(\boldsymbol{X}) > F_i^{(T)}(\boldsymbol{X})$.*

- *Non-robust features are learned: $\mathbb{P}_{(\boldsymbol{X_f}, y) \sim \mathcal{D}_{\mathcal{F}_{NR}}} \left[ \mathrm{argmax}_{i \in [k]} F_i^{(T)}(\boldsymbol{X_f}) \neq y \right] = o(1).$*

- *Standard test accuracy is good: $\mathbb{P}_{(\boldsymbol{X}, y) \sim \mathcal{D}} \left[ \mathrm{argmax}_{i \in [k]} F_i^{(T)}(\boldsymbol{X}) \neq y \right] = o(1).$*

- *Robust test accuracy is bad: for any given data $(\boldsymbol{X}, y)$, using the following perturbation $\boldsymbol{\Delta}(\boldsymbol{X}, y) := (\boldsymbol{\delta}_1, \boldsymbol{\delta}_2, \ldots, \boldsymbol{\delta}_P)$, where $\boldsymbol{\delta}_p := -\beta_p \boldsymbol{v}_y + \epsilon \boldsymbol{v}_{y'}$ for $p \in \mathcal{J}_{NR}$; $\boldsymbol{\delta}_p := \boldsymbol{0}$ for $p \in \mathcal{J}_R$, and $y'$ is randomly chosen from $[k] \setminus \{y\}$ (which does not depend on the model $\boldsymbol{F}^{(T)}$ and is illustrated in Figure 2), we have*

$$\mathbb{P}_{(\boldsymbol{X}, y) \sim \mathcal{D}} \left[ \mathrm{argmax}_{i \in [k]} F_i^{(T)}(\boldsymbol{X} + \boldsymbol{\Delta}(\boldsymbol{X}, y)) \neq y \right] = 1 - o(1).$$

**Proof Idea:** Our proof is divided into the following three steps (the proof approach is almost identical to that of Theorem D.1, with the only difference being that we need to demonstrate that during the standard training process, the noise terms remain small at all times). Except for special mention, the logic and process of proving all lemmas are similar to the simplified case without noise.

### E.1.1  WEIGHT DECOMPOSITION FOR STANDARD TRAINING

**Lemma E.2** (Weight Decomposition for Standard Training). *For any time $t \geq 0$, each neuron $\boldsymbol{w}_{i,r}$ $((i,r) \in [k] \times [m])$, we have*

$$\boldsymbol{w}_{i,r}^{(t)} = \boldsymbol{w}_{i,r}^{(0)} + A_{i,r}^{(t)} \boldsymbol{u}_i + B_{i,r}^{(t)} \boldsymbol{v}_i + \sum_{j \neq i} (C_{i,r,j}^{(t)} \boldsymbol{u}_j + D_{i,r,j}^{(t)} \boldsymbol{v}_j) + \sum_{(\boldsymbol{X}, y) \in \mathcal{Z}} \sum_{p \in [P]} \sigma_{i,r}((\boldsymbol{X}, y), p) \boldsymbol{\xi}_p,$$

*where $A_{i,r}^{(t)}, B_{i,r}^{(t)}, C_{i,r,j}^{(t)}$ and $D_{i,r,j}^{(t)}$ and $\sigma_{i,r}((\boldsymbol{X}, y), p)$ are some time-variant coefficients.*

### E.1.2  NOISE TERMS ARE SMALL

Different from the simplified scenario, we need to prove that the noise terms are always small, which can be presented as the following lemma.

**Lemma E.3** (Noise Correlations are Always Small). *For any time $t = O(\mathrm{poly}(d))$ and each training data point $(\boldsymbol{X}, y) \in \mathcal{Z}$ and for each patch index $p \in [P]$, we have $\langle \boldsymbol{w}_{i,r}^{(t)}, \boldsymbol{\xi}_p \rangle = \tilde{O}(1/\sqrt{d})$.*

*Proof.* By analyzing the iterative process of the noise terms, we have the following lemma:

**Lemma E.4** (Noise Correlation Update). *For every $(\boldsymbol{X}, y) \in \mathcal{Z}$ and $p \in [P]$, if $y = i$ then*

$$\left\langle \boldsymbol{w}_{i,r}^{(t+1)}, \boldsymbol{\xi}_p \right\rangle = \left\langle \boldsymbol{w}_{i,r}^{(t)}, \boldsymbol{\xi}_p \right\rangle + \widetilde{\Theta} \left( \frac{\eta}{N} \right) \widetilde{\mathrm{ReLU}}' \left( \left\langle \boldsymbol{w}_{i,r}^{(t)}, \boldsymbol{x}_p \right\rangle \right) \left( 1 - \mathrm{logit}_i \left( \boldsymbol{F}^{(t)}, \boldsymbol{X} \right) \right) \pm \frac{\eta}{\sqrt{d}}$$

*for similar reason, if $y \neq i$, then*

$$\left\langle \boldsymbol{w}_{i,r}^{(t+1)}, \boldsymbol{\xi}_p \right\rangle = \left\langle \boldsymbol{w}_{i,r}^{(t)}, \boldsymbol{\xi}_p \right\rangle - \widetilde{\Theta} \left( \frac{\eta}{N} \right) \widetilde{\mathrm{ReLU}}' \left( \left\langle \boldsymbol{w}_{i,r}^{(t)}, \boldsymbol{x}_p \right\rangle \right) \mathrm{logit}_i \left( \boldsymbol{F}^{(t)}, \boldsymbol{X} \right) \pm \frac{\eta}{\sqrt{d}}$$

Using the same line of reasoning as in the simplified case (Lemma D.5, Lemma D.6 and Lemma D.8), we can derive the following two lemmas:

**Lemma E.5.** *For each class $i \in [k]$ and all time $t$ such that $\max_{(\boldsymbol{X},y) \in \mathcal{Z}_i} F_y^{(t)}(\boldsymbol{X}) \geq \log(d)$ and $\max_{r \in [m]} A_{i,r}^{(t)} = O(\text{polylog(d)}/\sqrt{d})$ and $\max_{r \in [m]} C_{i,r,y}^{(t)} = O(\text{polylog(d)}/\sqrt{d})$ and $\max_{r \in [m]} D_{i,r,y}^{(t)} = O(\text{polylog(d)}/\sqrt{d})$ for each $i \in [k], y \in [k] \setminus \{i\}$, we have $\min_{(\boldsymbol{X},y) \in \mathcal{Z}_i} F_y^{(t)}(\boldsymbol{X}) = \Omega(\max_{(\boldsymbol{X},y) \in \mathcal{Z}_i} F_y^{(t)}(\boldsymbol{X}))$.*

**Lemma E.6.** *For some time $T_0$, we have $\sum_{t=T_0}^{T} \mathbb{E}_{(\boldsymbol{X},y) \sim \mathcal{Z}} \left[ 1 - \text{logit}_y \left( \boldsymbol{F}^{(t)}, \boldsymbol{X} \right) \right] = \tilde{O}(\eta^{-1})$.*

Combined with Lemma E.4, Lemma E.5 and Lemma E.6, and $N = \text{poly}(d)$, we can prove this lemma. $\qquad\square$

### E.1.3 FEATURE LEARNING FOR STANDARD TRAINING

Theorem E.1 is a direct corollary of the following lemma:

**Lemma E.7.** *For sufficiently large time $T = \Theta(\text{poly}(d))$, we have $\max_{r \in [m]} B_{i,r}^{(t)} = \Theta(\log(d))$ and $\max_{r \in [m]} A_{i,r}^{(t)} = O(\text{polylog(d)}/\sqrt{d})$ and $\max_{r \in [m]} C_{i,r,y}^{(t)} = O(\text{polylog(d)}/\sqrt{d})$ and $\max_{r \in [m]} D_{i,r,y}^{(t)} = O(\text{polylog(d)}/\sqrt{d})$ for each $i \in [k], y \in [k] \setminus \{i\}$.*

*Proof.* Due to Lemma E.3, we can prove this lemma using the exact same logic as that used to prove Lemma D.8. $\qquad\square$

### E.2 PROOF FOR ADVERSARIAL TRAINING

**Theorem E.8.** *For sufficiently large $d$, suppose we train the model using the adversarial training algorithm starting from the random initialization, then after $T = \Theta(\text{poly}(d)/\eta)$ iterations, with high probability over the sampled training dataset $\mathcal{Z}$, the model $\boldsymbol{F}^{(T)}$ satisfies:*

- *Adversarial training is perfect: for all $(\boldsymbol{X}, y) \in \mathcal{Z}$ and all perturbation $\boldsymbol{\Delta}$ satisfying $\|\boldsymbol{\Delta}\|_\infty \leq \epsilon$, all $i \in [k] \setminus \{y\}: F_y^{(T)}(\boldsymbol{X} + \boldsymbol{\Delta}) > F_i^{(T)}(\boldsymbol{X} + \boldsymbol{\Delta})$.*

- *Robust features are learned: $\mathbb{P}_{(\boldsymbol{X_f}, y) \sim \mathcal{D}_{\mathcal{F}_R}} \left[ \text{argmax}_{i \in [k]} F_i^{(T)}(\boldsymbol{X_f}) \neq y \right] = o(1)$.*

- *Robust test accuracy is good:*
$$\mathbb{P}_{(\boldsymbol{X},y) \sim \mathcal{D}} \left[ \exists \boldsymbol{\Delta} \in \left( \mathbb{R}^d \right)^P \text{ s.t. } \|\boldsymbol{\Delta}\|_\infty \leq \epsilon, \text{argmax}_{i \in [k]} F_i^{(T)}(\boldsymbol{X} + \boldsymbol{\Delta}) \neq y \right] = o(1).$$

**Proof Idea:** Our proof approach is almost identical to that of Theorem D.15, with the only difference being that we need to demonstrate that during the adversarial training process, the noise terms remain small at all times.

### E.2.1 WEIGHT DECOMPOSITION FOR ADVERSARIAL TRAINING

**Lemma E.9** (Weight Decomposition for Adversarial Training). *For any time $t \geq 0$, each neuron $\boldsymbol{w}_{i,r}$ $((i,r) \in [k] \times [m])$, we have*

$$\boldsymbol{w}_{i,r}^{(t)} = \boldsymbol{w}_{i,r}^{(0)} + A_{i,r}^{(t)} \boldsymbol{u}_i + B_{i,r}^{(t)} \boldsymbol{v}_i + \sum_{j \neq i} (C_{i,r,j}^{(t)} \boldsymbol{u}_j + D_{i,r,j}^{(t)} \boldsymbol{v}_j) + \sum_{(\boldsymbol{X},y) \in \mathcal{Z}} \sum_{p \in [P]} \sigma_{i,r}((\boldsymbol{X}, y), p) \boldsymbol{\xi}_p,$$

*where $A_{i,r}^{(t)}, B_{i,r}^{(t)}, C_{i,r,j}^{(t)}$ and $D_{i,r,j}^{(t)}$ and $\sigma_{i,r}((\boldsymbol{X}, y), p)$ are some time-variant coefficients.*

### E.2.2 NOISE TERMS ARE SMALL

Similar to standard training, we also need to prove that the noise terms are always small, which can be presented as the following lemma.

**Lemma E.10** (Noise Correlations are Always Small). *For any time $t = O(\text{poly}(d))$ and each training data point $(\boldsymbol{X}, y) \in \mathcal{Z}$ and for each patch index $p \in [P]$, we have $\langle \boldsymbol{w}_{i,r}^{(t)}, \boldsymbol{\xi}_p \rangle = \tilde{O}(1/\sqrt{d})$.*

*Proof.* The proof logic is similar to Lemma E.3. $\qquad\square$

### E.2.3 Feature Learning for Adversarial Training

Theorem E.8 is a direct corollary of the following lemma:

**Lemma E.11.** *For sufficiently large time* $T = \Theta(\text{poly}(d))$ , *we have* $\max_{r \in [m]} A_{i,r}^{(t)} = \Theta(\log(d))$ *and* $\max_{r \in [m]} B_{i,r}^{(t)} = \tilde{O}(1/d^{c_0})$ *and* $\max_{r \in [m]} C_{i,r,y}^{(t)} = O(\text{polylog(d)}/\sqrt{d})$ *and* $\max_{r \in [m]} D_{i,r,y}^{(t)} = O(\text{polylog(d)}/\sqrt{d})$ *for each* $i \in [k], y \in [k] \setminus \{i\}$.

*Proof.* Due to Lemma E.10, we can prove this lemma using the exact same logic as that used to prove Lemma D.26. $\qquad\square$

