# OpenReview forum: "Adversarial Training Can Provably Improve Robustness: Theoretical Analysis of Feature Learning Process Under Structured Data"
_ICLR.cc/2025/Conference — ICLR 2025 Poster_

### Official Review · Reviewer_XDz5 · 2024-10-30

**Soundness:** 3
**Presentation:** 3
**Contribution:** 3
**Rating:** 6
**Confidence:** 4

**Summary:**

This paper studies the adversarial robustness of neural networks theoretically from the perspective of feature learning theory. The analysis is done for a two-layer ReLU convolutional network under constructed data that has sparse robust features and dense non-robust features. Main results are,
* standard training of gradient descent learns non-robust features. Therefore, adversarial perturbations exist along the negative non-robust feature directions.
* adversarial training with one step gradient ascent provably learns the robust features, thus improving the robustness of the network.

Empirical results on MNIST, CIFAR-10 and SVHN demonstrate the above theoretical findings.

**Strengths:**

1. The theoretical analysis is rigorous in the feature learning regime, and the derived results are interesting contributions to the robustness of neural networks, adding to the previous works.
2. The paper is clearly written with a detailed related work section. The discussions on the theoretical results and assumptions are very helpful in understanding the paper.
3. Experiments also support the theoretical findings.

**Weaknesses:**

1. Extending the analysis beyond the two layer smoothed ReLU convolution network to realistic, deeper neural networks seems difficult.
2. The adversarial training is with just one step gradient ascent which is usually not the case in practice.
3. The definition of robust and non-robust features require parallel to the coordinate axis along with orthonormal condition which is restrictive.

**Questions:**

1. From the analysis, there seems to be no tradeoff between standard and robust accuracies in adversarial training contrary to the empirical observations [P1]. Can the analysis show this tradeoff? Perhaps, in theorem 4.4, it might be worthwhile to include what happens to the standard accuracy $P_{(X,y)\sim \mathcal{D}}[\arg\max F_i^{(T)}(X) \ne y]$ and also what happens when $(X_f,y) \sim \mathcal{D}_{NR}$.
2. The adversarial training analysis shows a phase transition where non-robust features are learned in phase 1, and then robust feature learning increases over the non-robust in phase 2. This is also observed in the simulated experiments. However, some works with similar robust and non-robust feature definitions suggest robust features are learned first. For instance, [P2] shows it empirically using the neural tangent kernel, while Proposition 5.1 in [P3] shows it using generalized additive models theoretically. The phase transition is also not really observed in real data and neural networks with activations other than smoothed ReLU. Can the authors comment on why there is a discrepancy in the results? Adding a discussion contrasting to these results would be helpful.
3. The analysis considers smoothed ReLU activation following the feature learning analysis from Allen-Zhu & Li 2023a. Why is this particular choice made? What happens with other activations?
4. Minor comment: in line 179, 'where $u,v$ are the corresponding' - u,v should be boldfaced.

[P1] Tsipras et al. "Robustness May Be at Odds with Accuracy." ICLR 2019.

[P2] Tsilivis and Kempe. "What can the neural tangent kernel tell us about adversarial robustness?." NeurIPS 2022.

[P3] Singh et al. "Robust Feature Inference: A Test-time Defense Strategy using Spectral Projections." TMLR 2024.

---

> ### Author Response · Authors · 2024-11-24
> **Response to Review XDz5 (1/2)**
>
> We sincerely thank the reviewer for the thoughtful review and positive feedback, and for highlighting the strength of our theoretical contributions and the clarity of our writing. We are very glad to address the questions and suggestions raised by the reviewer, which we believe will help further refine our work. Below are our responses to the questions and suggestions raised by the reviewer.
>
> >**[W1]** The adversarial training is with just one step gradient ascent which is usually not the case in practice.
>
> **[E1]** While adversarial training against one-step attacks (such as FGSM) does not produce robust models in practice, we would first like to point out that, under our theoretical framework, adversarial training based on one-step attacks is sufficient to improve network robustness against arbitrary attacks (see details in Theorem 4.4). Furthermore, we believe that, under our synthetic structured data setup, multi-step adversarial training performs similarly to the one-step method.
>
> We also emphasize that analyzing one-step adversarial training for two-layer neural networks is highly challenging due to the inherently non-convex and non-concave nature of the min-max optimization problem in adversarial training. To address this challenge, we developed a novel and highly non-trivial analysis technique to track the dynamics of the training process. To the best of our knowledge, we are the first to provide a refined analysis of the feature learning process in standard one-step adversarial training for non-linear two-layer networks.
>
> Additionally, we empirically verify our theoretical findings on real-world image datasets (MNIST, CIFAR10, and SVHN) using multi-step adversarial training and multi-step attacks. The numerical results are consistent with our theoretical insights derived from analyzing one-step adversarial training in our toy theoretical model.
>
> >**[W2]** The definition of robust and non-robust features require parallel to the coordinate axis along with orthonormal condition which is restrictive.
>
> **[E2]** We would like to clarify that the one-hot assumption is adopted for simplicity in mathematical calculations. Indeed, since our study focuses on $\ell_{\infty}$-robustness, considering only the orthogonality of class signal vectors is insufficient to fully characterize the properties of robustness under the $\ell_{\infty}$ norm. For example, two vectors with equal $\ell_2$ norms can have $\ell_{\infty}$ norms that differ by a factor of $d$, where $d$ is the dimensionality of the vectors. Therefore, it is necessary to explicitly describe the $\ell_{\infty}$ properties of the class signal vectors. For simplicity, we assume them to be one-hot vectors in this context.

---

> ### Author Response · Authors · 2024-11-24
> **Response to Review XDz5 (2/2)**
>
> >**[Q1]** From the analysis, there seems to be no tradeoff between standard and robust accuracies in adversarial training contrary to the empirical observations [P1]. Can the analysis show this tradeoff? Perhaps, in theorem 4.4, it might be worthwhile to include what happens to the standard accuracy $\mathbb{P} _{(\boldsymbol{X},y)\sim \mathcal{D}} [\operatorname{argmax}F_i^{(T)}(\boldsymbol{X})\ne y]$ and also what happens when $(\boldsymbol{X}_f, y)\sim\mathcal{D} _{NR}$.
>
> **[A1]** We would like to clarify that, while there is a tradeoff between standard and robust accuracies in adversarial training, contrary to the empirical observations [1], perfect robust accuracy can imply perfect standard accuracy under our synthetic data setup. We believe that extending our theoretical framework to study the tradeoff between robustness and accuracy represents a valuable and interesting direction for future research.
>
> >**[Q2]** The phase transition is also not really observed in real data and neural networks with activations other than smoothed ReLU. Can the authors comment on why there is a discrepancy in the results? Adding a discussion contrasting to these results would be helpful.
>
> **[A2]** Indeed, the time when phase transition happens is related to the one-step gradient ascent learning rate $\tilde{\eta}$ (roughly speaking, the threshold time is nearly $\Theta(\tilde{\eta}^{-1})$). Thus, in practice, due to the relatively large learning rate (i.e., step size) of PGD for finding adversarial examples (e.g., for CIFAR10, $\tilde{\eta}$ may be chosen as $2/255$, whereas the gradient descent learning rate is typically just $10^{-3}$), the duration of the first stage is very short, which may lead to the non-observation of phase transition in real-world experiments.
>
> >**[Q3]** The analysis considers smoothed ReLU activation following the feature learning analysis from Allen-Zhu & Li 2023a. Why is this particular choice made? What happens with other activations?
>
> **[A3]**  We would like to point out that we use the smoothed ReLU to replace the original ReLU due to the discontinuity of ReLU at zero, which is a common approach in the feature learning theory literature, as seen in [2][3] in the related work section. However, we believe that the theoretical insights derived from analyzing smoothed ReLU networks can also be applied to other non-linear activation functions commonly used in practice, such as the standard ReLU and GeLU activation functions.
>
> >**[Q4]** Minor comment: in line 179, 'where u,v are the corresponding' - u,v should be boldfaced.
>
> **[A4]** Thank the reviewer for pointing out this typo. We have fixed it in the revision of our paper.
>
> **Reference**
>
> [1] Tsipras et al. "Robustness May Be at Odds with Accuracy." ICLR 2019.
>
> [2] Allen-Zhu, Z. and Li, Y. (2023b). Towards understanding ensemble, knowledge distillation and self-distillation in deep learning. In The Eleventh International Conference on Learning Representations.
>
> [3] Chidambaram, M., Wang, X., Wu, C. and Ge, R. (2023). Provably learning diverse features in multi view data with midpoint mixup. In International Conference on Machine Learning. PMLR.

---

> > ### Comment · Reviewer_XDz5 · 2024-11-28
> >
> > Thank you for the response and clarification.

---

> > > ### Author Response · Authors · 2024-11-28
> > > **Response to Review XDz5**
> > >
> > > Thank you very much for your kind review, and we are glad that you enjoyed our paper!

---

### Official Review · Reviewer_cqxX · 2024-11-02

**Soundness:** 2
**Presentation:** 3
**Contribution:** 2
**Rating:** 6
**Confidence:** 4

**Summary:**

The paper proposes a theoretical framework to explain why standard training leads models to learn non-robust features, whereas adversarial training encourages models to focus on robust features. This theoretical insight supports previously observed empirical results in the literature.

To illustrate this, the authors construct data using two distinct sets of features: robust-feature patches and non-robust-feature patches. These two sets are disjoint and collectively span the entire feature space \([P]\) (Definition 2.2), with the additional assumption that all features are orthonormal.

Using this setup and the associated assumptions, the paper demonstrates that a network relying solely on non-robust features becomes vulnerable to adversarial perturbations. Conversely, a network that learns robust features is shown to achieve robustness, as formalized in Propositions 3.1 and 3.2.

Finally, Theorem 4.3 demonstrates that standard training encourages the model to learn non-robust features, achieving low loss on clean data but high loss under adversarial conditions. In contrast, Theorem 4.4 shows that adversarial training shifts the model’s focus toward learning robust features, rather than non-robust ones.

**Strengths:**

- The paper provides a compelling theoretical explanation for a well-known empirical phenomenon, as demonstrated in prior work [1].

[1] Ilyas, Andrew, et al. "Adversarial examples are not bugs, they are features." Advances in neural information processing systems 32 (2019).

**Weaknesses:**

The theoretical framework relies on several significant assumptions:
- **Types of Features:** The paper assumes only two types of features: robust and non-robust, both of which contribute meaningfully to model predictions. However, this simplification may not fully capture the complexity of real data, as observed in [1], which categorizes features into robust/non-robust and useful/non-useful, based on their contribution to the model’s predictions. The set of non-robust features could be infinite, whereas this paper imposes a limit.

- **Orthogonality Assumption:** The core theorem relies on the assumption that features are orthogonal. However, in practical settings, features are not typically orthogonal. The proofs, particularly of Propositions 3.1 and 3.2 (provided in Appendix C), depend on this orthogonal assumption by considering robust features $(\(u_y\))$ orthogonal to non-robust features $(\(v_y\))$.

- **Data Construction Assumption:** Each data point $\(X\)$ is composed of multiple patches $\(\{x_1, \dots, x_P\}\)$, where each patch represents either a robust or non-robust feature (Definition 2.2, Step 3), with certain scaling and randomness. These features are modeled in the input space, but to satisfy the strict assumptions made in the paper, it would seem more appropriate to consider them in the latent space.

- **Definitions of Robust and Non-Robust Features (Assumption 2.3):**
  - Robust features are defined as stronger than non-robust features.
  - Non-robust features are defined as denser than robust features.
  - However, these definitions lack interpretability, which would be beneficial for a more complete understanding.
  - Under this assumption, standard training picks up denser features, while adversarial training picks up stronger features, assumed to be robust features.
  - The paper lacks experiments to demonstrate the existence of these properties (strength and density) in real-world datasets.

**Questions:**

1. In Proposition 3 and Theorem 4.3, what if the adversarial perturbation is designed as $\(\delta_p = -\beta_p v_y + \epsilon v_t\)$, where $\(v_t\)$ is orthogonal to all $\(v_i\)$ for $\(i \in [k]\)$? For example, could an adversarial perturbation be constructed without using any known non-robust features from the set$ \(J_{NR}\)$? Given the high dimensionality of $\(d\)$, and the potential infinitude of non-robust features, this scenario might be feasible.

2. In the definition of “Small Perturbation Radius” (lines 281-284), two data points $\(X\)$ and $\(X'\)$ with distinct labels may have significant differences with high probability. While this may hold for real data, what if $\(X'\)$ is also an adversarial example?

3. In line 285, the authors assume that $\(F\)$ is an accurate classifier on $\((X, y)\)$. However, since $\(F\)$ is only a two-layer neural network, this assumption may not be entirely valid. There could be a substantial portion of the dataset on which $\(F\)$ misclassifies. Can the authors discuss which parts of the theory would be impacted by relaxing this assumption? For instance, in Propositions 3.1 and 3.2, \$(\arg \max F_i(X + \Delta) \neq y\)$ could hold even with $\(\Delta = 0\)$ if $\(F_i(X) \neq y\)$.

---

> ### Author Response · Authors · 2024-11-24
> **Response to Review cqxX**
>
> We sincerely thank the reviewer for the thoughtful feedback, and for highlighting the strength of our theoretical contributions and the clarity of our writing. We are very glad to address the questions and suggestions raised by the reviewer, which we believe will help further refine our work. Below are our responses to the questions and suggestions raised by the reviewer.
>
> >**[Q1]** In Proposition 3 and Theorem 4.3, what if the adversarial perturbation is designed as $\boldsymbol{\delta}_p = -\beta_p \boldsymbol{v}_y + \epsilon \boldsymbol{v}_t$, where $v_t$ is orthogonal to all $v_i$ for $(i∈[k])$? For example, could an adversarial perturbation be constructed without using any known non-robust features from the set (JNR)? Given the high dimensionality of (d), and the potential infinitude of non-robust features, this scenario might be feasible.
>
> **[A1]** We thank the reviewer for the valuable suggestion. We would like to point out that the perturbation proposed by the reviewer is a type of untargeted attack. For instance, assuming the network learns only the non-robust features $\boldsymbol{v}_y$ , the output class prediction of the network would be random due to the orthogonal property and the random initialization of network weights. However, this construction still includes a known non-robust feature $\boldsymbol{v}_y$.
>
> >**[Q2]** In the definition of “Small Perturbation Radius” (lines 281-284), two data points $\boldsymbol{X}$ and $\boldsymbol{X}'$ with distinct labels may have significant differences with high probability. While this may hold for real data, what if $\boldsymbol{X}'$ is also an adversarial example?
>
> **[A2]** We would like to clarify that this data assumption is directly applied to real data. As mentioned in the paper, we emphasize that we require $\boldsymbol{X}$ and $\boldsymbol{X}'$ to both be sampled from the real data distribution $\mathcal{D}$ (in line 282 of our paper), so it does not include cases where $\boldsymbol{X}'$ is an adversarial example. This assumption is consistent with the empirical observation that the typical perturbation radius is often much smaller than the separation distance between different classes [1].
>
> >**[Q3]** In line 285, the authors assume that $\boldsymbol{F}$ is an accurate classifier on $(\boldsymbol{X}, y)$. However, since  $\boldsymbol{F}$ is only a two-layer neural network, this assumption may not be entirely valid. There could be a substantial portion of the dataset on which  $\boldsymbol{F}$ misclassifies. Can the authors discuss which parts of the theory would be impacted by relaxing this assumption? For instance, in Propositions 3.1 and 3.2, $\operatorname{argmax}F_i(\boldsymbol{X} + \boldsymbol{\Delta})$ could hold even with if $F_i(\boldsymbol{X}) \ne  y$ .
>
> **[A3]** We thank the reviewer for the insightful suggestion. In fact, we would like to clarify that this assumption can be relaxed by defining adversarial examples without requiring correct classification on the clean data. All our analysis techniques remain valid under this relaxed assumption. We have revised the assumption in the updated version of our paper. Thank you again for the valuable suggestion.
>
> **Reference**
>
> [1]  Yang, Y.-Y., Rashtchian, C., Zhang, H., Salakhutdinov, R. R. and Chaudhuri, K. (2020). A closer look at accuracy vs. robustness. Advances in neural information processing systems, 33 8588–8601.

---

> > ### Comment · Reviewer_cqxX · 2024-11-25
> > **Further feedback from the reviewer**
> >
> > I thank the authors for their efforts in addressing my concerns. While most of them have been satisfactorily resolved, I remain somewhat concerned about the fundamental assumption of the paper. The paper assumes only two types of features: robust and non-robust, both of which are considered meaningful to model predictions. However, this simplification may not fully capture the complexity of real-world data. As highlighted in [1], features can be categorized into robust/non-robust and useful/non-useful, depending on their contribution to the model’s predictions. Importantly, the set of non-robust features could theoretically be infinite, whereas this paper imposes a limitation.
> >
> > In particular, regarding Question 1, I inquired about scenarios where the model learns from useful but non-robust features—such as predicting a cow based on the grass in the background—and I would appreciate further clarification or discussion from the authors on this point.
> >
> > That said, I appreciate the overall idea and flow of the paper.
> >
> > Thank you.

---

> > > ### Author Response · Authors · 2024-11-25
> > > **Response to Further Feedback from the Reviewer cqxX**
> > >
> > > Dear reviewer cqxX,
> > >
> > > Thank you very much for your feedback! We acknowledge, as highlighted in [1], that features can be categorized as robust/non-robust and useful/non-useful, depending on their contribution to the model's predictions. The set of non-robust features could theoretically be infinite. In our paper, we simplify the data setup by assuming that both robust and non-robust features are useful, focusing specifically on cases where non-robust features are finite.
> > >
> > > While there remains a gap between our synthetic structured data assumption and real-world data, as observed in [1], we emphasize that we are the first to consider a realistic image-like data model. Indeed, as mentioned in the related work section (lines 117–130), several previous theoretical studies [2][3][4] have analyzed adversarial robustness with linear classifiers using a mixture-of-Gaussians data setup. However, standard training does not explicitly converge to non-robust solutions under these conditions. This result contrasts with empirical findings, where networks trained using standard methods exhibit poor robustness performance (e.g., [5][6][7]). For instance, as highlighted in [4], adversarial training, similar to standard training, directionally converges to the maximum $\ell_2$-margin solution when applied to a Gaussian-mixture data model with $\ell_2$ perturbations. This suggests that under their assumptions, standard training alone can achieve adversarial robustness due to the maximum-margin implicit bias, even though neural networks trained with standard methods generally lack robustness in practice.
> > >
> > > To the best of our knowledge, we are the first to attempt to understand adversarial training under a realistic assumption. We also want to point out that considering more realistic data assumptions—where non-robust features are infinite, as observed in [1]—is an important and interesting direction for future research.
> > >
> > > Thanks for the valuable feedback again, and for highlighting our overall idea and flow of the paper.
> > >
> > > **Reference**
> > >
> > > [1] Ilyas, Andrew, et al. "Adversarial examples are not bugs, they are features." Advances in neural information processing systems 32 (2019).
> > >
> > > [2] Li, Y., X.Fang, E., Xu, H. and Zhao, T. (2020). Implicit bias of gradient descent based adversarial training on separable data. In International Conference on Learning Representations.
> > >
> > > [3] Javanmard, A. and Soltanolkotabi, M. (2022). Precise statistical analysis of classification accuracies for adversarial training. The Annals of Statistics, 50 2127–2156.
> > >
> > > [4]  Chen, J., Cao, Y. and Gu, Q. (2023). Benign overfitting in adversarially robust linear classification. In Uncertainty in Artificial Intelligence. PMLR.
> > >
> > > [5] Biggio, B., Corona, I., Maiorca, D., Nelson, B., Šrndi´ c, N., Laskov, P., Giacinto, G. and Roli, F. (2013). Evasion attacks against machine learning at test time. In Machine Learning and Knowledge Discovery in Databases: European Conference, ECML PKDD 2013, Prague, Czech Republic, September 23-27, 2013, Proceedings, Part III 13. Springer.
> > >
> > > [6] Szegedy, C., Zaremba, W., Sutskever, I., Bruna, J., Erhan, D., Goodfellow, I. and Fergus, R. (2013). Intriguing properties of neural networks. arXiv preprint arXiv:1312.6199.
> > >
> > > [7] Goodfellow, I. J., Shlens, J. and Szegedy, C. (2014). Explaining and harnessing adversarial examples. arXiv preprint arXiv:1412.6572.

---

> > > > ### Comment · Reviewer_cqxX · 2024-11-26
> > > > **Further feedback**
> > > >
> > > > I thank the authors for their response. While I agree that the proposed data structure is an improvement over the mixture-of-Gaussian data setup, I find it still not entirely realistic.
> > > >
> > > > More specifically, as highlighted in my previous comment:
> > > >
> > > > > - **Data Construction Assumption:** Each data point \(X\) is composed of multiple patches \(\{x_1, \dots, x_P\}\), where each patch represents either a robust or non-robust feature (Definition 2.2, Step 3), with certain scaling and randomness. These features are modeled in the input space, but to satisfy the strict assumptions made in the paper, it would seem more appropriate to consider them in the latent space.
> > > >
> > > > Since the data patches are defined in the input space, and each patch is constructed from robust and non-robust features, I am unclear on how the authors envision a robust feature such as the shape of an object (e.g., a cow) being effectively represented. The shape of an object can vary significantly, suggesting that the number of robust features required to represent such a shape could be extremely large. In my view, it would make more sense to define these features (robust and non-robust) in the latent space, where the dimensionality is much lower and representations are more abstract.
> > > >
> > > > Additionally, as noted in another weakness section, the paper assumes that *"Robust features are defined as stronger than non-robust features, and non-robust features are defined as denser than robust features."* However, it is unclear how these properties (strength and density) are interpreted when features are in the input space. For example, how can a robust feature exhibit stronger intensity than a non-robust feature in the input space? Furthermore, the paper lacks empirical validation to demonstrate the existence of these properties in real-world datasets.
> > > >
> > > > That said, despite these remaining concerns, I find the theoretical framework and data construction interesting and believe they have the potential to inspire follow-up works. I also appreciate the authors’ effort in addressing my concerns. For these reasons, I would like to increase my score and lean toward supporting this work.
> > > >
> > > > Thank you.

---

> > > > > ### Author Response · Authors · 2024-11-26
> > > > > **Response to Further Feedback from the Reviewer cqxX**
> > > > >
> > > > > Thank you very much for your kind review, and we are glad that you enjoyed our paper!

---

### Official Review · Reviewer_gMkj · 2024-11-03

**Soundness:** 4
**Presentation:** 3
**Contribution:** 3
**Rating:** 6
**Confidence:** 3

**Summary:**

This work centers on adversarial robustness—the capacity of machine learning models to be resistant to so-called adversarial examples. Its goal is to provide an optimization-focused perspective on the effectiveness of adversarial training (currently the most widely used approach to achieve adversarial robustness) and contrasts it to standard training methods that do not provide this kind of resilience.

More precisely, the paper sets up a stylized data distribution setting (building on settings proposed in the past) and subject to it the authors rigorously prove that a simple deep learning model (two-layer convnet) becomes adversarially robust when trained using adversarial training, but becomes (demonstrably) not adversarially robust if trained using the standard training.

**Strengths:**

+ Bringing a new perspective on a very core and extremely well studied topic
+ Establishing the result required a major technical effort and delivered some interesting insights into underlying training dynamics of two-layer convnets

**Weaknesses:**

- The considered data distribution is quite interesting and has a clear motivations but at the end of the day it is really heavily stylized, especially the choice to make the class signal be essentially a (scaled) one-hot vector
- It is unclear what are the take-aways from this results (beyond the interesting insights into the training dynamics of the two-layer network), given how heavily stylized the setting is and that the outcomes are fully in line with what one would expect based on the prior work

**Questions:**

How would you motivate the assumption that the class signal vectors are one-hot? How important it is from a technical point of view?

Overall, this is a clear and well-executed result, so in the end it is about weighing the pros against cons. I personally am positive about this result.

---

> ### Author Response · Authors · 2024-11-24
> **Response to Review gMkj**
>
> We sincerely thank the reviewer for the encouraging and insightful feedback, and for highlighting the strength of our theoretical contributions and the clarity of our writing. We are very glad to address the question raised by the reviewer, which we believe will help further refine our work. Below is our response to the question raised by the reviewer.
>
> >**[Q]** How would you motivate the assumption that the class signal vectors are one-hot? How important it is from a technical point of view?
>
> **[A]**  We would like to clarify that the one-hot assumption is adopted for simplicity in mathematical calculations. Indeed, since our study focuses on $\ell_{\infty}$-robustness, considering only the orthogonality of class signal vectors is insufficient to fully characterize the properties of robustness under the $\ell_{\infty}$ norm. For example, two vectors with equal $\ell_2$ norms can have $\ell_{\infty}$ norms that differ by a factor of $d$, where $d$ is the dimensionality of the vectors. Therefore, it is necessary to explicitly describe the $\ell_{\infty}$ properties of the class signal vectors. For simplicity, we assume them to be one-hot vectors in this context.

---

> > ### Comment · Reviewer_gMkj · 2024-11-26
> >
> > Ok, this makes sense. Thank you for the answer.

---

> > > ### Author Response · Authors · 2024-11-26
> > > **Response to Review gMkj**
> > >
> > > Thank you very much for your kind review, and we are glad that you enjoyed our paper!

---

### Official Review · Reviewer_UbB1 · 2024-11-06

**Soundness:** 3
**Presentation:** 4
**Contribution:** 3
**Rating:** 6
**Confidence:** 3

**Summary:**

The paper presented provides a theoretical framework to understand adversarial examples and the mechanisms by which adversarial training improves model robustness. The authors focus on a two-layer smoothed ReLU convolutional neural network trained on structured data composed of robust and non-robust features. The goal is to demonstrate how standard training methods tend to favor non-robust features, causing adversarial vulnerability, and how adversarial training can shift this learning towards more robust features.

**Strengths:**

+ The paper proves rigorous theoretical results that elucidate the dynamics of feature learning under adversarial conditions. The clear distinction between robust and non-robust features and their impact on network vulnerability is particularly insightful.

+ The paper is well-structured and easy to follow. The assumptions, theoretical statements, and proof techniques are clearly presented.

+ The theoretical findings are well-supported by experiments on real-image datasets like MNIST, CIFAR10, and SVHN, enhancing the credibility of the results.

**Weaknesses:**

- The complexity of the models and the heavy reliance on specific assumptions about data structure may limit the general applicability of the results.

**Questions:**

Overall, I enjoyed reading the paper. The paper provides a viable theoretical framework for characterizing the feature learning process of standard and adversarial training of two-layer neural networks. I like the level of formalism presented in this paper, which clarifies the theoretical settings and main theoretical results precisely. I think the theoretical framework and the proof techniques can potentially be an important step to eventually understanding the internal mechanism of adversarial learning.

My main questions relate to the assumptions used to derive the theoretical results. The paper assumes the activation is smoothed ReLU and considers one-step gradient descent for generating adversarial examples. These assumptions are different from common practice in adversarial training. Can the authors clarify whether their theoretical framework can be applied to standard ReLU-activated neural nets and multi-step PGD? If this is difficult, it would be helpful to explain the bottleneck for extending the theoretical results to more typical settings. In addition, I did not fully understand why the patch-structured data model (Definition 2.2 and Assumption 2.3) is more realistic than the data model (i.e., the Gaussian mixture model or the multi-cluster data model) assumed in previous works. It is hard to tell from my perspective which data model is more realistic since they are all synthetic data distributions. Can the authors provide some explanations or references?

A few specific questions are listed below:

1. The relationship between robust/non-robust features and the input data is sort of linear, up to some noise. Is it possible to consider a non-linear relationship (e.g., using a ground-truth feature extractor to map the input space to some latent space)?

2. The robust/non-robust features are assumed to be well-separated with respect to particular input dimensions. Generally speaking, there could be cases where robust and non-robust features overlap with each other within the input, and for each input, the robust/non-robust feature dimensions may vary. Is it possible to use the proposed theoretical framework to analyze the feature learning process for such a more generalized setting?

---

> ### Author Response · Authors · 2024-11-24
> **Response to Review UbB1 (1/2)**
>
> We sincerely thank the reviewer for the positive support and valuable feedback! We greatly appreciate the insightful review, and the recognition of highlighting the significance of our contribution and solidity of our theory, as well as the clarity of our writing. We are very glad to address the questions and suggestions raised by the reviewer, which we believe will help further refine our work. Below are our responses to the questions and suggestions raised by the reviewer.
>
> >**[Q1]** Can the authors clarify whether their theoretical framework can be applied to standard ReLU-activated neural nets and multi-step PGD?
>
> **[A1]**  We would first like to clarify the reasons for our choice of smoothed ReLU and one-step PGD in our theoretical model. Specifically, we use the smoothed ReLU to replace the original ReLU due to the discontinuous property of ReLU at zero, which is a common approach in the literature of feature learning theory papers, such as [1][2] in the related work section.
>
> Additionally, we point out that, under our theoretical framework, adversarial training based on one-step attacks is sufficient to improve network robustness against arbitrary attacks (see details in Theorem 4.4). We also emphasize that analyzing one-step adversarial training for two-layer neural networks is highly challenging due to the inherently non-convex and non-concave nature of the min-max optimization problem in adversarial training. To address this, we developed a novel and highly non-trivial analysis technique to track the dynamics of the training process.
>
> We believe that extending our results to standard ReLU-activated neural networks and multi-step PGD is an important and interesting future direction for our work.
>
> >**[Q2]** It is hard to tell from my perspective which data model is more realistic since they are all synthetic data distributions. Can the authors provide some explanations or references?
>
> **[A2]**  As mentioned in the related work section (lines 117–130), we note that several previous theoretical studies [3][4][5] have considered the mixture-of-Gaussian data setup to analyze adversarial robustness with linear classifiers. However, standard training does not explicitly converge to non-robust solutions under these conditions. This result contrasts with empirical findings, where networks trained using standard methods exhibit poor robustness performance (e.g., [6][7][8]). For instance, as highlighted in [5], adversarial training, similar to standard training, also directionally converges to the maximum $\ell_2$-margin solution when applied to a Gaussian-mixture data model with $\ell_2$ perturbations. This implies that, under their assumptions, standard training alone can achieve adversarial robustness due to the maximum-margin implicit bias, even though neural networks trained with standard methods generally lack robustness in practice.
>
> In this paper, we aim to bridge the gap between theory and practice by adopting a more structured data assumption and employing a non-linear two-layer CNN as the learner. This setup ensures the existence of both robust and non-robust global minima due to the non-linearity of the data model and the non-convexity of the learner model (see the detailed discussion in Section 3). Specifically, we follow the patch-structured data framework proposed in [2], providing a synthetic data setup based on the empirical observations of [9]. Our Assumption 2.3 can be linked to a downsized version of convolutional networks applied to image classification data. With a small kernel size, high-magnitude features that are easily perceivable by humans in an image typically appear only in a few patches (e.g., the ears of a cat or the nose of an elephant), while most other patches resemble random noise to human observers (e.g., the textures of cats and elephants blending into a random background). Illustrations of real images and our patch data are provided in Figures 1 and 2.

---

> ### Author Response · Authors · 2024-11-24
> **Response to Review UbB1 (2/2)**
>
> >**[Q3]** The relationship between robust/non-robust features and the input data is sort of linear, up to some noise. Is it possible to consider a non-linear relationship (e.g., using a ground-truth feature extractor to map the input space to some latent space)?
>
> **[A3]** We thank the reviewer for the thoughtful suggestion. Indeed, due to the non-linearity of the smoothed ReLU activation function, the one-step adversarial training method results in a highly non-convex and non-concave min-max optimization problem. To address this, we developed a novel and highly non-trivial analysis technique to track the dynamics of the training process. Adding a non-linear feature extractor to our data structure would make the analysis of the feature learning process even more challenging, which we believe represents an important and interesting direction for future research.
>
> >**[Q4]** The robust/non-robust features are assumed to be well-separated with respect to particular input dimensions. Generally speaking, there could be cases where robust and non-robust features overlap with each other within the input, and for each input, the robust/non-robust feature dimensions may vary. Is it possible to use the proposed theoretical framework to analyze the feature learning process for such a more generalized setting?
>
> **[A4]** We thank the reviewer for the insightful suggestion. We would like to point out that our theoretical framework for analyzing the feature learning process can be applied to the more generalized setting mentioned by the reviewer. Indeed, we can consider the case where robust and non-robust features overlap with each other within the input patch. Formally, the data input can be written as $\boldsymbol{X} = (\boldsymbol{x}_1, \boldsymbol{x}_2, \dots, \boldsymbol{x}_P)$ and $\boldsymbol{x}_p = \alpha_p \boldsymbol{u}_y + \beta_p \boldsymbol{v} +  \sum _{ \boldsymbol{f} \in \mathcal{F}\setminus  \{\boldsymbol{u}_y, \boldsymbol{v}_y\}  }  \gamma _{p, f}\boldsymbol{f}  + \boldsymbol{\xi}_p$, where the first two terms are class signals, the third term is feature noise, and the last term is random noise. Then, the modified Assumption 2.3 can be presented as follows:
>
> 1. There exists at least one robust patch $p$ such that for any arbitrary patch $p’$ (possibly $p’=p$), we have $\alpha_p \gg \beta_{p’}$.
> 2. There exists a constant $\gamma \geq 0$ such that $\sum _{p \in [P]} \alpha_p^{\gamma} \ll \sum _{p \in [P]} \beta_p^{\gamma}$.
> 3. For each feature noise, the coefficient satisfies $\gamma _{p,f} = \Theta(\sigma_n / \sqrt{d})$, where random noise $\boldsymbol{\xi}_p \sim \mathcal{N}(\boldsymbol{0}, \sigma_n^2 \mathcal{I}_d)$.
>
> Under our generalized data setup and Assumption 2.3, we can apply a similar analysis technique to prove results analogous to Theorem 4.3 and Theorem 4.4.
>
> **Reference**
>
> [1] Chidambaram, M., Wang, X., Wu, C. and Ge, R. (2023). Provably learning diverse features in multi view data with midpoint mixup. In International Conference on Machine Learning. PMLR.
>
> [2] Allen-Zhu, Z. and Li, Y. (2023b). Towards understanding ensemble, knowledge distillation and self-distillation in deep learning. In The Eleventh International Conference on Learning Representations.
>
> [3] Li, Y., X.Fang, E., Xu, H. and Zhao, T. (2020). Implicit bias of gradient descent based adversarial training on separable data. In International Conference on Learning Representations.
>
> [4] Javanmard, A. and Soltanolkotabi, M. (2022). Precise statistical analysis of classification accuracies for adversarial training. The Annals of Statistics, 50 2127–2156.
>
> [5]  Chen, J., Cao, Y. and Gu, Q. (2023). Benign overfitting in adversarially robust linear classification. In Uncertainty in Artificial Intelligence. PMLR.
>
> [6] Biggio, B., Corona, I., Maiorca, D., Nelson, B., Šrndi´ c, N., Laskov, P., Giacinto, G. and Roli, F. (2013). Evasion attacks against machine learning at test time. In Machine Learning and Knowledge Discovery in Databases: European Conference, ECML PKDD 2013, Prague, Czech Republic, September 23-27, 2013, Proceedings, Part III 13. Springer.
>
> [7] Szegedy, C., Zaremba, W., Sutskever, I., Bruna, J., Erhan, D., Goodfellow, I. and Fergus, R. (2013). Intriguing properties of neural networks. arXiv preprint arXiv:1312.6199.
>
> [8] Goodfellow, I. J., Shlens, J. and Szegedy, C. (2014). Explaining and harnessing adversarial examples. arXiv preprint arXiv:1412.6572.
>
> [9] Ilyas, A., Santurkar, S., Tsipras, D., Engstrom, L., Tran, B. and Madry, A. (2019). Adversarial examples are not bugs, they are features. Advances in neural information processing systems, 32.

---

> > ### Comment · Reviewer_UbB1 · 2024-11-28
> >
> > Thanks for the clarifications. I keep my positive view of this paper and believe the theoretical analysis tools introduced by this paper are a valuable contribution to the field.

---

> > > ### Author Response · Authors · 2024-11-29
> > > **Response to Review UbB1**
> > >
> > > Thank you very much for your kind review, and we are glad that you enjoyed our paper!

---

### Official Review · Reviewer_wbRQ · 2024-11-10

**Soundness:** 3
**Presentation:** 3
**Contribution:** 2
**Rating:** 6
**Confidence:** 2

**Summary:**

This paper conducts theoretical analyses on how different the learned features are by normal training and adversarial training. The authors use two layer neural networks with smoothed ReLU as the model, they show that normal training tend to extract non-robust features, while adversarial training encourages the model to learn robust features. Observations in numerical experiments are consistent with theoretical analyses.

**Strengths:**

This paper has the following strengths and contributions:

1. This work theoretically investigates how different adversarial training learns features compared with normal training. Although the results are based on a simple two-layer neural network, it is still meaningful and beneficial for the community as it may be the basis for extension to general neural architectures.

2. The theoretical analyses are consistent with numerical experiments on various datasets.

**Weaknesses:**

I have the following major questions or concerns regarding the current manuscript:

1. The title is a bit misleading. Usually, 'provable robustness' means 'robustness verification', the conclusion of this work does not involve robustness verification.

2. The conclusions of theoretical analyses should involve the magnitude of perturbation $\epsilon$. As we can see when $\epsilon = 0$, adversarial training will be degraded to normal training. Therefore, I believe adversarial training can learn a considerable amount of "non-robust features" when $\epsilon$ is small. A more comprehensive theoretical contribution should be a spectrum instead of discussing two distinct cases: the model learns non-robust features in normal training, as the value of $\epsilon$ increases, the ratio of non-robust features v.s. robust features decreases and in the end, the model will primarily focus on robust features. Can the authors extend your results in this manner? In the current manuscript, Lemma 5.2 cannot degrade to Lemma 5.1 by setting $\epsilon = 0$.

3. The authors assume to use one-step attack in adversarial training (Remark 2.5). However, adversarial training against one-step attacks (like FGSM) does not actually produce robust models. The models trained in this way may be easily broken by a stronger attack.

Based on the weakness above, as well as the questions pointed out in the next section, I think the current manuscript needs a comprehensive edit before publication. I acknowledge the contribution made in this paper, but it still needs improvement. I encourage the authors to address my concerns in the rebuttal.

**Questions:**

My major questions are already pointed out in the "weakness" section above, in addition to that, I have the following minor questions:

1. In line 247, the author does not provide any intuition or reference about the design of the smoothed ReLU. Why don't the authors use softplus or GeLU? They are popular smooth alternatives of ReLU. In addition, the function proposed by authors here seems discontinuous at the point $z = \rho$?

2. In line 281, why $\epsilon = \Theta(\sigma_n \sqrt{d})$? In practice, we typically use smaller $\epsilon$ for $l_\infty$ bounded perturbations when the dimension $d$ is larger. For example, we usually use $\epsilon = 0.3$ for MNIST, $\epsilon = 8/255$ for CIFAR10, and $\epsilon=2/255$ for ImageNet, these are the datasets with increasing input dimensions.

>>> After rebuttal, I appreciate the author's explanations, I will not object to the acceptance of this paper. However, the author should clearly explain the effect of $\epsilon$, as a continuous variable, on feature learning in adversarial training.

---

> ### Author Response · Authors · 2024-11-24
> **Response to Review wbRQ (1/2)**
>
> We sincerely thank the reviewer for the thoughtful feedback, and for highlighting our solid theoretical contributions and the clarity of our writing. We are very glad to address the questions and suggestions raised by the reviewer, which we believe will help further refine our work. Below are our responses to the questions and suggestions raised by the reviewer.
>
> >**[W1]** The title is a bit misleading. Usually, 'provable robustness' means 'robustness verification', the conclusion of this work does not involve robustness verification.
>
> **[E1]** We would like to clarify that we did not claim anything about “provable robustness” or “robustness verification” in our paper. Specifically, the title of our paper indicates that, under our patch-structured data assumption, we rigorously prove (i.e., provably demonstrate) that adversarial training ensures the network learns robust features, thereby improving model robustness. The term “provably” is widely used in the literature of learning theory, as seen in [1][2][3] in the related work section, where it uniformly refers to the rigorous mathematical demonstration of certain important properties of an algorithm.
>
> >**[W2]** The conclusions of theoretical analyses should involve the magnitude of perturbation $\epsilon$. As we can see when $\epsilon = 0$, adversarial training will be degraded to normal training. Can the authors extend your results in this manner? In the current manuscript, Lemma 5.2 cannot degrade to Lemma 5.1 by setting $\epsilon = 0$.
>
> **[E2]** We would like to clarify that Lemma 5.2 can exactly reduce to Lemma 5.1 by setting $\epsilon = 0$ in the current manuscript. Notably, Lemma 5.1 and Lemma 5.2 apply to different time ranges: Lemma 5.1 is valid for any training time $t \geq 0$, whereas Lemma 5.2 focuses specifically on the early stage (the polynomial part) of adversarial training, i.e., it applies for time $0 \leq t \leq T_0$. Within the overlapping time range of Lemma 5.1 and Lemma 5.2, Lemma 5.2 reduces to Lemma 5.1 when $\epsilon = 0$, as the smoothed ReLU lies in the polynomial regime and $1 - \operatorname{logit} = \Theta(1)$ (see details in Lemma D.5 and Lemma D.19).
>
> >**[W3]** The authors assume to use one-step attack in adversarial training (Remark 2.5). However, adversarial training against one-step attacks (like FGSM) does not actually produce robust models. The models trained in this way may be easily broken by a stronger attack.
>
> **[E3]** While adversarial training against one-step attacks (such as FGSM) does not produce robust models in practice, we would first like to point out that, under our theoretical framework, adversarial training based on one-step attacks is sufficient to improve network robustness against arbitrary attacks (see details in Theorem 4.4). Furthermore, we believe that, under our synthetic structured data setup, multi-step adversarial training performs similarly to the one-step method.
>
> We also emphasize that analyzing one-step adversarial training for two-layer neural networks is highly challenging due to the inherently non-convex and non-concave nature of the min-max optimization problem in adversarial training. To address this, we developed a novel and highly non-trivial analysis technique to track the dynamics of the training process. To the best of our knowledge, we are the first to provide a refined analysis of the feature learning process in standard one-step adversarial training for non-linear two-layer networks.
>
> Additionally, we empirically verify our theoretical findings on real-world image datasets (MNIST, CIFAR10, and SVHN) using multi-step adversarial training and multi-step attacks. The numerical results align with our theoretical insights derived from analyzing one-step adversarial training in our toy theoretical model.

---

> ### Author Response · Authors · 2024-11-24
> **Response to Review wbRQ (2/2)**
>
> >**[Q1]** In line 247, the author does not provide any intuition or reference about the design of the smoothed ReLU. Why don't the authors use softplus or GeLU? They are popular smooth alternatives of ReLU. In addition, the function proposed by authors here seems discontinuous at the point $z=\rho$?
>
> **[A1]** We would like to clarify that the smoothed ReLU is continuous at the point $ z = \rho $, as we can calculate the values and derivatives of two parts at $ z = \rho $:
>  1. At the polynomial part, $ \operatorname{ReLU}(z) = \rho / q $, $ \operatorname{ReLU}'(z) = z^{q-1} / \rho^{q-1} = 1 $ when $ z \to \rho^- $.
>  1. At the linear part, $ \operatorname{ReLU}(z) = \rho / q $, $ \operatorname{ReLU}'(z) = 1 $ when $ z \to \rho^+ $.
>
> Here, we use the smoothed ReLU to replace the original ReLU due to the discontinuous property of ReLU at zero, which is widely applied in the literature of feature learning theory papers, such as [3][4] in the related work section.
>
> >**[Q2]** In line 281, why $\epsilon = \Theta(\sigma_n \sqrt{d})$? In practice, we typically use smaller ϵ for l∞ bounded perturbations when the dimension d is larger. For example, we usually use $\epsilon = 0.3$ for MNIST, $\epsilon = 8/255$ for CIFAR10, and $\epsilon = 2/255$ for ImageNet, these are the datasets with increasing input dimensions.
>
> **[A2]** We would like to point out that, under our theoretical framework, we use a relatively small perturbation, even though the perturbation is a function of the data dimension $d$. As mentioned in lines 281–284 of our paper, under our setting, for two data points $(\boldsymbol{X}, y), (\boldsymbol{X}', y') \sim \mathcal{D}$ with distinct labels $y \ne y' \in [k]$, it can be checked that w.h.p. $\|\|\boldsymbol{X}-\boldsymbol{X}'\|\|_{\infty} \geq \Omega(\mathbb{E}[\alpha_p]) \gg \Theta(\sigma_n \sqrt{d}) = \Theta(\epsilon)$, which is consistent with the empirical observation that the typical perturbation radius is often much smaller than the separation distance between different classes [5].
>
> **Reference**
>
> [1]  Du, S. S., Zhai, X., Poczos, B. and Singh, A. (2019b). Gradient descent provably optimizes over parameterized neural networks. In International Conference on Learning Representations.
>
> [2]  Jelassi, S., Sander, M. and Li, Y. (2022). Vision transformers provably learn spatial structure. Advances in Neural Information Processing Systems, 35 37822–37836.
>
> [3]  Chidambaram, M., Wang, X., Wu, C. and Ge, R. (2023). Provably learning diverse features in multi view data with midpoint mixup. In International Conference on Machine Learning. PMLR.
>
> [4]  Allen-Zhu, Z. and Li, Y. (2023b). Towards understanding ensemble, knowledge distillation and self-distillation in deep learning. In The Eleventh International Conference on Learning Representations.
>
> [5] Yang, Y.-Y., Rashtchian, C., Zhang, H., Salakhutdinov, R. R. and Chaudhuri, K. (2020). A closer look at accuracy vs. robustness. Advances in neural information processing systems, 33 8588–8601.

---

> > ### Comment · Reviewer_wbRQ · 2024-11-25
> > **Update**
> >
> > I thank the authors for their responses, which address some of my concerns and questions. I have the following questions and would like to discuss them with the authors.
> >
> > 1. I still don't understand and don't think why Lemma 5.2 degrades to Lemma 5.1 when $\epsilon = 0$. Any more detailed clarifications?
> >
> > 2. I still don't think $\epsilon = \Theta(\sigma_n \sqrt{d})$ is a good assumption, because $\sigma_n$ can be generally considered as a constant, $\epsilon$ will become large in the high dimensionality cases. By contrast $|X - X'|_\infty$ is bounded by $1$, so $|X - X'|_\infty >> \Theta(\sigma_n \sqrt{d})$ will generally not hold unless $\sigma_n = 0$ or dynamically adapt to the dimensionality.

---

> > > ### Author Response · Authors · 2024-11-25
> > > **Response to Update from the Reviewer wbRQ**
> > >
> > > Dear reviewer wbRQ,
> > >
> > > Thanks very much for your update. We will address the two questions raised by the review as below.
> > >
> > > > **[Q1]** I still don't understand and don't think why Lemma 5.2 degrades to Lemma 5.1 when $\epsilon = 0$. Any more detailed clarifications?
> > >
> > > **[A1]** We would like to clarify the detailed derivation to show that, under the early stage of training dynamics, Lemma 5.2 exactly reduces to Lemma 5.1 as follows.
> > >
> > > 1. First, for Lemma 5.1, we consider the early stage where the smoothed ReLU activation lies on the polynomial part. Indeed, we have $\tilde{\operatorname{ReLU}}’(z) = z^{q-1} / \rho^{q-1}$, where $z = \alpha_p A_{i,r}^{(t)}$. By applying Lemma D.5 and Lemma D.6, we know that $1 - \operatorname{Logit}_i (\boldsymbol{F}, \boldsymbol{X}) \approx \Theta(1)$.
> > >
> > >    Combining the two results above, we have that $A_{i,r}^{(t+1)} \approx A_{i,r}^{(t)} + \Theta(\eta)(A_{i,r}^{(t)})^{q-1} \mathbb{E}[\alpha_p^{q}]$ (Lemma 5.1 at the early stage, $B_{i,r}^{(t)}$ is similar).
> > >
> > > 2. Then, for Lemma 5.2, we consider the case when $\epsilon = 0$, which implies that $\operatorname{min}(\frac{\epsilon}{\alpha_p}, \tilde{\Theta}(\eta) \sum_{s \in [m]} (A_{i,s}^{(t)})^q) = 0$. Thus, Lemma 5.2 shows that $A_{i,r}^{(t+1)} \approx A_{i,r}^{(t)} + \Theta(\eta)(A_{i,r}^{(t)})^{q-1} \mathbb{E}[\alpha_p^{q}]$, which is the same as in Lemma 5.1.
> > >
> > > > **[Q2]** I still don't think $\epsilon = \Theta(\sigma_n \sqrt{d})$ is a good assumption, because $\sigma_n$ can be generally considered as a constant, $\epsilon$ will become large in high-dimensional cases.
> > >
> > > **[A2]** Indeed, as we mentioned in Assumption B.3 (see Appendix B), we choose $\sigma_n = \frac{1}{\sqrt{d}}$, which implies that $\epsilon$ will be a constant when the dimension $d$ increases. This small random noise assumption is widely used in the literature of feature learning theory papers, such as [1][2][3][4].
> > >
> > > Thanks for the positive feedback again, and for highlighting our overall idea and flow of the paper.
> > >
> > > **Reference**
> > >
> > > [1] Allen-Zhu, Z. and Li, Y. (2023b). Towards understanding ensemble, knowledge distillation and self-distillation in deep learning. In The Eleventh International Conference on Learning Representations.
> > >
> > > [2] Chidambaram, M., Wang, X., Wu, C. and Ge, R. (2023). Provably learning diverse features in multi view data with midpoint mixup. In International Conference on Machine Learning. PMLR.
> > >
> > > [3] Chen, Z., Deng, Y., Wu, Y., Gu, Q., & Li, Y. (2022). Towards understanding mixture of experts in deep learning. arXiv preprint arXiv:2208.02813.
> > >
> > > [4] Zou, D., Cao, Y., Li, Y., & Gu, Q. (2023, July). The benefits of mixup for feature learning. In International Conference on Machine Learning (pp. 43423-43479). PMLR.

---

> > > ### Author Response · Authors · 2024-12-02
> > > **Looking Forward to the Reviewer's Reply**
> > >
> > > Dear reviewer wbRQ,
> > >
> > > As the deadline approaches, we kindly ask if our responses have adequately addressed your concerns. We would greatly appreciate your feedback to ensure we have fully resolved any outstanding issues. Thank you for your time and consideration.

---

> > > > ### Comment · Reviewer_wbRQ · 2024-12-03
> > > > **Update**
> > > >
> > > > Dear authors,
> > > >
> > > > Thanks for your response. After checking the literatures mentioned, my second question is addressed. However, why does the smoothed ReLU function exhibit the polynomial part in the early stage of training? In addition, I believe a comprehensive investigate on the effect of $\epsilon$ on feature learning can make the manuscript stronger. Thanks!

---

> > > > > ### Author Response · Authors · 2024-12-03
> > > > > **Looking Forward to the Reviewer's Reply**
> > > > >
> > > > > Dear reviewer wbRQ,
> > > > >
> > > > > As the rebuttal deadline approaches in two hours, we kindly ask if our responses have adequately addressed your concerns. We would greatly appreciate your feedback to ensure we have fully resolved any outstanding issues. Thank you for your time and consideration!

---

> > > > > > ### Author Response · Authors · 2024-12-03
> > > > > > **Response to Review wbRQ**
> > > > > >
> > > > > > Thank you very much for your kind review, and we are glad that you enjoyed our paper!

---

> ### Author Response · Authors · 2024-12-03
> **Response to Update from the Reviewer wbRQ**
>
> Dear reviewer wbRQ,
>
> Thanks very much for your update. We will address the questions raised by the review as below.
>
> >**[Q1]** However, why does the smoothed ReLU function exhibit the polynomial part in the early stage of training?
>
> **[A1]** As we mentioned in line 457 of our paper, due to our small initialization technique, the values of $A_{i,r}^{(t)}$ and $B_{i,r}^{(t)}$ will be sufficiently small such that the smoothed ReLU function exhibits the polynomial part in the early stage of training. This small initialization technique is widely used in literature of feature learning paper with smoothed ReLU activation, such as [1][2][3].
>
> >**[Q2]** In addition, I believe a comprehensive investigate on the effect of $\epsilon$ on feature learning can make the manuscript stronger.
>
> **[A2]** We sincerely thank the reviewer for the valuable suggestion. We will add a comprehensive investigate on the effect of $\epsilon$ on feature learning in the revision of our paper.
>
> Thanks for the positive feedback again, and we would greatly appreciate it if you would consider raising your score!
>
> **Reference**
>
> [1] Allen-Zhu, Z. and Li, Y. (2023b). Towards understanding ensemble, knowledge distillation and self-distillation in deep learning. In The Eleventh International Conference on Learning Representations.
>
> [2] Chidambaram, M., Wang, X., Wu, C. and Ge, R. (2023). Provably learning diverse features in multi view data with midpoint mixup. In International Conference on Machine Learning. PMLR.
>
> [3] Li, J., Pan, J., Tan, V. Y., Toh, K. C., & Zhou, P. (2024). Towards Understanding Why FixMatch Generalizes Better Than Supervised Learning. arXiv preprint arXiv:2410.11206.

---

### Author Response · Authors · 2024-11-24
**Global Response to Reviewers**

We sincerely thank all reviewers for their positive support and valuable feedback! We greatly appreciate the insightful reviews and the recognition of the significance of our contributions, the strength of our theory and experiments, and the clarity of our writing. Their feedback has been instrumental in helping us improve our manuscript. We have provided a revised manuscript (with text highlighted in blue) and outline our revisions in this global response:

 1. We have relaxed the definition of adversarial examples in lines 285–288, as mentioned by reviewer cqxX, where we no longer require the classifier $\boldsymbol{F}$ to correctly classify the clean data point $(\boldsymbol{X}, y)$.

 1. We have corrected typos in line 197, as pointed out by reviewer XDz5.

Once again, we thank all reviewers for their insightful and valuable comments.

---

### Meta-Review · Area_Chair_Wpqc · 2024-12-19

**Metareview:**

Summary: This paper studies the difference of adversarial training vs. normal training, by looking at the learned features on a two-layer network with smoothed ReLU activation. The paper shows that adversarial training tends to learn robust features, while normal training tends to learn non-robust features. The paper sends a similar take-away message as [1].

[1] Ilyas, Andrew, et al. "Adversarial examples are not bugs, they are features." Advances in neural information processing systems 32 (2019).


Strengths:
1. The paper analyzes the benefits of adversarial training in terms of learning robust features, with rigorous theoretical proofs.
2. Experiments are performed in alignment with the theoretical discovery.

Weaknesses:
1. It is unclear what new take-home message this paper sends compared with [1]. Adversarial training learns more robust features than normal training has been well-studied before (though this paper tries to provide a rigorous proof under certain assumptions).
2. Strong assumptions. The analysis is built upon a stylized data distribution and a two-layer smoothed ReLU convolutional neural network. This may limit its extension to broader applications, e.g., other data distribution and network architectures.
3. The theory does not show a relation with variable $\epsilon$, which is the adversarial attack budget.
4. The theoretical analysis of adversarial training is with just one step gradient ascent (FGSM), which is vulnerable to attacks in practice.

All 5 reviewers consistently vote for 6: marginally above the acceptance threshold. AC would follow reviewers' opinion and recommend for acceptance. However, AC would encourage the authors to take weaknesses into consideration in the future research.

**Additional Comments On Reviewer Discussion:**

All 5 reviewers consistently vote for 6: marginally above the acceptance threshold. AC would follow reviewers' opinion and recommend for acceptance.

The most significant concern by all reviewers is about the strong assumptions made in this paper. These include strong assumption on data distribution, network architecture, adversarial budget, definitions of robust and non-robust features, etc (even after the rebuttal). Given that this paper is with a rigorous theoretical analysis, AC believes some of these assumptions are acceptable as this is the first work with such rigorous theoretical analysis on neural networks.

---

### Decision · Program_Chairs · 2025-01-22

Accept (Poster)